# Human neural dynamics of real-world and imagined navigation

Martin Seeber [1] ✉, Matthias Stangl [1,2], Mauricio Vallejo Martelo [1], Uros Topalovic [1], Sonja Hiller [1], Casey H. Halpern[3,4], Jean-Philippe Langevin [5,6], Vikram R. Rao[7], Itzhak Fried [1,5], Dawn Eliashiv[8] & Nanthia Suthana [1,5,9,10] ✉

The ability to form episodic memories and later imagine them is integral to the human experience, influencing our recollection of the past and envisioning of the future. While rodent studies suggest the medial temporal lobe, especially the hippocampus, is involved in these functions, its role in human imagination remains uncertain. In human participants, imaginations can be explicitly instructed and reported. Here we investigate hippocampal theta oscillations during real-world and imagined navigation using motion capture and intracranial electroencephalographic recordings from individuals with chronically implanted medial temporal lobe electrodes. Our results revealed intermittent theta dynamics, particularly within the hippocampus, encoding spatial information and partitioning navigational routes into linear segments during real-world navigation. During imagined navigation, theta dynamics exhibited similar patterns despite the absence of external cues. A statistical model successfully reconstructed real-world and imagined positions, providing insights into the neural mechanisms underlying human navigation and imagination, with implications for understanding memory in real-world settings.

Human cognition is a complex interplay of processes, with the formation of episodic memories and the imaginative capacity to revisit, manipulate and extrapolate from these memories representing two of its most fundamental aspects. These cognitive functions lie at the core of human experience, shaping our recollection of past experiences and our ability to construct visions of the future. While a substantial portion of research focuses on understanding the neural dynamics underlying these processes in the context of spatial navigation in freely moving rodents, it is widely held that the medial temporal lobe (MTL), particularly the hippocampus, plays a central role[1,2]. Previous research in rodents has identified 'place cells' tuned to specific locations[3–5], forming neuronal sequences spanning entire movement trajectories[6–9]. Importantly, these neuronal sequences persist during immobile periods[10–13], implying internal generation and making them well suited for organizing episodic memories[8,14,15], mentally simulating these memories[16] and planning future behaviours[8,17].

[1]Department of Psychiatry and Biobehavioral Sciences, Jane and Terry Semel Institute for Neuroscience and Human Behavior, University of California Los Angeles, Los Angeles, CA, USA. [2]Department of Biomedical Engineering and Department of Psychological and Brain Sciences, Center for Systems Neuroscience, Cognitive Neuroimaging Center, Neurophotonics Center, Boston University, Boston, MA, USA. [3]Department of Neurosurgery, Stanford University School of Medicine, Stanford, CA, USA. [4]Department of Neurosurgery, Perelman School of Medicine, Richards Medical Research Laboratories, Pennsylvania Hospital, University of Pennsylvania, Philadelphia, PA, USA. [5]Department of Neurosurgery, David Geffen School of Medicine, University of California Los Angeles, Los Angeles, CA, USA. [6]Neurosurgery Service, Department of Veterans Affairs Greater Los Angeles Healthcare System, Los Angeles, CA, USA. [7]Department of Neurology and Weill Institute for Neurosciences, University of California San Francisco, San Francisco, CA, USA. [8]Department of Neurology, University of California Los Angeles, Los Angeles, CA, USA. [9]Department of Psychology, University of California Los Angeles, Los Angeles, CA, USA. [10]Department of Bioengineering, University of California Los Angeles, Los Angeles, CA, USA. ✉e-mail: seeber@ucla.edu; nanthia@ucla.edu

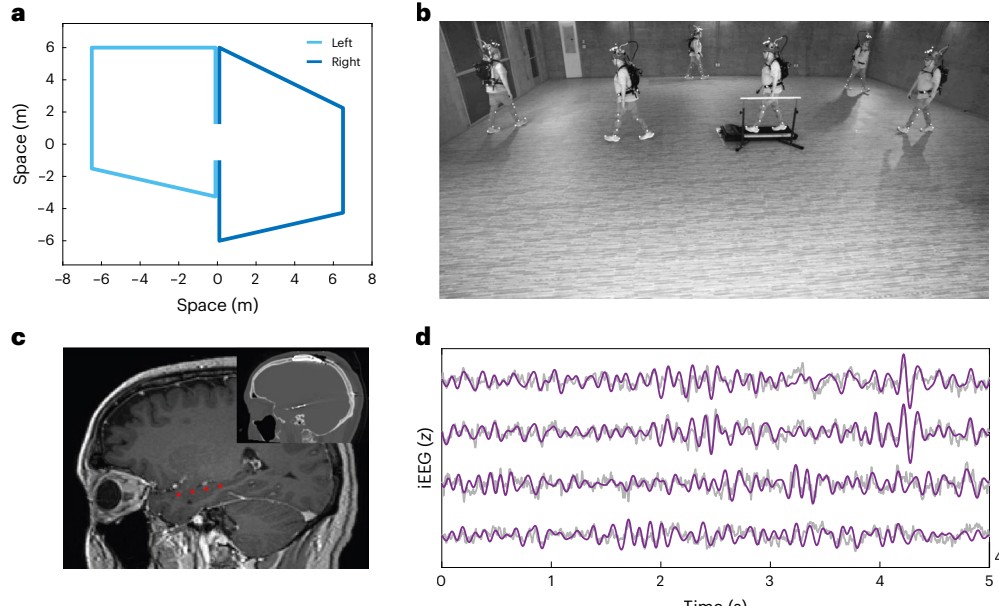

**Fig. 1 | Experimental paradigm. a**, Participants completed a spatial navigation task that involved the learning of two distinct routes: a leftward route represented in light blue and a rightward route represented in dark blue. **b**, An illustrative time-lapse motion capture shows a selected participant navigating the rightward route. Treadmill walking trials were interspersed with real-world walks. During treadmill walking, participants were instructed to mentally simulate their previous or upcoming route. Initially, before their real-world navigation walks, participants engaged in treadmill walking without the additional component of imagined navigation. **c**, Preoperative magnetic resonance image and postoperative computed tomography (inset) were used to identify the location of intracranial electrode contacts. The electrode locations of four electrode contacts (comprising two bipolar channels) in the hippocampus from an example participant are indicated in red. **d**, iEEG was continuously recorded during both real-world and imagined navigation. Exemplary broadband iEEG activity (grey) during real-world navigation is superimposed with the filtered signal (magenta) within the theta band (4–12 Hz).

However, it remains unclear whether such mechanisms exist in humans during real-world ambulatory navigation and especially during overt, on-demand imagination or re-experiencing of episodic memories.

To coordinate hippocampal neuronal sequences[6,7,18–21], theta oscillations within the frequency range of ~4–12 Hz[22,23] hold significance, as their disruption can lead to spatial memory impairments[2,24,25]. In humans, hippocampal theta oscillations manifest in brief bouts rather than continuously, as observed in rodents, during both virtual and real-world navigation[26–29]. This cross-species difference raises uncertainty regarding whether these intermittent bouts in humans can adequately support neuronal sequences and subsequent segmentation of distinct events during real-world navigation. Furthermore, whether hippocampal theta bouts can be internally generated and segment memories, such as during imagination, remains entirely unknown. Identifying such mechanisms would provide important insights into the shared neural organization principles between real-world spatial navigation and memory processes[8,14], effectively bridging decades of findings across species and integrating research on spatial navigation and memory.

In this Article, we compare theta dynamics in the human MTL between real-world and imagined navigation. Imagined navigation involved memories characterized by distinct temporal features, facilitating a direct comparison to real-world navigation. To explore this, we examined intracranially recorded neural oscillations from five participants who had undergone chronic implantation of the responsive neurostimulation (RNS) device, RNS System (NeuroPace), for the treatment of epilepsy[30]. Specifically, we directly compared MTL theta dynamics during real-world navigation with periods where participants mentally simulated navigating the exact same routes while walking on a treadmill.

We observed theta dynamics within the MTL that varied depending on the participant's position within segments of the navigational routes, effectively encoding spatial aspects of those routes. These theta dynamics demonstrated temporal consistency across individual trials, were consistently observed in all participants and occurred during imagined navigation while walking on a treadmill. We further demonstrated the feasibility of estimating relative positions within navigational route segments using theta dynamics of real-world navigation and applied this model to accurately reconstruct imagined positions.

## Results

### Theta dynamics during real-world navigation

We investigated whether neural dynamics in the human MTL are influenced by position, time or progress on specific spatial routes, building upon previous research[5,8,10,12,14,31–33]. To achieve this, we tasked participants with walking along two distinct spatial trajectories (Fig. 1a,b and Extended Data Fig. 1a). Since the angles and locations of turns on these routes were not visible during real-world navigation, participants had to learn and remember the walking patterns, with turns serving as critical retrieval points. Theta dynamics (3–12 Hz) were extracted from intracranial electroencephalogram (iEEG) data recorded from chronically implanted electrodes (Fig. 1c,d). All participants were able to successfully learn the navigational routes as instructed (Fig. 2a and Extended Data Figs. 1b and 2). By synchronizing and aligning iEEG and motion capture data, we were able to compare theta dynamics with the positions along each spatial route.

We observed an increase in MTL amplitudes within the theta-frequency range, most pronounced at upcoming turns just before the completion of a linear segment within a walking route (Fig. 2b–d and Supplementary Video 1). This pattern remained consistent across all five participants while they navigated both left and right walking routes (Extended Data Figs. 3 and 4a). Interestingly, participant 2 (P2) consistently incorporated an additional turn not originally part of the route (Extended Data Fig. 2), and this effect was still observed (Extended Data Fig. 3). We further confirmed that theta activity aligned to route segments through time–frequency analyses (Fig. 2d) (see Supplementary

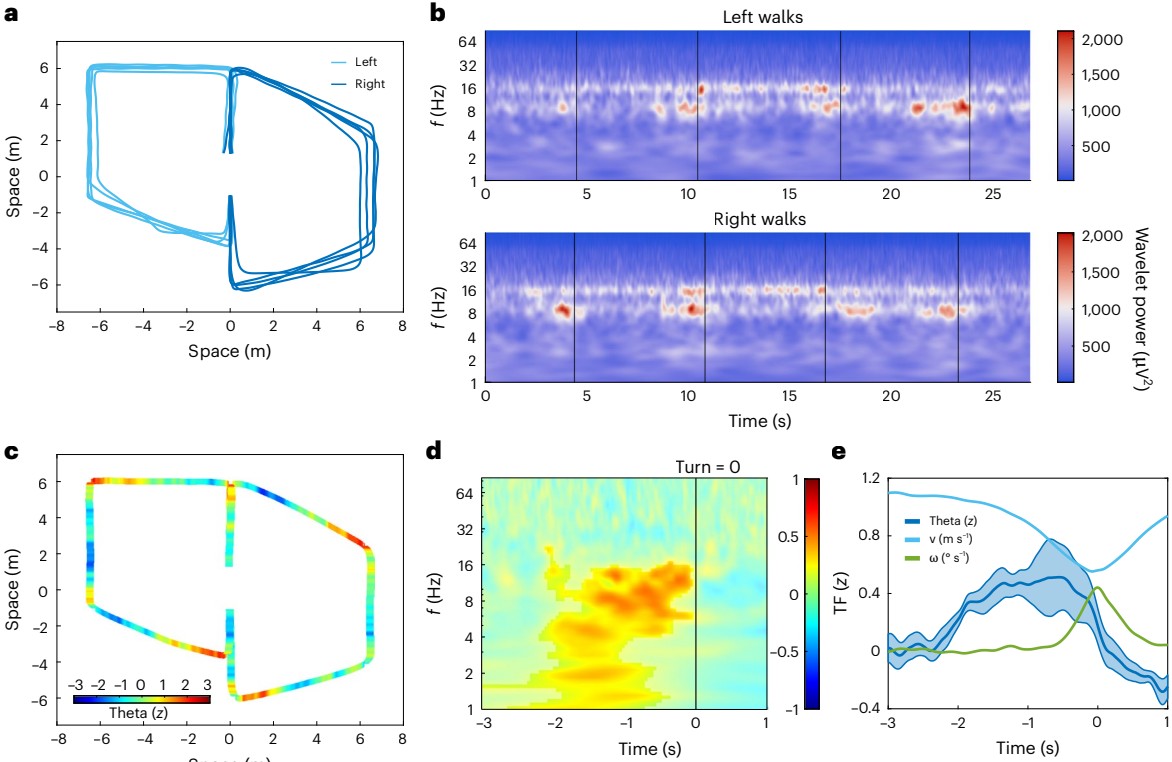

**Fig. 2 | Theta dynamics during real-world navigation. a**, Distinct lines illustrate the average walking routes of each individual participant. **b**, Time–frequency plots of exemplary hippocampal activity averaged across trials reveal task-related and temporally organized theta oscillations during both left (top) and right walks (bottom). The black vertical lines demarcate where turns occurred in the routes. **c**, Theta activity (z-scored) within the left anterior hippocampus (group average, one channel per participant) is overlaid onto the motion trajectories, demonstrating heightened theta power as participants approached upcoming turns. **d**, Average time–frequency (TF) activity aligned with all turns confirms the engagement and the temporal relationship of theta activity preceding turns (time = 0). The significant activity is indicated by saturated colours (two-sided cluster-based permutation test, $P < 0.001$), while non-significant activity is shown in faded colours. **e**, The mean theta activity ± standard error (see Supplementary Table 1 for frequency ranges) juxtaposed with the speed and angular velocity of the hips, averaged across the group and aligned to turns (time = 0).

Table 1, Extended Data Fig. 5a,c and Methods for for theta-frequency ranges at the subject-specific level). To investigate whether oscillatory bouts in the theta range were more related to physical turns or specific instances along the routes, we compared the first five walks during and after the experiment's learning phase. After participants learned to walk the routes as instructed, the prevalence of theta bouts preceding turns was significantly higher than during the learning phase ($r_d = 0.04$, 95% confidence interval (CI) 0.01 to 0.12, $P = 0.021$, $d = 0.68$; Extended Data Fig. 1c). Notably, theta activity reached its peak significantly before the actual physical turns occurred ($\Delta t = -0.94$ s, 95% CI −1.05 to −0.80, $P < 0.001$, $d = -1.09$; Extended Data Fig. 6a–c), as captured by hip rotation (Fig. 2e). As participants approached a turn, they naturally reduced their walking speed (Fig. 2e). Concurrently, there was a significant increase in power at theta frequencies (cluster-based permutation test, $P < 0.001$) preceding this deceleration (Fig. 2d,e), and this pattern consistently preceded other behavioural variables, including head rotation, body turning and eye movements (Extended Data Fig. 7a,b and Supplementary Table 2). The participants directed their gaze ahead as instructed during their walking path but tended to look more towards the left during leftward walks and to the right during rightward walks (Extended Data Fig. 8a). Although theta-range amplitudes increased before turns, we did not find evidence for such increases being time-locked to saccades (Extended Data Fig. 8b,c). Thus, these theta dynamics cannot be easily attributed to confounding factors such as walking speed, heading, hip rotation or eye movements.

To ensure that participants maintained awareness of their position during real-world navigation, we instructed them to report their position relative to previously learned locations following the presentation of an auditory cue (see the Methods for more details). This self-location task consistently induced activity in the theta and beta frequency ranges across all participants (Extended Data Fig. 9). An initial enhancement of power in the theta band was followed by a suppression of power in theta/alpha frequencies and an enhancement of beta/gamma power during the reporting phase, indicated by the participants via button press.

In addition to the activities at theta frequencies, we observed two additional spectral peaks in the delta (1–3 Hz) and beta (13–30 Hz) frequency ranges (Extended Data Fig. 5c). We analysed the activity within these frequency bands using the same approach as for theta, examining amplitude dynamics as a function of position during real-world navigation. The time-varying amplitudes in the delta range showed a dependency on route segments (Extended Data Fig. 10a) comparable with theta, while the beta frequency activity increased during the third route segment (Extended Data Fig. 10c). During this segment, the participants performed an additional task of identifying the closest hidden object at a random moment, signalled by an auditory cue.

Consistent with previous findings[26–29], theta activities appeared in brief bouts rather than as continuous oscillations, as seen in rodents. The mean prevalence of theta bouts was 21.2 ± 6.6%, with an average duration of 0.524 ± 0.077 s across participants similar to previous reports[27,28]. We analysed the timing of these bouts of increased amplitudes at theta frequencies on a single-trial level and observed their alignment with upcoming turns (Fig. 3b). Strikingly, the percentage of trials in which theta bouts occurred at specific time points closely mirrored the trial-averaged amplitude dynamics in the theta-frequency range (Extended Data Fig. 5b). This finding suggests that theta bouts

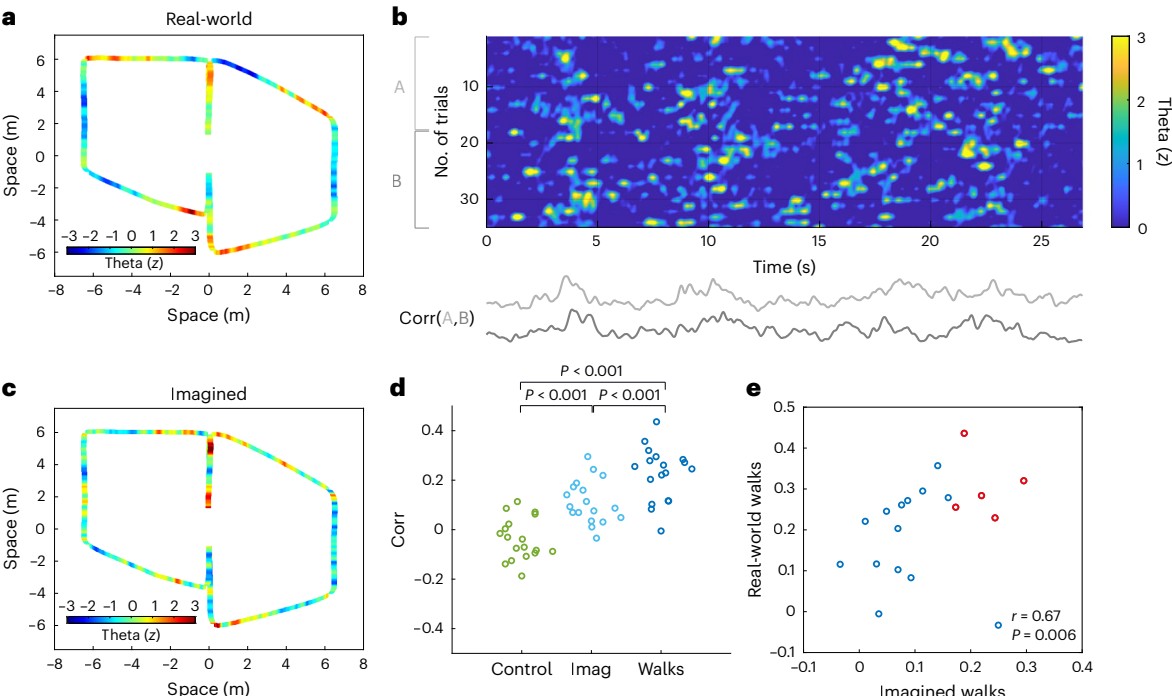

**Fig. 3 | Comparative theta dynamics of real-world and imagined navigation. a**, The theta activity averaged across the group during real-world navigation superimposed on the mean motion trajectory of all participants. **b**, The temporal consistency of theta dynamics was assessed by calculating the correlation (Corr) between the mean signals obtained from randomly dividing the data (trials) into two halves (A and B). **c**, The theta activity during imagined navigation of an exemplary channel rendered on the group mean motion trajectory. **d**, Temporal consistency of theta dynamics was significantly higher during both imagined (Imag) and real-world (Walks) navigation (one-sided permutation test, family-wise error corrected) when compared to sole treadmill walking (Control). The data from each of the 18 intracranial electrode channels is represented by individual circles. **e**, Notably, temporal consistency was spatially correlated (two-sided linear mixed-effect model, $\beta = 0.75$, d$f$ of 16, 95% CI 0.25 to 1.24, $P = 0.006$) between real-world and imagined navigation. The electrodes showing higher consistency in theta dynamics during real-world navigation also demonstrated comparable consistency during imagined navigation, suggesting the presence of analogous anatomical regions engaged in both types of navigation. Interestingly, the highest consistency electrodes were located in the left anterior hippocampus (red circles), in contrast to other regions within the MTL (blue circles).

exhibited task-related alignment across trials. More specifically, these bouts aligned with particular instances along the navigational routes, effectively dividing participants' movement trajectories into distinct linear segments.

## Shared neural dynamics in real and imagined navigation

The dynamics within the hippocampus are known for their capacity to encode task-related structures associated with position, time and, in a broader sense, sequences. Building upon the knowledge that hippocampal neuronal dynamics can be internally generated[10–13], we investigated whether MTL oscillatory dynamics exhibit similarities during real-world and imagined navigation. To explore this, we incorporated intervals of real-world navigation trials interspersed with treadmill walking periods. During the initial treadmill walks (control trials), participants were not given any explicit instructions other than to walk at a steady speed. However, on the later treadmill walks (imagination trials), they were instructed to recollect their previous or upcoming routes while walking at the same steady speed. Each treadmill condition consisted of 24 trials, resulting in a total of 72 trials (24 trials each for control, imagination left route and imagination right route) (Supplementary Table 1).

Given the pronounced presence of task-related structured theta dynamics during real-world navigation, we explored whether analogous temporal patterns were evident during imagined navigation (Fig. 3a,c). Our analysis showed that the temporal consistency across trials (Fig. 3b) was significantly higher for theta dynamics during imagined navigation compared with the control condition ($r_d = 0.15$, 95% CI 0.09 to 0.26, $P < 0.001$, $d = 1.83$) of sole treadmill walking (Fig. 3d and

Extended Data Fig. 4b). The highest level of temporal consistency was observed during real-world navigation ($r_d = 0.26$, 95% CI 0.22 to 0.36, $P < 0.001$, $d = 3.06$), whereas this consistency was absent during sole treadmill walking ($P = 0.964$). We did not find an effect of the time reference (previous/upcoming route) of imaginations but a modest ($r_d = 0.18$, 95% CI 0.12 to 0.24, $P = 0.023$, $d = 1.51$) session effect, consistent with learning throughout the experiment (Extended Data Fig. 6e,f). However, we did not find evidence for a time-on-task effect when comparing the first and second halves of control trials (Extended Data Fig. 6g). Next, we examined the congruence of functional anatomy within the MTL during both real-world and imagined navigation. Our analysis revealed a spatial correlation in the temporal consistency values (linear mixed-effect model, $r = 0.67$, $\beta = 0.75$, d$f = 16$, 95% CI 0.25 to 1.24, $P = 0.006$) between these conditions. Notably, recording channels demonstrating heightened temporal consistency during real-world navigation, such as within the left anterior hippocampus, were found to manifest a similar pattern during imagined navigation, suggesting the involvement of comparable anatomical regions within the MTL in both modes of navigation (Fig. 3e).

In addition to the theta-frequency band (Fig. 3d), temporal consistency across trials was significantly higher during real-world navigation compared with the control condition for the delta ($r_d = 0.29$, 95% CI 0.20 to 0.38, $P < 0.001$, $d = 1.80$) (Extended Data Fig. 10b) and beta ($r_d = 0.20$, 95% CI 0.15 to 0.25, $P < 0.001$, $d = 2.21$) (Extended Data Fig. 10d) frequency bands. Delta frequency dynamics during imagined navigation were more consistent compared with the control condition ($r_d = 0.14$, 95% CI 0.03 to 0.25, $P = 0.032$, $d = 0.87$) (Extended Data Fig. 10b), whereas beta frequencies did not show consistent dynamics

($P$ = 0.247) (Extended Data Fig. 10d). Furthermore, the effect size (Cohen's $d$) of temporal consistency differences between imagined navigation and the control condition was more pronounced in the theta range ($d$ = 1.83) than in the delta range ($d$ = 0.87).

To validate participants' engagement in imagining the distinct navigational routes, we examined their gaze. During real-world navigation, participants tended to focus their gaze ahead on the walking path, resulting in relatively higher probabilities of left or right gazes for leftward and rightward walks, respectively. While this effect was less pronounced during imagined navigations, gaze positions still subtly differentiated between imagined left and right routes (Extended Data Fig. 8a). Lastly, we did not find a direct effect of eye velocity on theta amplitudes during imaginations ($P$ = 0.568) nor evidence for increased theta amplitudes time-locked to saccades during either imagined navigation or control trials (Extended Data Figs. 7d and 8d,e). We next explored the feasibility of estimating positions of real-world walks from theta dynamics and assessed whether such reconstructions could generalize to imagined navigation.

## Reconstructing imagined positions

We examined the feasibility of reconstructing participants' route progression based on neural activity patterns. This exploration was motivated by the observation that both real-world and imagined navigation induced temporally structured theta dynamics within the MTL. To this end, we investigated whether these theta dynamics encoded the inherent task structure of the routes as illustrated in Fig. 4a. These routes were designed with five linear segments connected by four turns. After adjusting for variations in segment lengths, we found that theta-range amplitudes were consistently similar at corresponding relative positions within each segment (Fig. 4e). However, distinct recording electrodes exhibited peak activations at slightly varying positions within a route segment (Fig. 4f). In light of this finding, we developed a statistical model that considered the unique timings of each electrode to capture the relative positions of route segments, treating them as two-dimensional circular variables, a methodology based on prior research[34]. Modelling the linearized position as a circular variable is a useful mathematical abstraction where movement can be thought of as a change of position on a ring. In our case, each full circle on that ring would correspond to the traversal of one route segment. This approach allows us to capture the continuous, cyclical modulation of theta activity across route segments, which can vary smoothly over time. By using sine and cosine functions to model theta dynamics, we account for these cyclical changes more effectively. Using a linear regression model, we used theta dynamics from all MTL channels, pooled across participants, as predictors, with the route structure (comprising cosine and negative sine (−sine) phase alignment) as the response variables. Regression coefficients were learned from subsets of the data, and the model's ability to generalize to unseen trials was tested (using 10 × 10 cross-validation) to prevent overfitting. Our results indicated that relative segment positions could be reconstructed better than chance during real-world navigation ($\delta$ = 0.63, 95% CI 0.61 to 0.64, $P$ < 0.001). The performance of the reconstruction was evaluated as normalized probability densities for each relative position within a segment. The time-resolved reconstruction revealed that position estimates matched not only at upcoming turns but encompassed the entire segment of a route (Fig. 4g). Overall reconstruction errors were quantified as the angle between actual and estimated positions because relative positions were treated as circular variables (Fig. 4h). In these main analyses, we used identical route structures for all participants. However, to account for the specific route followed by P2, we conducted a supplemental analysis that adapted the model to include an additional turn. This adaptation yielded similar results (Supplementary Fig. 1), supporting the robustness of the model across participants.

We applied the main position reconstruction model, which was initially trained on real-world walking data, to imagined navigation

data. In real-world walking, position and time are linearly related due to well-defined walking velocities. However, in imagined navigation, the imagined velocity is not directly available. We also anticipated that the timing of imagination may not precisely match that of real walks. While the task structure remains consistent during imagination, route segments may undergo stretching and bending. To address this, we aligned estimated imagined positions and the task structure using linear time warping (see Supplementary Table 3 for individual participant time shifts). After alignment, we found that the position estimation model effectively generalized to imagination trials (Fig. 4b,c) not used in the model's training and alignment procedure (10 × 10 cross-validation). The reconstruction errors for imagined positions were significantly smaller ($\Delta\delta$ = −0.43, 95% CI −0.46 to −0.40, $P$ < 0.001) than those derived from the control condition, where imaginations were absent but physical behaviour was identical (Fig. 4g,h). Theta dynamics exhibited a significant correlation between real-world and imagined navigation ($r$ = 0.30, 95% CI 0.28 to 0.31, $P$ = 0.003) (Fig. 4d and Extended Data Fig. 6h) and were related to the route structure (Fig. 4a) for both real-world ($r$ = 0.51, 95% CI 0.49 to 0.52, $P$ < 0.001) and imagined navigation ($r$ = 0.29, 95% CI 0.28 to 0.31, $P$ = 0.010). However, no such correlation was observed during treadmill walking alone ($r$ = −0.01, 95% CI −0.03 to 0.01, $P$ = 0.528). We observed that theta dynamics were primarily related to the relative position within the route segments, mirroring the patterns observed during real-world navigation (Extended Data Fig. 7c,d). Further, the impact of estimated relative position during imagined navigation was significant ($\beta_d$ = 0.02, 95% CI 0.01 to 0.04, $P$ = 0.006, $d$ = 1.93), while evidence for other behavioural variables was absent (head rotation: $P$ = 0.051, hip rotation: $P$ = 0.982, eye movement: $P$ = 0.568, speed: $P$ = 0.92), suggesting that these other factors did not significantly impact theta dynamics. Cross-correlations between theta-range dynamics and route layout, after normalizing for segment length discrepancies, revealed four lateral peaks at multiples of the segment length, corresponding to the four turns on each walking route (Fig. 4e). These results indicate that theta dynamics within distinct segments were similar and repeated themselves along the entire movement trajectory when accounting for varying segment lengths.

## Discussion

Using rare mobile iEEG recordings in freely moving human participants, we demonstrated that intermittent theta dynamics in the human MTL are well suited to internally organizing the recollection of past episodes or imagination of future behaviours. Specifically, our findings revealed that transient oscillations in theta-frequency range aligned at upcoming turns during real-world navigation, moments, which served as critical retrieval points essential for transitioning between navigational route segments. These dynamics exhibited a similar temporal pattern during imagination of previous and upcoming routes. The presence of comparable characteristics during both real-world and imagined navigation suggests neural progression along abstract structures delineating multipartite imagined routes into more elementary sections. Notably, this repetitive temporal structure was absent when individuals engaged in treadmill walking alone, where mental recapitulation of previously learned spatial routes was absent despite identical physical behaviour.

Our finding of structured theta bouts during imagined navigation suggests that hippocampal networks can internally generate neural dynamics representing distinct sections of a task. This task structure may involve sequences of route segments linked by action points, such as turns observed in real-world navigation or event timing during imagination and cognitive tasks with non-spatial components. These findings align with prior rodent studies demonstrating the ability of hippocampal networks to internally generate neuronal sequences[10,11], with theta oscillations crucial for their temporal organization[6,7,18–21]. It is conceivable that the transient oscillations at theta frequencies we report here in humans could reflect neuronal

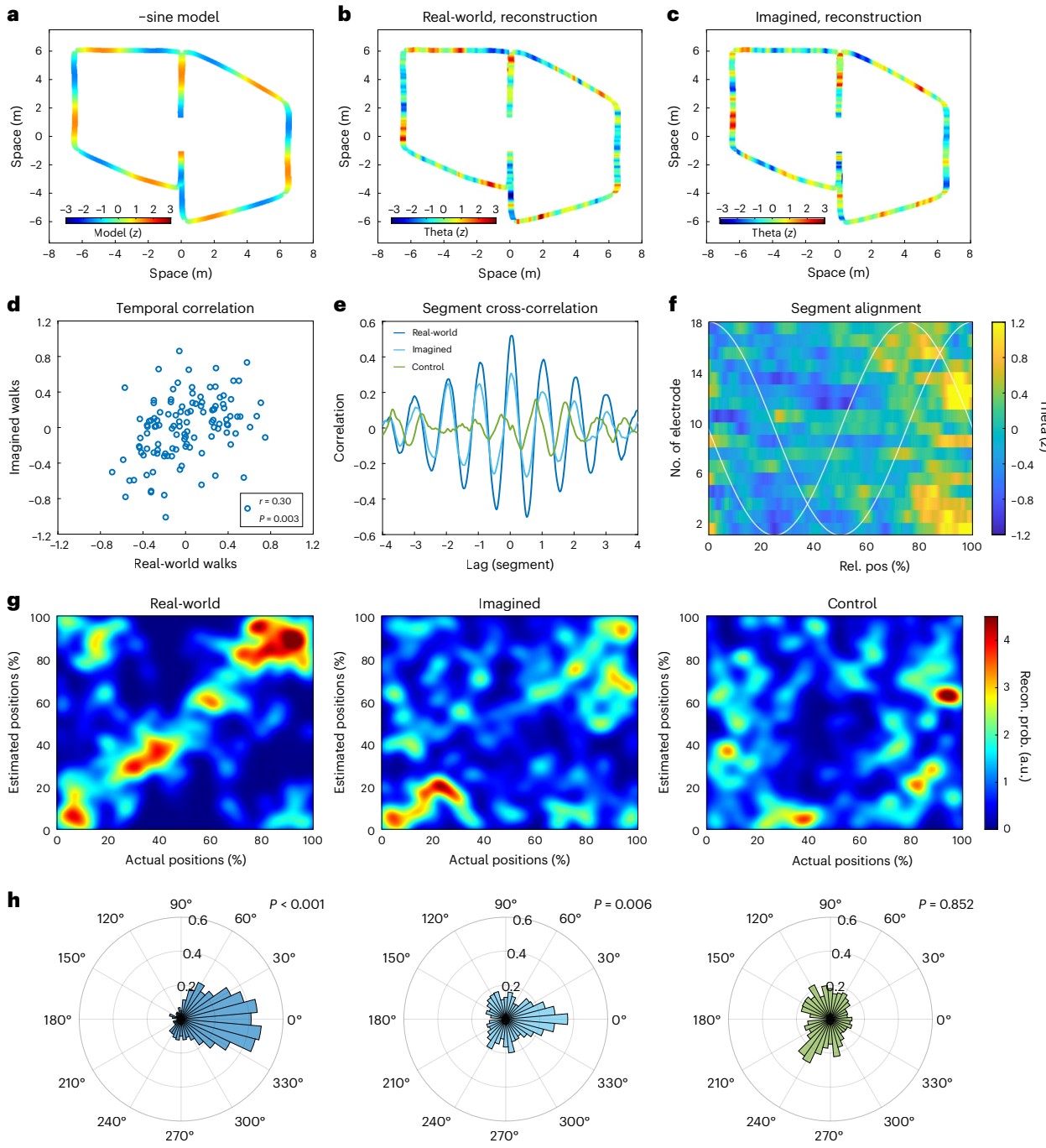

**Fig. 4 | Reconstructing relative position from neural dynamics. a**, The model of the navigational route structure is represented as a −sine pattern peaking before turns. **b**, Reconstruction of this −sine route representation from theta dynamics during real-world navigation. **c**, Reconstruction of the −sine route representation from theta dynamics during imagined navigation. **d**, Scatterplot of the reconstructed route representations of real-world and imagined navigation (one-sided permutation test, $r = 0.30$, 95% CI 0.28 to 0.31, $P = 0.003$). Each dot represents data averaged over 0.5-s time bins, derived from data pooled across all channels. **e**, A cross-correlation analysis between theta dynamics and the route structure (adjusted for segment length) revealed a significant correlation (lag of 0) for both real-world and imagined navigation, while sole treadmill walking did not exhibit such a correlation. The similarity and repetition of theta dynamics in each route segment led to four lateral peaks in the cross-correlations at lags that matched the length of one segment on a walking route. **f**, The theta dynamics are illustrated as a function of relative position within all segments across all 18 electrode channels. Notably, the timing of theta dynamics varied across channels, roughly following a cosine or −sine wave pattern (white lines). These two orthogonal theta modulations effectively encoded relative

position as a circular variable. **g**, Reconstruction of relative segment positions from theta dynamics depicted using 2D histograms that show both the actual and estimated positions. The colour-coded representations illustrate the probabilities of all possible combinations of actual and estimated positions. An ideal reconstruction outcome would manifest as a diagonal pattern. Left: during real-world navigation, the estimated positions (cross-validated) clustered closely around the actual physical positions within each segment. Middle: during imagined navigation, the estimated positions also aligned consistently with the positions estimated from the duration of the imagination periods particularly at the beginning and end of each route segment, illustrating heightened accuracy in those instances. Right: however, during sole treadmill walking where imagined navigation was absent, reconstruction failed to yield accurate results. **h**, Histograms depicting the errors in reconstruction (measured in degrees as a circular variable) for each condition revealed distinct patterns. In both real-world (left) and imagined (middle) navigation conditions, the errors clustered around zero, indicating accurate reconstruction (one-sided circular permutation test). However, during sole treadmill walking (right), this clustering around zero was absent, signifying less accurate reconstruction.

populations overrepresenting forthcoming turns, marking the transitions between segments. Indeed, hippocampal place cells and other spatially tuned cell types can bias their activity towards behaviourally relevant locations or moments[12,13,35,36]. In navigational tasks featuring repetitive segments, such as ours, place cells and grid cells tend to form repetitive firing sequences that reset at turning points within a maze[33]. Task-related theta dynamics observed during imagined navigation also complement recent research suggesting that hippocampal theta oscillations play a key role in dynamically exploring locations related to both current and hypothetical behaviours in rodents[20,37]. Internally generated dynamics such as these are particularly relevant for episodic memory retrieval, where sensory input may be minimal.

These findings also resonate with studies on event boundaries delineating ongoing experiences[38–40]. In our current study, turns can be interpreted as event boundaries, which are predictable since they are defined by participants' actions. This partitioning of navigational routes into segments also deepens our understanding of the human MTL, showing oscillations in the theta range can adhere to abstract route structures rather than physical environmental constraints. Such partitioning might enable flexible combination of previously experienced fragmented episodes to adapt to dynamic real-world task demands. Moreover, the reassembly of neural sequences through transitions might facilitate hippocampal circuitry to generate novel trajectories relevant to envisioning future behaviours. Theta bouts preceding turns also complement prior findings of theta bouts occurring after cognitive boundaries during movie watching[41]. While self-generated actions are predictable, the cuts between movie scenes are not, explaining the temporal alignment of theta bouts before versus after such boundaries in proactive versus passive behavioural paradigms.

Recent studies on simulated navigation have indicated that memory is a primary driver of hippocampal theta oscillations in humans[42]. Our study goes further to describe the fine-grained temporal structure of theta dynamics, which partitioned participants' trajectories into discrete segments during real-world and imagined navigation. We did not find differences in the temporal consistency of imagined previous versus upcoming routes. These results suggest that MTL dynamics are similar between actual, recapitulated and possible future behaviours. It is worth noting that we instructed participants to imagine their movements along the navigational routes on the same time scale as during real-world navigation. Episodic memory recall is typically temporally compressed[42], potentially leading to more events and theta bouts per second, resulting in elevated average theta power. Moreover, increased theta power has also been reported during spatial memory retrieval in stationary, view-based navigational tasks[43,44]. Such work has also reported that amplitudes in the theta range increase before and during movement along longer virtual paths[45,46] but decrease as participants approach goals[47]. These findings are consistent with our results, suggesting that participants may have simulated the trajectory ahead of them at movement onset[46,47] or before a turn leading to the next real or imagined route segment in our study. This may be similar to how rodents tile the paths ahead of them[6,7,9,48] or how humans navigate virtual paths towards distinct goals[49,50] and experience episodic progression[19,41].

Imaginations are often intertwined with previous experience, making it challenging to fully separate processes related to imagination and memory retrieval. While our experiment specifically included a condition where participants were asked to imagine navigating future routes, these trajectories had been previously experienced and were not novel. Nevertheless, we interpret and discuss our findings as imagined navigations, as their mental progressions along routes pertain to future or hypothetical scenarios[17,51,52].

Consistent with functional neuroimaging research on egocentric navigation strategies[53], sequence representation[54] and imagination of events[55,56], our study identifies the most structured activity in the left anterior hippocampus. Although we ensured coverage of MTL regions during participant recruitment, electrode placements were determined solely based on clinical treatment criteria. As a result, implantation sites varied across participants within the MTL. Apart from the anatomical cluster in the left anterior hippocampus, where we observed functional similarities across all participants, the variability in electrode placement limits our ability to make subregional MTL functional comparisons in this study.

In line with previous literature on hippocampal recordings in humans, we observed spectral peaks at ~4 and ~8 Hz, at the edges of the traditional theta-frequency range[22,26,27,57]. Consequently, we used individualized theta-frequency ranges within the broader 3–12 Hz range. Although these frequencies partially overlap with scalp electroencephalogram (EEG) alpha rhythms (Supplementary Fig. 2), oscillations around ~8 Hz in the MTL are considered functionally distinct from alpha activity recorded over visual and sensorimotor areas on the scalp[58,59]. Further research is needed to clarify the relationship between MTL theta-range activity and scalp recordings. Outside the theta range, we observed spectral peaks around 2 Hz in the delta range and approximately 16 Hz in the beta range. The delta-range activities overlap with the 'slower theta' rhythms reported in episodic memory tasks[22,60] and view-based navigation in virtual environments[57,61]. This overlap may explain the structured dynamics at 1–3 Hz during both real-world and imagined navigation. However, temporal consistency effects during imagined navigation were more pronounced in the theta range than in the delta range. Beta-range activities may be linked to the self-location subtask. Increased MTL oscillatory activity in the beta range has been reported during fast walking versus slow walking[27], while decreased activity in the beta range has been reported during episodic memory retrieval in stationary participants[22]. In addition, a recent study in freely moving rodents suggested that beta activities might result from two independent theta-frequency inputs representing forward and reverse trajectories at roughly opposite theta phases[21]. The precise nature of these beta activities remains an open question, with ongoing debate about whether they are related to sensory processing[22], reflect harmonics due to changes in the theta oscillation shapes[27] or result from two distinct theta-frequency inputs to the hippocampus[21].

Our findings open exciting avenues for future research in real-world spatial navigation, episodic memory and imaginable behaviours. The non-continuous nature of human theta oscillations reported in our study presents an opportunity to delve into the timing and structure of these transient oscillations in various cognitive tasks and behaviours. Subsequent studies could elucidate the neuronal generators of hippocampal theta bouts by simultaneously examining single neurons and local field potentials in freely moving humans[62], which could shed light on the relationship between distinct functional cell types, theta bouts and different navigational strategies. Real-world spatial navigation paradigms, in particular, offer a valuable platform for uncovering neural activation patterns relative to task-specific variables. These patterns enable comparisons between navigational and cognitive tasks with similar temporal organization. In studies involving human participants, imaginations can be explicitly instructed and verbally reported, allowing for the investigation of the neural mechanisms underlying abstract, hypothetical or unprecedented future scenarios, paralleling real-world behaviours.

In summary, our findings highlight the role of transient theta oscillations in partitioning complex navigational routes and episodes into discrete segments, showing a striking similarity in real-world and imagined navigation. This suggests a common neural organization framework underlying both types of navigation independent of environmental constraints. The close resemblance between real-world and imagined navigation aligns with the notion that the neural mechanisms and functional architecture of memory may have evolved from those initially developed for spatial navigation[8,15]. Together with our findings, these parallels underscore the possibility that MTL dynamics

contribute to the temporal organization across various cognitive domains, including spatial navigation, episodic memory and the contemplation of conceivable future scenarios.

## Methods

### Participants

Five participants (24–40 years old; three males and two females) who had been chronically implanted with the US Food and Drug Administration-approved RNS System (NeuroPace) for the treatment of pharmaco-resistant focal epilepsy[30] volunteered for this study. The electrode placements were determined exclusively by clinical treatment criteria. The individuals with electrodes implanted in the MTL and low rates of epileptic activities were recruited for the study. Notably, participants P1 and P4 have contributed to previous studies, with P1 participating in Stangl et al.[28] and P4 involved in Maoz et al.[29]. the participants received financial compensation for taking part in this study. Before participation, all individuals willingly provided informed consent, following a protocol approved by the UCLA Medical Institutional Review Board.

### Real-world and imagined navigation tasks

The participants were tasked with traversing two routes (Fig. 1a,b) within an indoor room (measuring 14.6 × 13.5 m$^2$), each route featuring four turns. Initially, these route shapes were displayed on a tablet screen, and precise tracking of participants' walking patterns was achieved through motion capture technology. At the beginning of each recording, participants learned to walk along these patterns without any visible cues indicating the route's shapes or the positions of the turns. After each walking trial, feedback was provided by overlaying their real movement trajectories onto the prescribed ideal routes (Extended Data Fig. 1a) displayed on a tablet interface, allowing them to improve their behaviour in the next trial. This initial learning phase included six life-size paper cut-out objects (three per route) positioned along the path. During the third walking segment of each left or right route (following the second turn), an auditory cue was introduced through a wireless loudspeaker after random delays. The participants were then instructed to determine which object they were closest to by pressing the appropriate left, middle or right button on a wireless handheld mouse, corresponding to the first, second or third object on their route. A second auditory cue signalled the accuracy of their response after the button press. This learning phase concluded once participants demonstrated consistent proficiency in navigating these routes. For most participants, this was evidenced by a gradual reduction in error between the actual and ideal routes over successive trials (Extended Data Fig. 1). However, for P2, after completing eight left and eight right learning trials and despite receiving multiple instructions, it became evident that their walking routes did not improve notably from trial to trial. Recognizing this and respecting P2's individual learning process, we instructed them to maintain consistency in their chosen routes going forward, even if these routes incorporated deviations such as the extra turn mentioned. This approach ensured that P2 could still participate in the main experiment while reflecting their unique navigation strategy. At this point, for P2 and after the other participants improved their performance, the objects were then removed, requiring participants to rely on memory for walking routes and object locations. After the learning phase, and during the main experiment, visual feedback was limited and provided only after completing three left and three right walks, serving as periodic indicators of their navigation performance.

Real-world navigation trials were interspersed with periods of treadmill walking. These treadmill periods were organized into three distinct blocks over the course of the experiment. Each block consisted of 12 left and 12 right real-world walks, alternating with 24 treadmill walks. Every real-world walk was either preceded or followed by a treadmill walking trial. During the first block of the experiment, participants engaged in treadmill walking devoid of any supplementary tasks. This experimental condition was consistently initiated first, ensuring a baseline of unspecific treadmill walking. Notably, the mention of imagined navigation was omitted to guarantee a naive experience of sole treadmill walking. In the subsequent two blocks of the experiment (second and third), the participants were instructed to mentally navigate their preceding or upcoming walking routes while walking on the treadmill. The utilization of imagined previous and upcoming routes served to investigate the distinction between remembering past spatial trajectories and simulating future ones. During all treadmill walking trials, participants were instructed to walk naturally and face their gaze forward.

These later two blocks were randomized across participants so that they imagined their previous/upcoming routes during the second/third block, respectively. The imagination periods commenced upon activating the treadmill, and the participants signalled the completion of their imagined routes by pressing a button on a wireless handheld mouse. Overall, participants completed a total of 144 walks, including 36 left, 36 right and 72 treadmill walks. The treadmill trials were distributed as follows: 24 imagined left, 24 imagined right and 24 sole treadmill walks without any imagination involved. These imagination trials were further divided into 24 instances of imagining a previous route and 24 instances of imagining an upcoming route. The exact numbers of trials are listed in Supplementary Table 1.

### Unity application

We developed an application using Unity (version 2021.19f1) to control the experimental paradigm, initiate and stop motion tracking recordings, and capture participants' behavioural responses. To facilitate seamless communication, we established a two-way interaction between the Unity application and the motion tracking software (as detailed below). This enabled the Unity application to initiate motion capture recordings and concurrently receive real-time motion data. Motion tracking data were integrated into Unity, enabling the logging of participants' walking trajectories and positions. These data served as the foundation for visualizing both the actual and ideal walking trajectories, offering participants valuable feedback. For the self-location subtask, auditory cues were generated based on participants' positions. Using a wireless handheld mouse, participants' responses were collected and transmitted, subsequently being recorded and stored within the same Unity application for analysis.

### Motion capture

To capture participants' movements, we used the OptiTrack system (Natural Point). This setup involved thirty high-resolution infrared cameras strategically positioned on the walls. These cameras recorded the entire laboratory space at a rate of 120 Hz, ensuring precise tracking with submillimeter precision. The positions and orientations of the hips, legs and feet were tracked using a lower-body skeleton fitted to the Helen Heyes marker set. Moreover, head positions and orientation were captured using a rigid body. The gathered motion capture data were analysed and exported using Motive 3.0 software. Motion capture signals were smoothed with a Gaussian filter using a window size of 0.2 s for noise suppression. All walking trajectories are shown in Extended Data Fig. 2. The movement speed was calculated as positional change in metres per second. Head and hip rotation were quantified as absolute values of angular velocities derived from the head and hip angular orientations (Yaw).

### iEEG recording

The RNS System (NeuroPace), approved by the US Food and Drug Administration, was specifically designed to treat pharmaco-resistant focal epilepsy. This system is engineered to detect epileptic activity and respond in a closed-loop manner by delivering precise electrical stimulation, effectively mitigating the risk of seizures. Notably, the RNS

System is implanted chronically, affording the ability to record iEEG activity while individuals move freely[63]. In our study, every participant had two depth electrode leads, each comprising four electrode contacts spaced at intervals of 10 mm. We selected specific contacts in each individual on the basis of their location in the MTL to record from up to four bipolar channels in each participant. The data acquisition occurred at a sampling rate of 250 Hz, using the RNS-320 model configured to operate within the broadest available bandwidth, spanning 1–90 Hz.

## iEEG data analyses

iEEG data were synchronized with motion capture and eye tracking using network time protocol-generated markers on all devices[63]. All signals were aligned accordingly and resampled to a frequency of 250 Hz. Time–frequency analyses were conducted through the utilization of continuous wavelet transform and analytic Morse wavelets, characterized by a symmetry parameter of three and time-bandwidth product set to 60. For frequency bin specification, we employed ten voices per octave within the range of 1–90 Hz. Time-varying amplitudes were computed for each frequency. Given the slightly varying durations of real and imagined walking, we employed dynamic time warping to align real walking trials to each other and identify the turns on each walk. This information was then used for piecewise linear time-warping motion tracking and time-varying frequency amplitudes, aligning them with the mean timing of all trials. Subsequently, we averaged all trials for left and right walks. The wavelet power was graphically represented in Fig. 2b and utilized to determine frequency spectra in the Extended Data Fig. 5c through temporal average. Throughout other analyses, we normalized time–frequency amplitudes via $z$-scoring to account for disparate signal strengths across different recording channels. The theta-frequency range was identified for each individual by locating the spectral peak, as local maxima (±frequency bin width, Supplementary Table 1), in the 3–12 Hz range. The neighbouring lower and higher spectral minima determined the exact theta range centred around theta peaks. The individual theta ranges are listed in Supplementary Table 1.

Time-varying theta amplitudes were rendered on the motion trajectories to directly illustrate their relations to the positions in each walking route. We first illustrate theta dynamics during imagined navigation, assuming the same time-to-position mapping as in real-world navigation (Fig. 3c), before reconstructing positions from neural data. The imagined navigation and control trials were linearly warped based on the average trial duration within each condition. To evaluate whether theta dynamics exhibited temporal structure across trials, we computed the temporal consistency by correlating the mean signals from randomly splitting trials within each condition. The average values of 1,000 random splits for each channel were then statistically compared between conditions (Fig. 3d and Extended Data Figs. 6d,e and 10b,d). We correlated the temporal consistency values between real-world and imagined walks to investigate whether temporally structured theta dynamics appear at anatomically similar sites in both conditions (Fig. 3e). The same rendering of time-varying amplitudes and temporal consistency analyses was applied analogously to activities in the delta (1–3 Hz) and beta (13–30 Hz) bands. These frequency bands were selected on the basis of the spectra shown in Extended Data Fig. 5c.

The individual subject data were aligned in the same manner as single trials, using dynamic time warping to match movement trajectories, followed by piecewise linear time warping for the time–frequency amplitudes. To present theta dynamics at the group level, the average time and movement trajectories were employed. To analyse the similarity of theta dynamics within each segment of a walking route, we abstracted the navigational task structure into a sinusoidal pattern that peaked at each turn. This approach ensured that each segment adhered to the same sinusoidal pattern, scaled according to its normalized length or relative time. The phase of this sinusoidal pattern corresponded to the relative position within each segment. For both imagined navigation and control walks, we first normalized segment lengths to the mean length of the real-world walking segments. This normalization process ensured that all segments in our analysis were of equal duration, regardless of their original physical length. We then computed the cross-correlation between theta dynamics (reconstruction output) and the sinusoidal model of each walking segment. The central peak of the resulting cross-correlogram represents the correlation between theta dynamics and the sinusoidal task structure, effectively capturing the spatial layout and timing across each walking route. Moreover, any side peaks in the cross-correlogram indicate similarities between theta dynamics and shifted task structures (note increases at multiples of a segment), further confirming the consistency of theta activity patterns relative to the route's structure.

We detected transient theta oscillations lasting at least two cycles and exceeding the 95% CI of frequency-specific power using the eBOSC toolbox[64,65]. From these detections, we computed the prevalence (percentage), duration and bout amplitude for each condition and frequency bin (Extended Data Fig. 5a,c) of times when a given oscillation was detected. Bout amplitudes were computed as the magnitude of oscillations that exceeded the detection threshold. Furthermore, we quantified the percentage of trials in which transient theta oscillations were detected for every time point along a navigational route, resulting in the theta rate (Extended Data Fig. 5b).

Following the observation that oscillatory bouts in the theta-frequency range tend to precede turns, we investigated the development of this effect during the experiment's learning phase. Since participants learned quickly, we had at least five left and five right trials available for each subject. We computed the prevalence of oscillatory bouts during the two seconds leading up to a turn for each MTL electrode, based on the timing observed in the main experiment. Then, we compared these prevalence values during the learning phase to those from the first five left and right trails of the main experiment, where participants had already learned to walk the routes precisely as instructed. To quantify learning, we calculated the distance as the root mean squared error between the ideal and actual walking route.

To explore the potential influence of eye movements on MTL recordings, we computed average time–frequency activities aligned with the onset of eye saccades and compared these plots with those aligned with turns (Extended Data Fig. 8b–e). Four saccades (Extended Data Fig. 8c) were randomly selected per trial to correspond with the four turns (shown in Extended Data Fig. 8b). In addition, we ensured that each fixation preceding a saccade lasted longer than 600 ms, as described in previous work[66]. This approach allowed us to match the number of observations across conditions precisely, as depicted in Extended Data Fig. 8b–e.

The temporal relationship between behavioural variables and theta dynamics was quantified by calculating cross-correlations between each channel's theta dynamics and the behavioural variables (Extended Data Fig. 7a,c), averaged across trials. All behavioural variables were recorded as continuous signals and were time warped, similar to the time-varying amplitudes in the theta-frequency range, to allow for comparison of their time courses with the neural data.

## Reconstructing imagined positions

By aligning all segments, we were able to calculate the average theta modulation pattern contingent on the relative positions within each segment across all recording channels. Theta dynamics at different electrodes exhibited peaks at marginally distinct relative positions (Fig. 4f). A modulation akin to a cosine wave would align precisely with the task structure, while a modulation similar to a −sine wave would be orthogonal to it, as depicted in Fig. 4f. The synergy of these two orthogonal components makes them apt for encoding relative position as a circular variable. Leveraging Euler's formula, the relative position can be mathematically derived as the angle of the resultant vector formed by these two orthogonal components.

We established a straightforward linear regression model to examine the feasibility of reconstructing the relative position from theta dynamics. This model encodes the relative segment position as a circular variable, generated through the weighted summation of the theta dynamics of all 18 channels within the MTL. The task structure was used as a response variable to estimate the cosine and −sine modulated components. To prevent overfitting, we used cross-validation (ten folds repeated ten times) to determine the regression weights. Subsequently, we tested whether the positions estimated from unseen trials aligned with the actual motion capture data. The outcome of this assessment was the computation of estimation errors, which represented the differences between actual and estimated positions.

During real-world navigation trials, we were able to use motion capture to account for minor variations in timing and movement trajectories between trials. However, for the imagined navigation trials, this information was naturally absent. To address this, we instructed participants to mentally navigate the routes in the same manner they did during real-world walking trials. The treadmill played a pivotal role in these trials and conditions, preserving gait patterns without actual progression through space. Rather than relying on motion capture, we recorded the durations between the start and completion of imaginations, which were defined by the onset of the treadmill and the button press reported by participants. During movement, position and time are linearly related through velocity. However, during mental navigation, the imagined velocity of the participant is not directly observable. The imagination trials were aligned via linear time warping for each participant and subsequently averaged across trials. To ensure consistency across the group, participants' data were adjusted to match the grand average imagination time, yielding a uniform duration. Given the potential variance in imagined velocity profiles across participants and conditions, our aim was to synchronize theta dynamics during imaginations with the navigational route structure of the imagined routes (as depicted in Fig. 4a). Although the exact timing of imagination remained unknown, we capitalized on the structural framework of each imagined route, characterized by five segments connected through four turns. In our main analyses, we used identical route structures for all participants. To address the specific route followed by P2, we conducted a supplemental analysis that adapted the model to include an additional turn (Supplementary Fig. 1).

We utilized the position estimation model developed from our real-world walking data to derive an initial estimate of imagined positions. Subsequently, we applied dynamic time warping to align these estimated relative positions with the task structure, allowing for a maximal time shift of ±2 s. These time offsets represent delays between corresponding estimated imagined and actual positions (Supplementary Table 3). The positive offsets indicate that neural data associated with imagined turns occurs after real turns, whereas negative offsets suggest it occurs beforehand. This alignment procedure was conducted for each participant while preserving the relative timing between recording channels within each participant. To prevent overfitting, we employed tenfold cross-validation repeated ten times to learn the temporal relationship from subpartitions of the data. Each participant's single trials were randomly grouped into ten subsets, each containing roughly 10% of the unique trials. From these subsets, nine (~90% of the trials) were used to train the alignment of time and position during imagined navigation. The trained alignment was then tested on the remaining subset of previously unseen imagination trials, serving as a test of the alignment algorithm's generalizability. This process was repeated for all ten combinations of training and test sets. In addition, the random grouping of subsets was repeated, and the training and testing procedure was applied across ten repetitions, resulting in a total of 100 iterations. Each iteration produced position estimates computed from the test set trials, independent of the subset used for training the alignment. The outcomes of these generalizations are presented throughout the manuscript. It is important to note that this alignment

procedure is only valid if the theta dynamics from different imagination trials exhibit similarities. The same procedures for position estimation and time alignment were applied to the sole treadmill walking data, serving as control analyses. By ensuring that results from all conditions were temporally aligned, we were able to calculate position reconstruction errors within each condition using the actual positions recorded during real-world navigation trials (Fig. 4h). Since relative position was modelled as a circular variable, we present reconstruction errors as polar histograms, showing angles between actual and estimated positions.

## Scalp EEG

This study primarily focuses on iEEG to investigate neural dynamics during real-world and imagined navigation. While scalp EEG data were collected concurrently, the primary analyses presented centre on iEEG findings. Future analyses will explore the effects observed in scalp EEG and examine their relationship to the iEEG results discussed in this study, providing a broader understanding of neural activity across recording modalities. For completeness, we briefly describe the scalp EEG data acquisition and analysis methods related to the preliminary analysis presented in Supplementary Fig. 2.

In brief, scalp EEG was recorded using the eego sports system (ANT Neuro), which includes a mobile amplifier, recording tablet and Waveguard EEG cap. The data were sampled at 1,024 Hz across 64 channels. Due to the participants' free movement, the scalp EEG data were contaminated with motion and muscular artifacts, and the telemetry required for RNS device access introduced additional noise. To mitigate stereotypical technical artifacts, we applied existing EEG correction methods, including the Zapline-plus[67] approach, and rejected epochs with high-frequency muscular activity from further analyses. We analysed two parieto-occipital channels (PO3 and PO4), selected for their typically pronounced scalp rhythms and reduced neck muscle artifacts compared with more posterior channels during walking.

## Electrode localization

Postoperative computed tomography scans were utilized to locate the leads and contacts of the electrodes. To achieve precise anatomical localization for each recording contact, co-registration was performed by aligning the computed tomography scans with preoperative magnetic resonance images (T1- and/or T2-weighted scans), following a previously established procedure[68]. Recording sites within the MTL included the hippocampus, amygdala, parahippocampal cortex and perirhinal cortex (Supplementary Table 1). Recording contacts situated outside the MTL were excluded from further analyses.

## Detection of epileptic events

Epileptic events, including interictal epileptiform discharges (IED), were identified using established methods[69] and tailored for RNS data[27,28]. In brief, two distinct thresholds were computed, and the samples that surpassed either threshold were designated as IED periods and subsequently excluded from further analyses. For this analysis, we computed the envelope of broadband iEEG (1–90 Hz) and filtered (15–80 Hz) signals. The thresholds for IED identification were determined as five times the median values of these signals. To capture both the up- and down-ramping epileptic activities before and after IED periods, we extended the identified periods by 256 ms (64 samples) in both directions. In cases where an IED period encompassed over 50% of a trial's duration, the entire trial was omitted from the analysis. In total, $3.8 \pm 3.0\%$ (median ± standard deviation) of the data were identified as IED periods. This relatively low percentage is attributed to our deliberate selection of participants with a limited number of epileptic events per day.

## Eye tracking

Eye movements were tracked using the Pupil Labs Invisible glasses (Pupil Labs GmbH). The eye data were captured at a frequency of 200 Hz with a resolution of 192 × 192 pixels and synchronized with

a scene camera, which sampled the surroundings at 30 Hz, offering a resolution of 1,088 × 1,080 pixels. To merge these two data streams into coherent gaze position data, we used the Pupil Cloud software[70]. Saccadic eye movements were identified using the Cluster-Fix toolbox[71], which identifies periods of gaze fixation using $k$-means clustering, and transitions between these fixations are classified as saccades. To assess gaze densities, we computed two-dimensional (2D) histograms, subsequently smoothed using a Gaussian kernel featuring a standard deviation of 1 pixel. These gaze densities were normalized relative to the expected value of a uniform distribution. The average gaze point in each participant's visual view was identified to align and centre individual gaze densities across participants and conditions. Differences in gaze densities were computed by subtracting gaze densities during left and right real-world and imagined walks (Extended Data Fig. 8a).

### Statistical comparisons

To quantify the temporal consistency of theta dynamics, we employed a methodology involving the correlation of mean signals derived from randomly dividing the data into two halves. Specifically, trial numbers within each condition were randomly grouped into two data sets. We computed the average signals for each dataset and then calculated their linearly correlation using the Pearson correlation coefficient. This procedure was repeated 1,000 times to yield a robust measure of how theta dynamics from one randomly selected half of the trials in a condition were associated with the other half. The experimental conditions were compared using multilevel block permutation tests[72,73] while restricting exchangeability blocks to each participant (Fig. 3d and Extended Data Figs. 6d,e and 10b,d). We used cluster bootstrap to provide CIs for these non-parametric comparisons. The temporal consistency values for each channel were then subjected to correlation to explore the presence of temporally structured theta dynamics at anatomically similar locations across different conditions. The statistical significance of this correlation was evaluated using linear mixed-effect models using the participant numbers as a grouping variable (Fig. 3e). These tests were applied to account for the grouped nature of our recordings consisting of multiple electrodes grouped per participant.

Non-parametric permutation tests were employed to determine the significance of theta amplitudes dependent on positions along the routes. Multiple comparisons were corrected using cluster-level statistics[72]. First, the mean theta dynamics across channels within each condition were computed. Second, we obtained a permutation distribution by sign-flipping randomly selected halves of the channels and averaging the results, repeated 1,000 times. Third, we applied a primary threshold of $P = 0.05$ to identify clusters of significant activity. Fourth, these cluster sizes were compared with those from the permutation distribution and were considered significant only if they survived a secondary threshold of $P = 0.05$ (cluster level), which corrected for multiple comparisons (Extended Data Fig. 6b).

We utilized non-parametric permutation tests to compare the position reconstruction errors against the control condition, which was sole treadmill walking. Specifically, we randomly reassigned the labels of reconstruction errors for the real-world and control conditions and averaged them 10,000 times, resulting in a permutation distribution. The actual reconstruction errors were then compared to this permutation distribution. The same procedure was analogously applied to the imagined navigation data. The probabilities of reconstructed positions (as depicted in Fig. 4g) were normalized relative to the expected value of a uniform distribution. This scale indicates by which factor the position reconstruction results exceed this chance distribution. We also compared the reconstruction errors of each condition with a null distribution generated by randomly shifting the reconstructed and modelled positions 10,000 times. This approach maintained the temporal correlation of neighbouring samples while simulating random temporal relations between conditions. By doing so, we could

more accurately assess the significance of our reconstruction errors and ensure a rigorous comparison.

We tested for a significant correlation between theta dynamics during real-world and imagined navigation trials, generalizing the alignment procedure. This correlation was compared with values computed from randomly shifting the real-world and imagined navigation data 10,000 times to disrupt their temporal relationship (Extended Data Fig. 6c). The same analyses were performed to assess the correlation of theta dynamics with route structures (Fig. 4e).

Moreover, we computed linear regression models of theta dynamics and behavioural variables (Extended Data Fig. 7b,d) on a single-trial level for each recording electrode. We then tested the effects at the group level and compared the resulting regression coefficients using multilevel block permutation tests while controlling for multiple comparisons[72,73].

### Reporting summary

Further information on research design is available in the Nature Portfolio Reporting Summary linked to this article.

## Data availability

The data supporting the findings of this study are openly available via Zenodo at https://doi.org/10.5281/zenodo.13743052 (ref. 74). Source data are provided with this paper.

## Code availability

The custom code to reproduce the findings of this study is openly available via Zenodo at https://doi.org/10.5281/zenodo.13743052 (ref. 74).

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

## Acknowledgements

This work was supported by the National Institutes of Health, the National Institute of Neurological Disorders and Stroke, under award numbers U01NS117838 to N.S. and K99NS126715 to M. Stangl, by the McKnight Foundation (Technological Innovations Award in Neuroscience to N.S.) and a Keck Junior Faculty Award (to N.S.). We thank all participants for taking part in the study, all members of the Suthana laboratory for discussions, D. Batista for support with the Unity application and M. Jenkens-Drake for helping with illustrations.

## Author contributions

Conceptualization: M. Seeber, M. Stangl and N.S. Methodology: M. Seeber., M.V.M., U.T. and N.S. Software: M. Seeber, M.V.M. and U.T. Analysis: M. Seeber. Investigation: M. Seeber, M. Stangl, M.V.M., U.T., S.H., C.H.H., J.-P.L., V.R.R., I.F., D.E. and N.S. Resources: C.H.H., J.-P.L., V.R.R., I.F., D.E. and N.S. Data curation: M. Seeber, M. Stangl, M.V.M., U.T. and S.H. Writing—original draft preparation: M. Seeber and N.S. Writing—review and editing: M. Seeber, M. Stangl, M.V.M., U.T., S.H., C.H.H., J.-P.L., V.R.R., I.F., D.E. and N.S. Visualization: M. Seeber. Supervision: N.S. Project administration: S.H., C.H.H., J.-P.L., V.R.R., I.F., D.E. and N.S. Funding acquisition: N.S.

## Competing interests

V.R.R. is on the Medical Advisory Board for NeuroPace, Inc. The other authors declare no competing interests.

## Additional information

**Extended data** is available for this paper at https://doi.org/10.1038/s41562-025-02119-3.

**Correspondence and requests for materials** should be addressed to Martin Seeber or Nanthia Suthana.

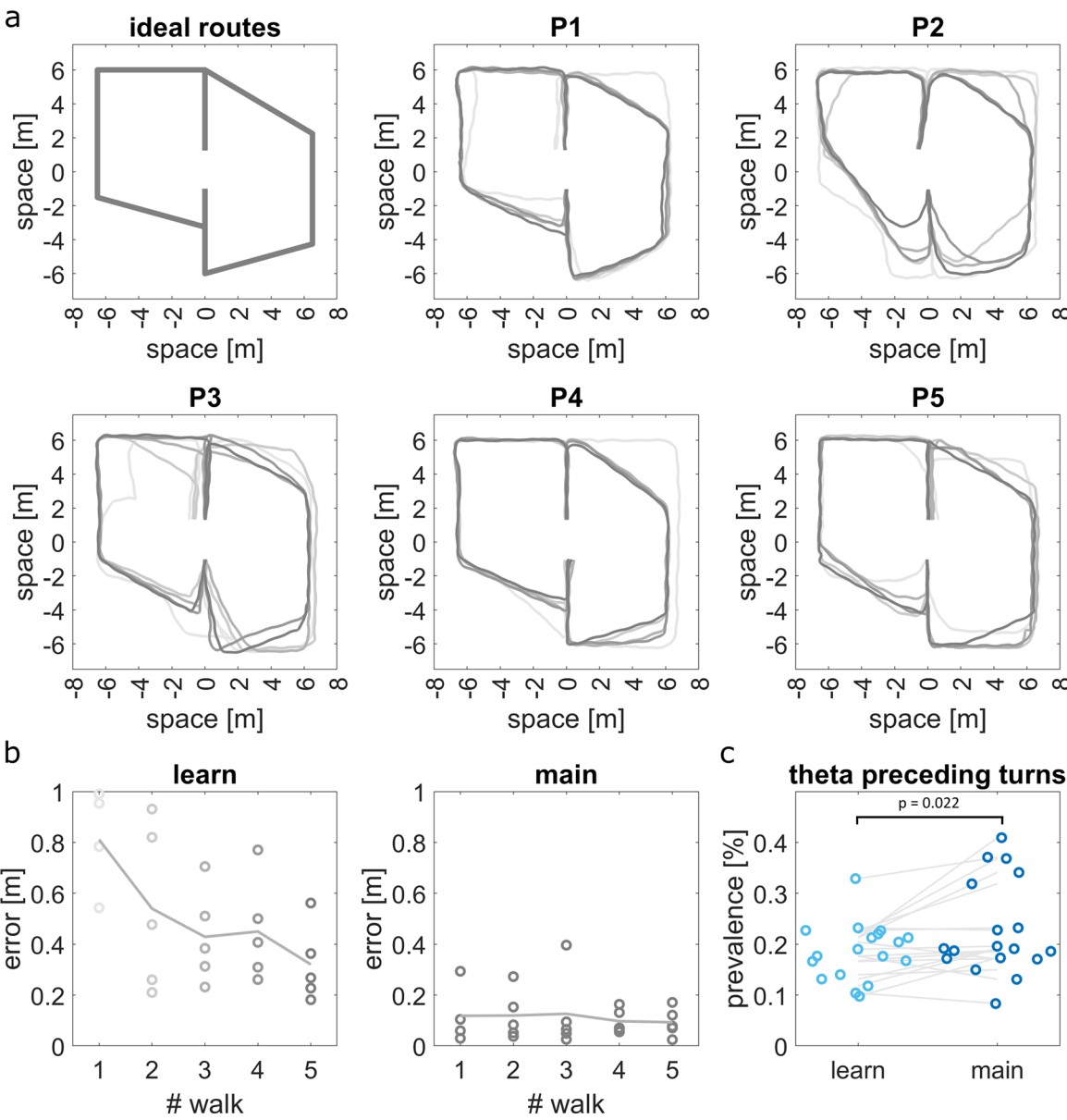

**Extended Data Fig. 1 | Learning phase. a**, The instructed, ideal route is depicted alongside the actual walks performed by all participants (P1-P5), with earlier and later walks illustrated in light and dark gray colors. **b**, As participants learned to navigate the instructed, ideal routes, they gradually reduced the error between actual and ideal routes within the first five left and right walks. Dots represent each participant's mean error rates (root mean square) depending on the number of walks performed, with earlier and later walks illustrated in light and dark gray, respectively. The behavioral errors during the first five left and right walks of the main experiment are shown for comparison. **c**, A comparison of bouts in the theta frequency range preceding turns showed their prevalence to be significantly higher during the main experiment (one-sided permutation test, $r_d = 0.04$, 95% CI = [0.01, 0.12], p = 0.021, d = 0.68) than during the learning phase. Dots represent the theta bout prevalence at each recording electrode during both the learning phase and the main experiment. Grey lines indicate the comparison between these two experimental phases for each electrode. The prevalence of the first five left and right walks in the learning phase and main experiment were compared, respectively.

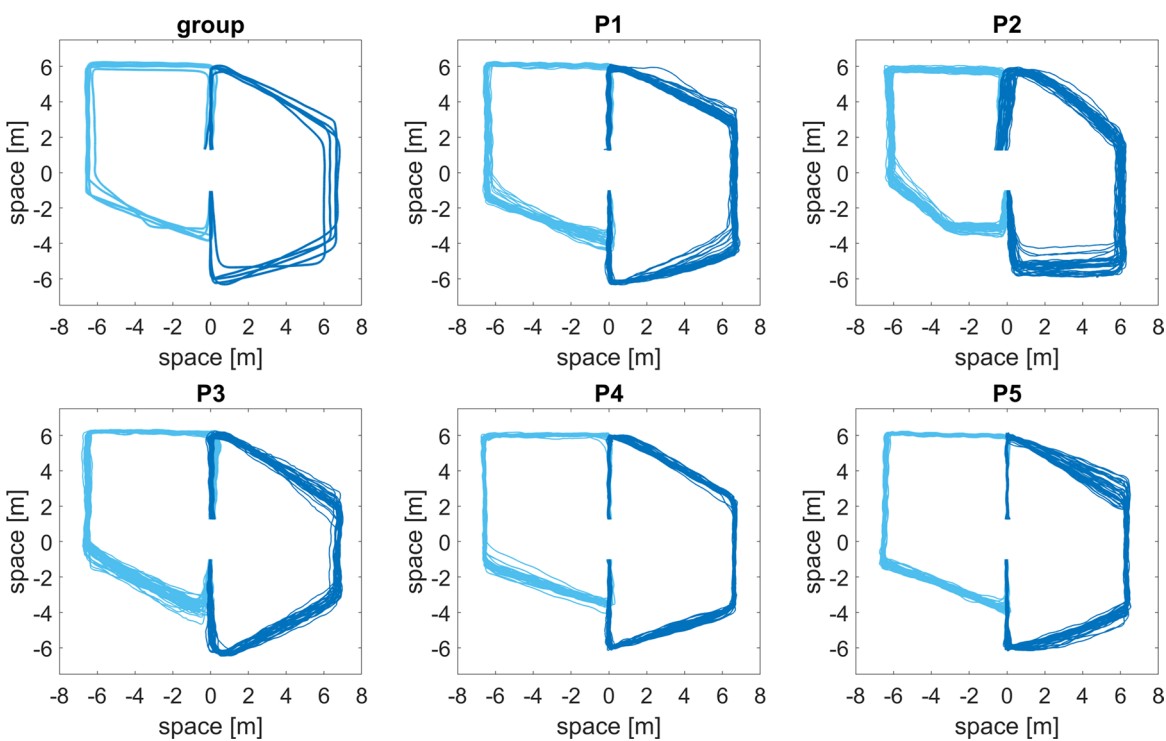

**Extended Data Fig. 2 | Motion capture data.** Average walking routes (N = 5 participants) of each participant shown in a group plot (top left). Walking trajectories of each trial for every participant (P1-P5) shown in light/dark blue for the left/right route, respectively.

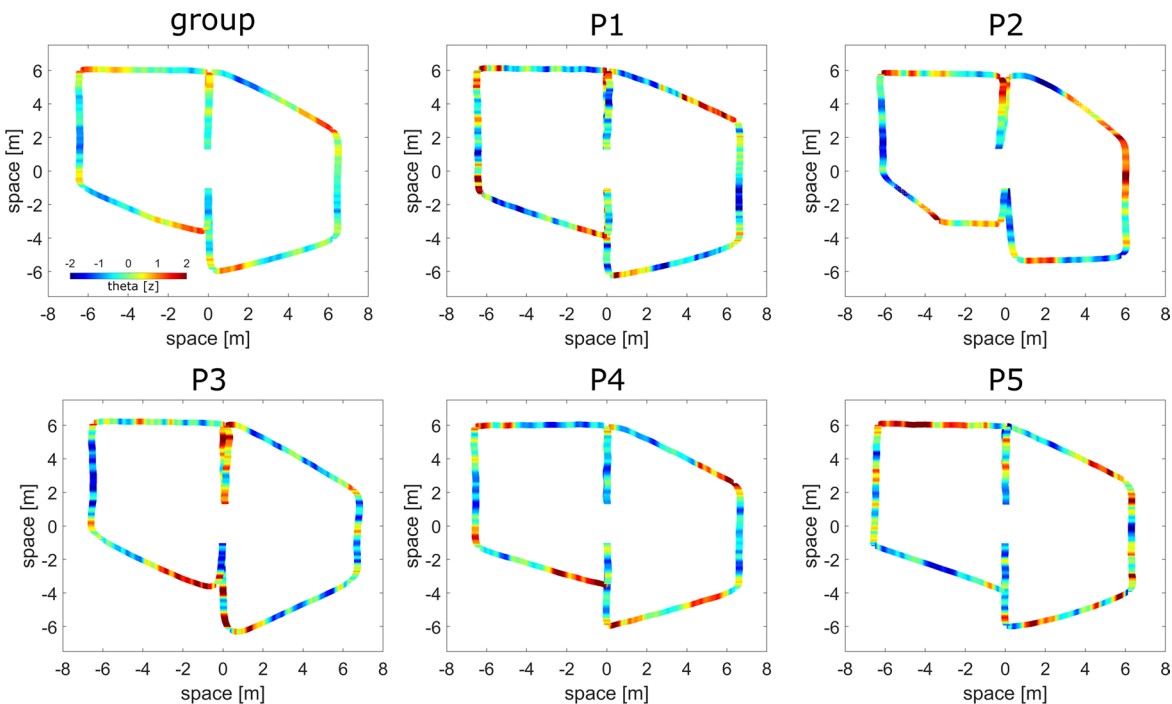

**Extended Data Fig. 3 | Theta activity rendered on motion trajectories.** Mean group and individual participant (P1-5) theta activity patterns within the left anterior hippocampus (top left panel) superimposed onto the mean behavioral motion trajectories, illustrating an increase in theta power as each of the participants approaches upcoming turns.

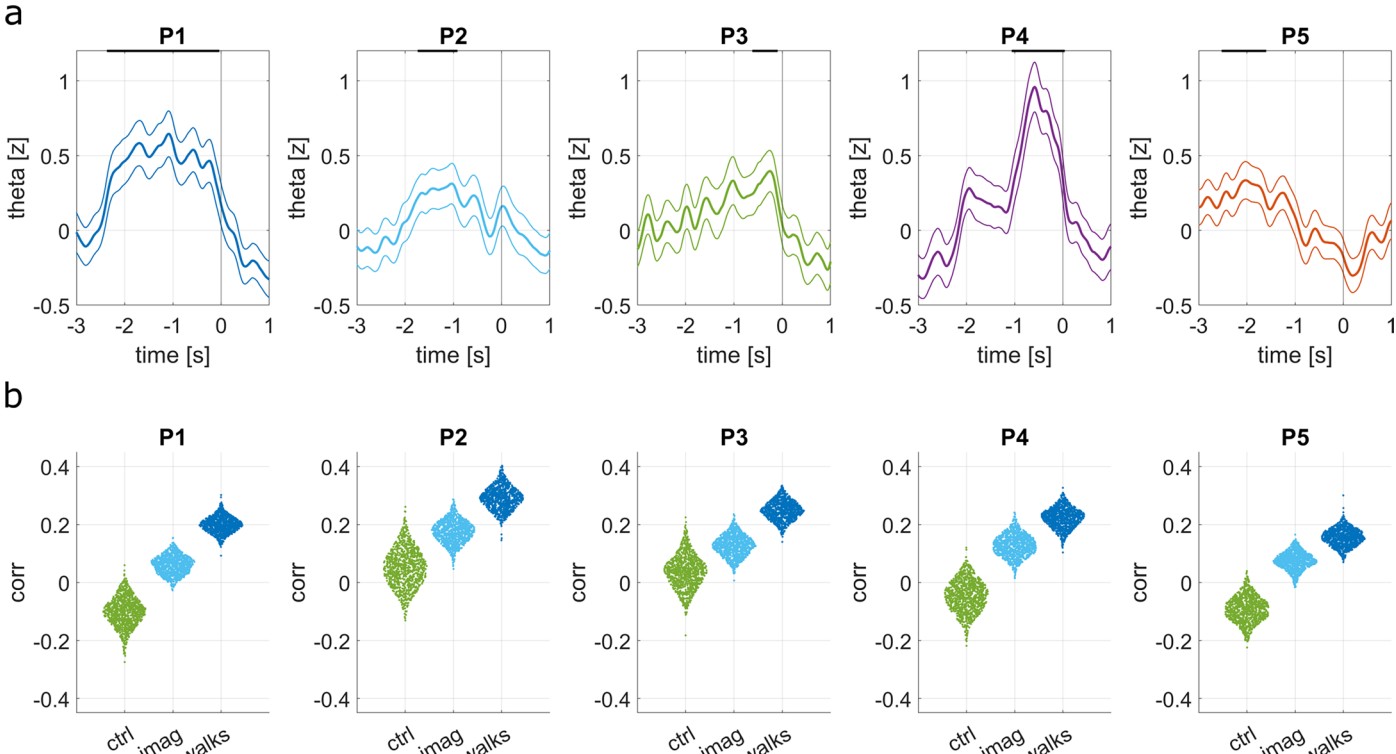

**Extended Data Fig. 4 | Individual subject analyses. a**, Mean theta activity (± SEM) across channels for each participant (P1-P5) aligned to turns (t = 0). Black bars on top of the plots indicate significant periods. **b**, Temporal consistency assessed for each participant. Individual circles represent data from correlations computed by randomly half-split data a thousand times. Differences between conditions are significant (one-sided permutation test, p < 0.001, FWE corrected) in every subject.

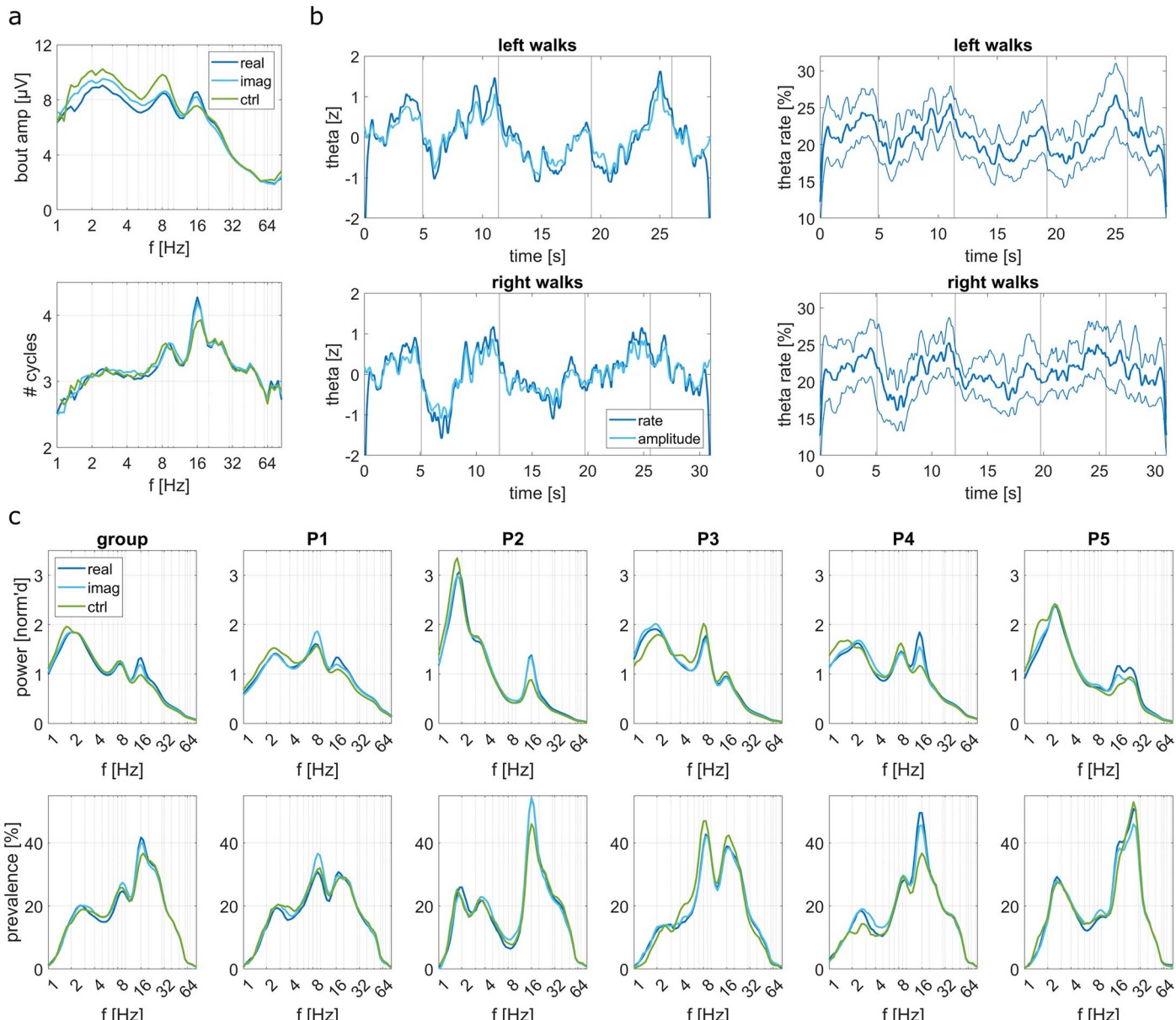

**Extended Data Fig. 5 | Frequency spectra and theta dynamics. a,** Frequency spectra of oscillatory bout amplitudes (top) and the average number of cycles (bottom) are depicted as averages across participants for real-world navigation (real), imagined navigation (imag), and the control condition (ctrl). **b,** Theta amplitude dynamics and time-resolved theta rates, z-scored for comparison (left), along with theta bout rates ( ± SEM) shown as detection percentages across trials (right) for left (top) and right (bottom) walks. Vertical lines represent turns. **c,** Normalized power spectra (top) and the prevalence of oscillatory bouts (bottom) are shown as averages across participants (group) and individually for each participant (P1-P5).

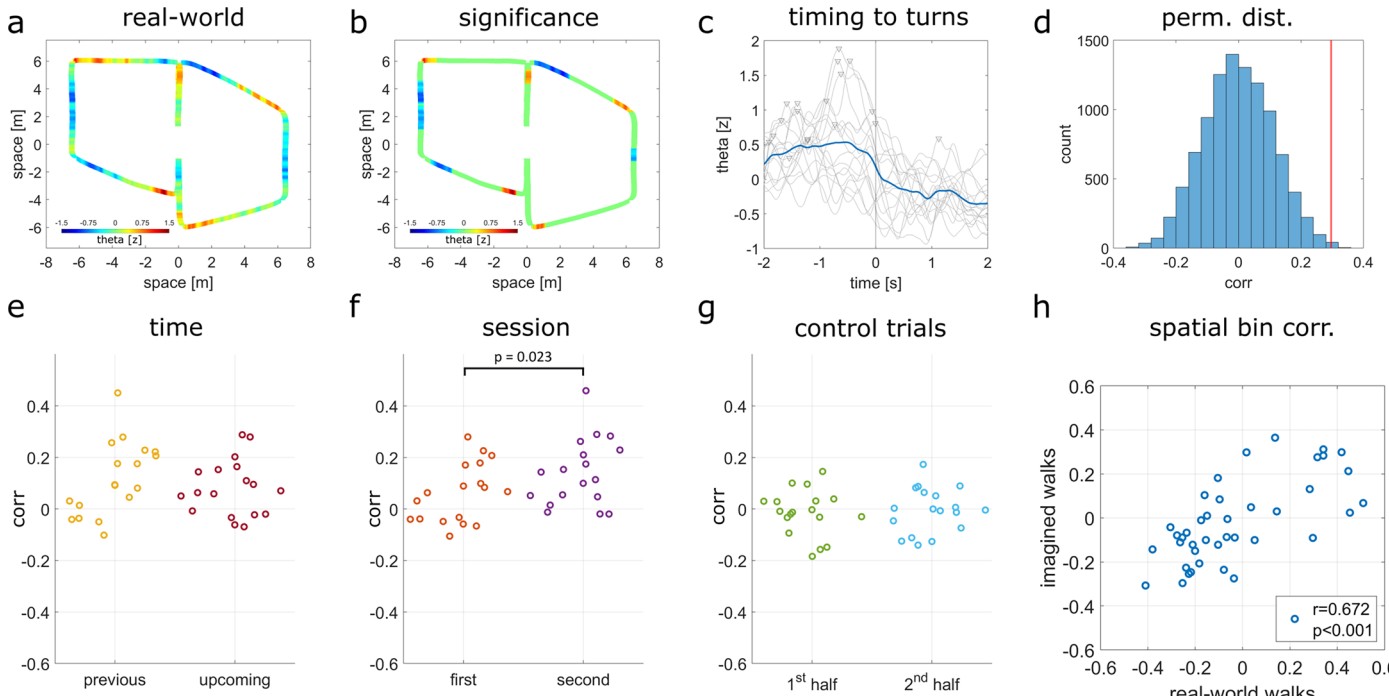

**Extended Data Fig. 6 | Theta activity, timing, and temporal consistency across conditions. a**, Superimposed averaged theta activity across all channels on the mean motion trajectories. **b**, Statistically significant theta clusters (p < 0.05, determined through a two-sided cluster-based permutation test). Non-significant periods are shown in green, while significant activities are depicted in non-green colors. **c**, Average theta activity aligned with turns (t = 0) is depicted in blue, while individual electrode data is presented in grey. Triangles indicate that peak theta activity precedes turns in 17 out of 18 channels, with 1 triangle indicating a peak occurring after the turn. **d**, Permutation distribution and actual correlation coefficient (corr) between theta dynamics of real-world and imagined navigation (red line). **e**, Temporal consistency, measured using a correlation coefficient (corr), during imagination of previous (past) and upcoming (future)

routes. **f**, Temporal consistency was higher within the second imagination session compared to the first (one-sided permutation test, $r_d = 0.18$, 95% CI = [0.12, 0.24], p = 0.023, d = 1.51). **g**, Temporal consistency was computed separately for the first and second halves of the treadmill control condition, with no evidence of a time-on-task effect observed in this control condition. **h**, Correlation between theta activity during real-world and imagined navigation (one-sided permutation test, r = 0.67, 95% CI = [0.50, 0.81], p < 0.001), binned by relative reconstructed position (each dot represents a 5% bin of the total route segment length). The x and y axes represent normalized theta activity pooled across channels after position reconstruction, using both route representation components (-sin, cos) to generate the reconstructed positions. This is calculated for each of the 20 spatial bins, resulting in 40 data points (2 × 20).

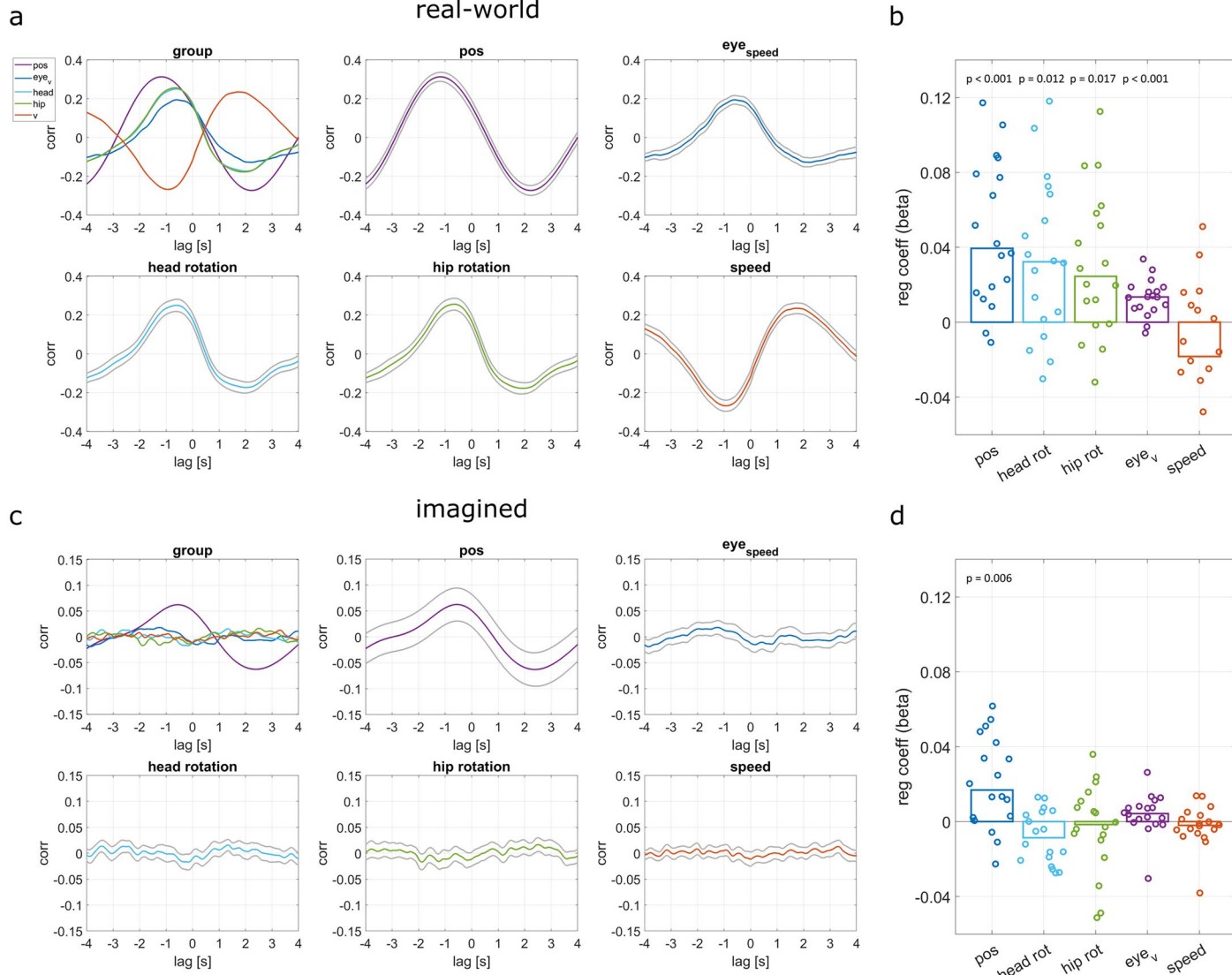

**Extended Data Fig. 7 | Relation of theta dynamics to behavioral variables.**
**a**, Cross-correlation of theta dynamics and behavioral variables during real-world navigation. Colored lines illustrate the group average, while grey lines indicate standard errors across the folds of cross-validation for each behavioral variable (position [pos], eye speed, head rotation, hip rotation, and movement speed [v]). Negative lags represent instances where theta dynamics preceded the behavioral signals. The top left panel summarizes the temporal relationships, revealing that theta activities peaked approximately 1 second before physical turns, followed by head rotation, eye saccades, speed reductions and hip rotation. The positive lagged correlation between theta and movement speed reflects the speed decrease before turns, which is then followed by a reduction in theta activity. **b**, Regression coefficients of ongoing (single-trial) theta dynamics during real-world navigation (two-sided permutation tests, FWE corrected), with labeled behavioral variables. **c**, Cross-correlation of theta dynamics and behavioral variables during imagined navigation, analogous to **(a)**. Theta dynamics are compared to behavioral variables after alignment to enable comparisons with position within the route segments. **d**, Regression coefficients for ongoing (single-trial) theta dynamics during imagined navigation (two-sided permutation tests, FWE corrected), similar to **(b)**.

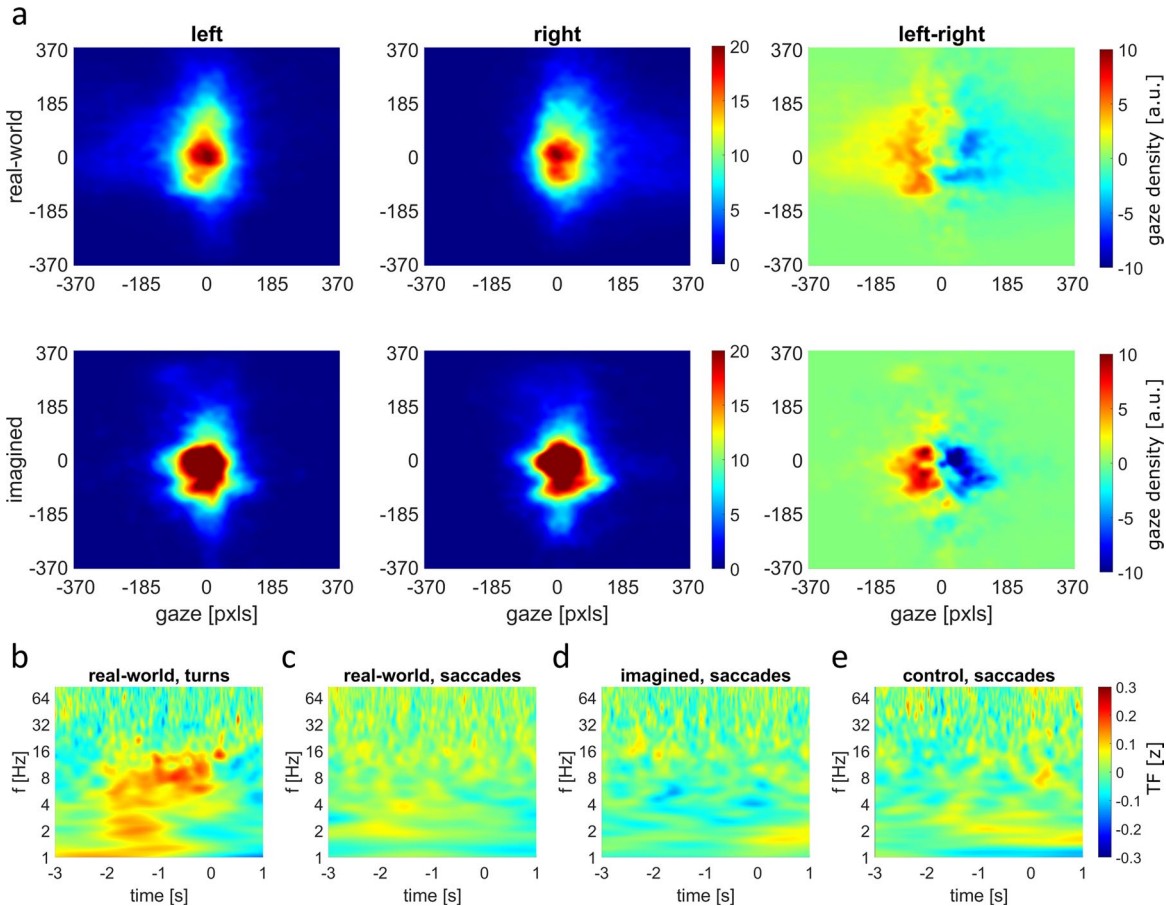

**Extended Data Fig. 8 | Gaze densities and MTL activity aligned to turns and saccades. a**, Shown are the 2D histograms, centered at participants' natural fixation point within their field of view. During real-world navigation on the left or right route, participants tended to direct their gaze more towards the respective left or right directions. Similarly, during imagined navigation, participants exhibited a slight bias in their gaze towards the left or right visual field, depending on whether they were simulating a left or right route. This pattern of gaze behavior confirmed their active engagement in the task. **b-c**, Time-frequency (TF) activity aligned to turns **(b)** and saccades **(c)** during real-world navigation. **d-e**, Time-frequency activity aligned to eye movements (saccades) during imagined navigation **(d)** and the control condition of treadmill walking without imagination **(e)**. Note pronounced theta activities precedes physical turns but not saccades.

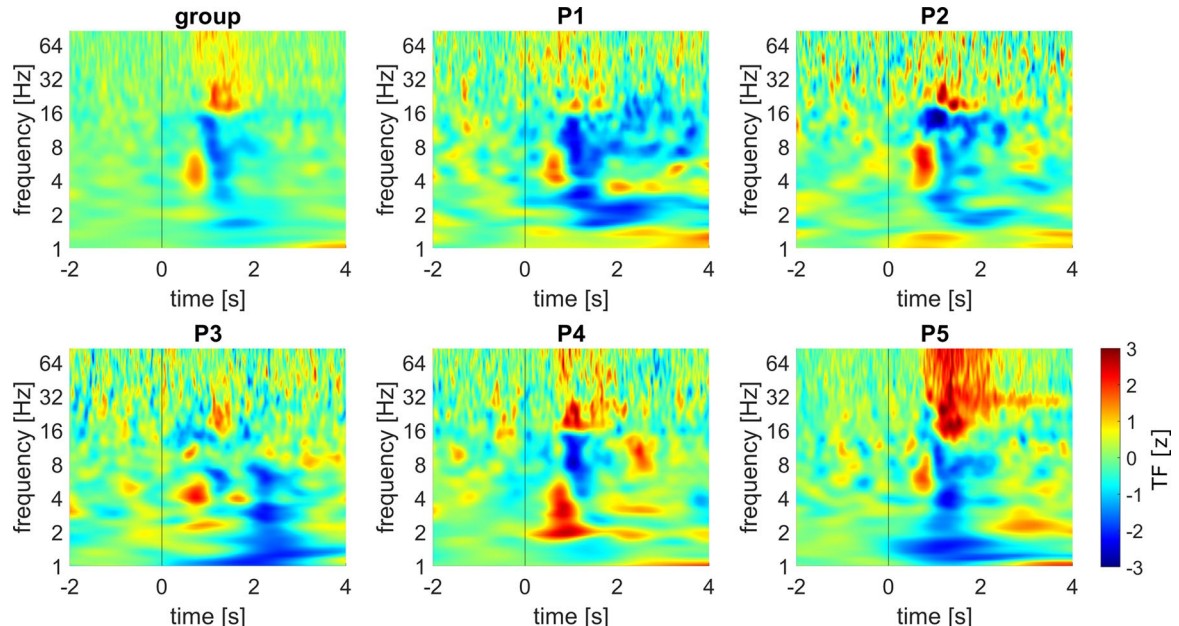

**Extended Data Fig. 9 | Self-location task.** Time-frequency (TF) activity of the self-location task is shown for the group (top left panel) and each participant (average across channels). TF plots are aligned to auditory cues (t = 0, black lines), which signal participants to report their location relative to learned positions.

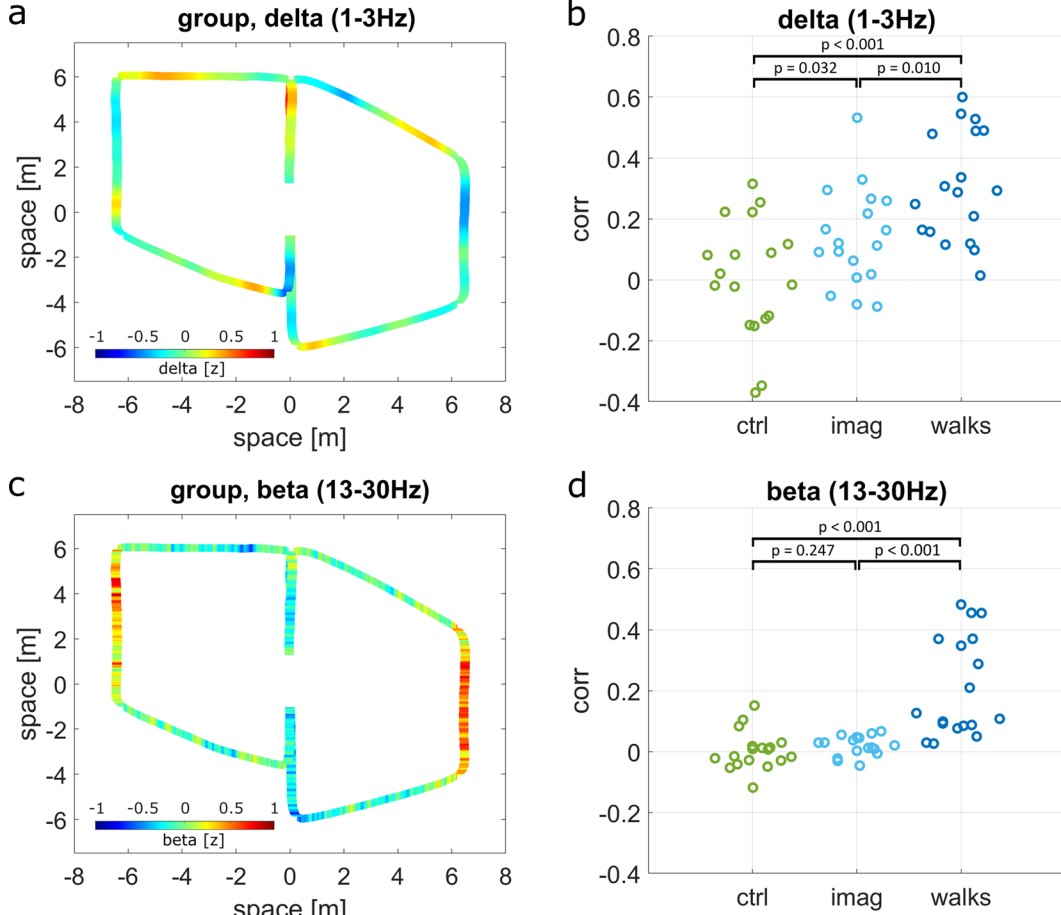

**Extended Data Fig. 10 | Delta and beta activities. a**, Amplitude dynamics in the delta frequency range are overlaid onto motion trajectories as an average across participants. **b**, Temporal consistency of delta dynamics was significantly higher during both imagined and real-world navigation (one-sided permutation test, FWE corrected) than for sole treadmill walking (control condition). **c**, Amplitude dynamics in the beta frequency range as a function of position on motion trajectories are shown as average across participants. Note the increased activities during the third (downward) segment of the routes, where participants estimated and reported their location relative to hidden objects. **d**, Temporal consistency of beta dynamics was significantly increased during real-world navigation (one-sided permutation test, FWE corrected) compared to both treadmill walking conditions.

# Reporting Summary

## Statistics

For all statistical analyses, confirm that the following items are present in the figure legend, table legend, main text, or Methods section.

| n/a | Confirmed | |
|---|---|---|
| ☐ | ☒ | The exact sample size (*n*) for each experimental group/condition, given as a discrete number and unit of measurement |
| ☐ | ☒ | A statement on whether measurements were taken from distinct samples or whether the same sample was measured repeatedly |
| ☐ | ☒ | The statistical test(s) used AND whether they are one- or two-sided<br>*Only common tests should be described solely by name; describe more complex techniques in the Methods section.* |
| ☐ | ☒ | A description of all covariates tested |
| ☐ | ☒ | A description of any assumptions or corrections, such as tests of normality and adjustment for multiple comparisons |
| ☐ | ☒ | A full description of the statistical parameters including central tendency (e.g. means) or other basic estimates (e.g. regression coefficient) AND variation (e.g. standard deviation) or associated estimates of uncertainty (e.g. confidence intervals) |
| ☐ | ☒ | For null hypothesis testing, the test statistic (e.g. *F*, *t*, *r*) with confidence intervals, effect sizes, degrees of freedom and *P* value noted<br>*Give P values as exact values whenever suitable.* |
| ☒ | ☐ | For Bayesian analysis, information on the choice of priors and Markov chain Monte Carlo settings |
| ☒ | ☐ | For hierarchical and complex designs, identification of the appropriate level for tests and full reporting of outcomes |
| ☐ | ☒ | Estimates of effect sizes (e.g. Cohen's *d*, Pearson's *r*), indicating how they were calculated |

*Our web collection on statistics for biologists contains articles on many of the points above.*

## Software and code

Policy information about availability of computer code

| Data collection | Neuropace RNS System; Unity game engine (version 2021.2.19f1); OptiTrack MOTIVE (version 3.0) |
|---|---|
| Data analysis | MATLAB R2021b (The MathWorks, Natick, MA, USA); Signal Processing Toolbox for MATLAB R2021b; Deep Learning Toolbox for MATLAB R2021b; Statistics and Machine Learning Toolbox for MATLAB R2021b; DSP System Toolbox for MATLAB R2021b; Financial Toolbox for MATLAB R2021b; Wavelet Toolbox for MATLAB R2021b; BOSC toolbox for MATLAB (eBOSC) |

For manuscripts utilizing custom algorithms or software that are central to the research but not yet described in published literature, software must be made available to editors and reviewers. We strongly encourage code deposition in a community repository (e.g. GitHub). See the Nature Portfolio guidelines for submitting code & software for further information.

## Data

Policy information about availability of data

All manuscripts must include a data availability statement. This statement should provide the following information, where applicable:
- Accession codes, unique identifiers, or web links for publicly available datasets
- A description of any restrictions on data availability
- For clinical datasets or third party data, please ensure that the statement adheres to our policy

The data supporting the findings of this study are openly available via Zenodo at https://doi.org/10.5281/zenodo.13743052.

# Research involving human participants, their data, or biological material

Policy information about studies with human participants or human data. See also policy information about sex, gender (identity/presentation), and sexual orientation and race, ethnicity and racism.

| | |
|---|---|
| Reporting on sex and gender | Three male and two female participants took part in this study. Recruitment of participants from both male and female sex was part of the study design. Sex- and gender-based analyses were not performed, given the small cohort size after subdividing by sex and gender. Additionally, sex- and gender-based analyses were not pursued, given that there was no a priori hypothesis of a difference in spatial navigation or episodic memory neurophysiology across sex or gender. As such, results apply to both male and female sex. |
| Reporting on race, ethnicity, or other socially relevant groupings | Recruitment of participants aimed at representing the demographics in the greater Los Angeles area. In our study and analyses, we do not discriminate participants by race, ethnicity, or other socially relevant groupings. |
| Population characteristics | Five participants (24-40 years old; three males, two females) who had been chronically implanted with the FDA-approved RNS System (NeuroPace, Inc.) for the treatment of pharmaco-resistant focal epilepsy volunteered for this study. The electrode placements were determined exclusively by clinical treatment criteria. More details are provided in Extended Data Table 1. |
| Recruitment | Participants were recruited via phone or e-Mail from a database of the University of California Los Angeles, University of California San Francisco, and Staford University. |
| Ethics oversight | All participants volunteered for the study by providing informed consent according to a protocol approved by the UCLA Medical Institutional Review Board (IRB). |

Note that full information on the approval of the study protocol must also be provided in the manuscript.

# Field-specific reporting

Please select the one below that is the best fit for your research. If you are not sure, read the appropriate sections before making your selection.

☒ Life sciences  ☐ Behavioural & social sciences  ☐ Ecological, evolutionary & environmental sciences

For a reference copy of the document with all sections, see nature.com/documents/nr-reporting-summary-flat.pdf

# Life sciences study design

All studies must disclose on these points even when the disclosure is negative.

| | |
|---|---|
| Sample size | Five participants (24-40 years old; three males, two females) with pharmaco-resistant focal epilepsy volunteered for this study. This sample size was chosen, and the experimental procedure was performed individually for each participant to enable data analyses not only on the group level (across all recording channels from all participants) but also independently for each participant in order to investigate the consistency of effects and their reliability across different participants. Further, the sample size selected in this study is comparable to prior studies reporting similar effects in intracranial EEG recordings in freely moving humans. |
| Data exclusions | The data from all participants was used for data analyses. Each participant had four recording channels. Across all participants, a total of 18 channels were located in MTL regions, including the hippocampus, perirhinal cortex, parahippocampal cortex, and subiculum. Recording channels outside of the MTL were excluded from the main analyses. |
| Replication | The experimental procedure was repeated five times independently with five different participants. All methods used to perform this study and analyses needed to replicate the presented findings are detailed in the Methods section of the manuscript. In the manuscript, we provide several analyses that show that the effect is strongly present and strikingly consistent across each individual participant that we have tested, suggesting that our results can be generalized beyond the tested sample and reproduced with new and different datasets. |
| Randomization | All participants were tested with the same experimental protocol with no separate experimental groups. The task comprised real-world and imagined navigation on two distinct spatial routes. Whether participants imagined navigating on their previous or upcoming route in the second or third recording block was counterbalanced. |
| Blinding | We have tested a rare group of participants with pharmacoresistant epilepsy who have been previously implanted with the NeuroPace RNSSystem for the treatment of their epilepsy. As such, all experiments were aware of this and not blinded with regard to the participants'condition. |

# Reporting for specific materials, systems and methods

We require information from authors about some types of materials, experimental systems and methods used in many studies. Here, indicate whether each material, system or method listed is relevant to your study. If you are not sure if a list item applies to your research, read the appropriate section before selecting a response.

## Materials & experimental systems

| n/a | Involved in the study |
|---|---|
| ☒ | Antibodies |
| ☒ | Eukaryotic cell lines |
| ☒ | Palaeontology and archaeology |
| ☒ | Animals and other organisms |
| ☒ | Clinical data |
| ☒ | Dual use research of concern |
| ☒ | Plants |

## Methods

| n/a | Involved in the study |
|---|---|
| ☒ | ChIP-seq |
| ☒ | Flow cytometry |
| ☐ | ☒ MRI-based neuroimaging |

# Plants

| | |
|---|---|
| Seed stocks | *Report on the source of all seed stocks or other plant material used. If applicable, state the seed stock centre and catalogue number. If plant specimens were collected from the field, describe the collection location, date and sampling procedures.* |
| Novel plant genotypes | *Describe the methods by which all novel plant genotypes were produced. This includes those generated by transgenic approaches, gene editing, chemical/radiation-based mutagenesis and hybridization. For transgenic lines, describe the transformation method, the number of independent lines analyzed and the generation upon which experiments were performed. For gene-edited lines, describe the editor used, the endogenous sequence targeted for editing, the targeting guide RNA sequence (if applicable) and how the editor was applied.* |
| Authentication | *Describe any authentication procedures for each seed stock used or novel genotype generated. Describe any experiments used to assess the effect of a mutation and, where applicable, how potential secondary effects (e.g. second site T-DNA insertions, mosiacism, off-target gene editing) were examined.* |

# Magnetic resonance imaging

## Experimental design

| | |
|---|---|
| Design type | MRI was used only to determine the localization of electrode contacts within the brain. |
| Design specifications | MRI was used only for electrode contact localization; thus, participants did not perform an experimental task during MRI scanning. |
| Behavioral performance measures | No behavioral performance measures were acquired or derived, since participants did not perform an experimental task during MRI scanning. |

## Acquisition

| | |
|---|---|
| Imaging type(s) | structural |
| Field strength | 3 Tesla |
| Sequence & imaging parameters | standard T1- and T2-weighted sequences |
| Area of acquisition | whole-brain |
| Diffusion MRI | ☐ Used    ☒ Not used |

## Preprocessing

| | |
|---|---|
| Preprocessing software | MRI data were preprocessed using FSL (FMRIB Software Library, Oxford University, UK; v5.0.11) for image registration with the FLIRT function (default parameters), and ITK-SNAP (version 3.8.0) for visualization and manual segmentation of electrode contacts. |
| Normalization | MRI images were not normalized. |
| Normalization template | MRI images were not normalized. |
| Noise and artifact removal | No noise or artifact removal procedures were applied. |
| Volume censoring | Volume censoring was not applied. |

## Statistical modeling & inference

| | |
|---|---|
| Model type and settings | No model-based analyses were performed using MRI data. |
| Effect(s) tested | MRI was used only for electrode contact localization; thus, no task- or stimulus-related analyses were performed using MRI data. |

Specify type of analysis: ☒ Whole brain ☐ ROI-based ☐ Both

| | |
|---|---|
| Statistic type for inference<br>(See Eklund et al. 2016) | No statistical analyses were performed using MRI data. |
| Correction | No statistical analyses were performed using MRI data; thus, no correction methods were applied. |

## Models & analysis

| n/a | Involved in the study |
|---|---|
| ☒ ☐ | Functional and/or effective connectivity |
| ☒ ☐ | Graph analysis |
| ☒ ☐ | Multivariate modeling or predictive analysis |

