## [Peer Review File · Nature Human Behaviour]

Human neural dynamics of real-world and imagined navigation

Corresponding Author: Dr Nanthia Suthana

Version 0:

Decision Letter:

10th July 2024

Dear Dr Suthana,

Thank you once again for your manuscript, entitled "Human neural dynamics of real-world and imagined navigation," and for your patience during the peer review process.

Your manuscript has now been evaluated by 3 reviewers, whose comments are included at the end of this letter. Although the reviewers find your work to be of interest, they also raise some important concerns. We are very interested in the possibility of publishing your study in Nature Human Behaviour, but would like to consider your response to these concerns in the form of a revised manuscript before we make a decision on publication.

We ask that you address all reviewers' concerns and provide additional information in support of your analytical approach.

In sum, we invite you to revise your manuscript taking into account all reviewer and editor comments. We are committed to providing a fair and constructive peer-review process. Do not hesitate to contact us if there are specific requests from the reviewers that you believe are technically impossible or unlikely to yield a meaningful outcome.

We hope to receive your revised manuscript within two months. I would be grateful if you could contact us as soon as possible if you foresee difficulties with meeting this target resubmission date.

- Include a "Response to the editors and reviewers" document detailing, point-by-point, how you addressed each editor and referee comment. If no action was taken to address a point, you must provide a compelling argument. When formatting this document, please respond to each reviewer comment individually, including the full text of the reviewer comment verbatim followed by your response to the individual point. This response will be used by the editors to evaluate your revision and sent back to the reviewers along with the revised manuscript.
- Highlight all changes made to your manuscript or provide us with a version that tracks changes.

Link Redacted

We look forward to seeing the revised manuscript and thank you for the opportunity to review your work. Please do not hesitate to contact me if you have any questions or would like to discuss these revisions further.

Sincerely,

Giacomo Ariani, PhD
Senior Editor
Nature Human Behaviour

Reviewer expertise:

Reviewer #1: Hippocampal function, navigation, memory, intracranial EEG

Reviewer #2: Hippocampal function, navigation, memory, intracranial EEG

Reviewer #3: Hippocampal function, navigation, memory, intracranial EEG

REVIEWER COMMENTS:

Reviewer #1:

Remarks to the Author:

The manuscript by Seeber et al. describes an interesting study with epilepsy patients implanted with a stimulation and recording device for seizure management, which was used to explore hippocampal LFP activities during real and imagined navigation. Increased spectral power and bouts of oscillations in a broad theta 4-12 Hz frequency range was reported to precede critical moments of turns between segments of walking in a straight path. The same pattern was reconstructed from epochs of imaginary navigation but not from control epochs of walking a treadmill without engaging imagination. Elegant and simple design combined with clear electrophysiological effects between the three conditions make the results of this study compelling in light of the adequately referenced previous work. Implications of the reported results go beyond navigation functions, providing an elegant mechanism for segmenting episodic memory or even organizing behavior at the stage of abstract thinking/mentation. Nature Human Behavior presents a suitable venue for the submitted study.

I have the following comments for the authors to address:

1. Classifying the LFP activities

The authors associate the reported effects with 'hippocampal theta oscillations' based on the previous rodent and human navigation studies. Figure 2D shows that the highest amplitude of the spectral activities observed on the group level before the turns is in the classic alpha frequency range (9-14 Hz). There are other spectral activities engaged in this critical moment in the delta and even low beta frequency ranges. Power spectra and the bout prevalence plots in the Extended Data Fig. 2a summarize the group results, showing at least three distinct peaks in this low frequency range of the LFP spectrum at: 2-4 Hz (delta?), 6-10 Hz (theta?), 10-14 Hz (alpha?). It is interesting that the greatest differences between the real, imagined and control conditions are present in the third (alpha?) peak in this plot, which is not really discussed by the authors. Neither are the distinct peaks in the power spectra and their different roles in navigation. The alpha rhythm is historically known to be involved especially with visual perception - I wonder if closing of the eyes could be a factor in the differences observed for this third spectral peak... I would recommend a more cautious reporting of these intricate differences between distinct spectral activities. For example, describing a particular spectral activity in a specific frequency range, e.g. 'increased power in the theta frequency range' is more appropriate than: 'increased power of theta oscillations'. The former doesn't make any assumption about a particular class of oscillations, which is especially useful when other overlapping frequency peaks are present in the same range. A similar predicament has just been reviewed in case of the spectral activities in the high gamma/ripple frequency range (Kucewicz et al. 2024 Brain). I suggest revising the phrasing used throughout the manuscript and discussing the other peaks, especially the nearby peak in the alpha frequency range. An alternative would be to use 'low' and 'high theta' oscillations (Goyal et al. 2020 Nature Comm) but these referred to more delta (2-3 Hz peak) and actual theta (8 Hz peak), respectively, more than the alpha peak observed here.

2. Bouts of oscillations

Observation of the bouts or transients of oscillations peaking at the same time as the increases in the spectral power is intriguing and suggests that the increased spectral power is driven by increased incidence/rate of these bouts. This would imply that the transients are a fundamental unit/event underlying the spectral power increases. Investigating these a little more in terms of their duration, amplitude and peak frequency could support associating the reported changes in the spectral power with actual oscillations and not other non-oscillatory sources of power (e.g. filtering of interictal epileptiform spikes or other artifacts). The authors would then be able to make more grounded claims about theta or alpha 'oscillations' rather than just spectral power.

It is interesting that similar oscillations in the higher frequency ranges have also been reported as these bouts, transients or bursts. Examples include beta/gamma bursts in monkeys (Lundqvist 2016 Neuron) or bursts of gamma, ripple and fast ripple oscillations in humans (see the Brain review above) associated with memory and cognitive functions. The principle appears to be the same across the frequency spectrum - it's the duration and amplitude of the transients that is proportionally increasing going down the spectrum. Thus, high frequency oscillations (HFOs) in the ripple and fast ripple ranges and the shortest in duration and lowest in amplitude on one side of the spectrum, whereas the gamma, beta, alpha and theta transients are progressively longer and stronger in amplitude according to the power law.

I recommend describing the reported bouts further in the manuscript.

3. Saccades during imagined navigation

It is not clear to me from reading the manuscript text how the saccades were detected during the imagined navigation. It is our natural tendency to close the eyelids during imagination even if one is instructed not to do so. It would be very useful for this study to report the proportion of time when the eye-tracking signal was not detected in the three conditions. My prediction is that the imagined condition would have the highest proportion. We typically see in our studies that this proportion is much higher during the recall phase of a memory task than during encoding, even though both use a fixation cross in the middle of the screen with subjects instructed to focus gaze. It turns out that when we imagine and recall memories then our gaze tends to shift to a specific corner of our field of view. In this case the three conditions cannot be compared as presented in the Extended Data Fig. 8, unless the proportion of time fixating on the center of the field is comparable.

Reviewer #2:

Remarks to the Author:

This is a highly interesting study that provides insight into unique intracranial data in freely moving humans. In particular, the here recorded hippocampus/MTL electrophysiology allows unparalleled comparisons to previous influential animal studies on navigation and memory as well as previous studies on virtual navigation in humans.

This generally enthusiastic assessment is, however, limited by the amount of detail that is missing in the current presentation of the data, which, at this stage, prevents an in-depth understanding of the reported results. The most glaring of such missing details are regarding how time alignment was done at various stages of the analysis, especially with the imagined navigation part of the experiment; this is critical to the main conclusions of the paper and the amount of detail is very limited and often unclear.

Major:

1. On several occasions throughout the manuscript, labels are missing, in particular for colorbars; sometimes, the colorbar itself is missing. Since it is not possible to fully understand figures without units, the authors should carefully revise their manuscript and add the missing information.

Some examples: Figs. 4a, ext. data Fig. 4: colorbar missing
Figs. 2c,d, 4c, ext. data Fig. 7: colorbar units missing.

2. Imagined navigation

The entire analysis with imagined navigation trials is very difficult to understand. Could the authors be more explicit about how the time alignment was done? More specific concerns follow:

- For the correlation coefficients in Figure 3(d), were these derived after the time alignment procedure laid out in lines 495-498, which involves the real-world navigation data? If so, doesn't this already lead to a spurious intratrial correlation?

Assuming that a different method was used than this, when the authors "reconstructed" positions, we are wondering how this was done here for their imagined condition. There will be an offset between starting to walk on the treadmill and imagination of the current location on the path. How did the authors locate turns in imagined walks?

- As above, a description of Fig. 3c in the results section is missing. How was this output created? In Figure 3,c, the author's comment that "Note that theta activity tended to increase earlier during imagined turns compared to real-world turns." appears unsustainable to us given that the data from real-world navigation was used to determine the imagined position in the first place and so any inferences about timing is a circular argument. In fact, it is puzzling that there is such a consistent pattern given that the time alignment procedure would lead to theta bursts coming at very similar pre-turn intervals as in the real-world navigation.

- Is it possible that the act of imagining a route causes theta bouts but not necessarily in the same task-structure dependent way as in real-world navigation? For example, if the act of imagining spatial navigation itself leads to an enhanced theta activity, one would end up simply aligning the peaks of the theta activity via the time alignment procedure and cause peaks to appear at "imagined turns". In this context, it would be useful to know how much time stretching and shrinking was typically needed for the alignment.

- Related to the above, when comparing their navigation/imagination data to the control trials, it seems that these comparisons are always confounded by time on task. In particular, when comparing the temporal consistency across conditions this could be an issue. Can the authors exclude the possibility that the correlation values were only low in the control condition because it was always the first part of the experiment?

- The possibility of a more informative metric to show that the imagined navigation has a similar task-related structure should be explored. For example, would it be possible to use motion/eye tracking to infer when the person may be executing an imagined turn on the trajectory?

3. Results related to Fig. 4 and spatial decoding.

- How exactly was the spatial decoding done? The authors refer to Agarwal et al., 2015. However, that method was dependent on high-density spatial sampling of LFPs, which was not the case here (max. 2 bipolar pairs per hemisphere). What was changed and

how and why would it still work?

- The authors then applied the position reconstruction model to imagined navigation data and aligned estimated imagined positions and the task structure using linear time warping. How exactly was the linear warping done? Please provide more details in both the methods and results section. How would offsets and breaks during imagination not be problematic? Was the alignment using linear warping also done for real-world and control walks? How does warped alignment in these conditions compare to unwarped alignment?
- Was the model depicted in Fig. 4a also used with P2's data? Since P2 used another route, can the authors adjust the model and see if outcomes are improved?
- Fig. 4c: is "#electrode" arranged in an order that corresponds to individual participants? e.g. do electrodes 15 to 18 belong to a single patient? If yes, does this explain the variation across electrodes depicted in Fig. 4c?
- How does the lag, provided as the number of segments in Fig. 4b, match the length of one segment on a walking route?
- Fig 4c shows that theta is high at before a turn and that activity lasts until shortly after the turn. Why does one need sine and cosine in Figure 4c? Would there be a simpler way to model this?

4. Theta Bouts

- How do the theta bouts in this study compare to previous studies in terms of peak frequency and duration? E.g., Aghajan reports theta bouts with average durations of 400 ms, which seems rather short as compared to what is shown in Fig. 3b.
- Please also clarify the following issue:
l. 124: "We analyzed the timing of these bouts on a single-trial level and observed their alignment with upcoming turns (Fig. 3b)." However, there is no mention of bouts in the caption of Fig. 3b or the figure itself. It seems that theta power was used instead in Fig. 3b? It seems that the figure corresponding to the bout alignment described in line 124 is missing.

5. Power spectra and theta dynamics

- Is the focus on the frequency (3-12 Hz) range justified when considering human data? Looking at Ext. Data Fig. 2a, it seems the authors are excluding the dominant peaks at both ends of the power spectrum, 2 Hz and 16 Hz. Are we maybe missing out on the most interesting signals in this experiment?

While there is an ongoing debate about how to define frequency bands and whether it is useful to do so, a direct translation of frequency band definitions from rodents to humans seems to be inappropriate, independent of these issues. The authors should provide their motivation and also make sure they are not discarding frequencies that are clearly modulated by the task based on arbitrary definitions.

- How do the here-reported power spectra compare to previous reports? For example, Quasim et al., 2021, Cell, find no peaks in the power spectra of human LFP recordings during virtual navigation (Fig. S1).
- Do individual participants show a clear theta peak in the power spectrum? Please provide individual power spectra per condition alongside the average power spectra in Ext. Data Fig. 2. Please provide the theta peaks for each participant in addition to the theta ranges in Table 1.
- The theta peaks seem to settle around 8 Hz in the average spectrum, which could be corresponding to the peak of the dominant EEG alpha rhythm. Did the patients wear surface EEG? If yes, it would be highly informative if the authors provide some basic parameters on the surface EEG data during the navigation conditions: What was each participant's alpha peak and how does it compare to the iEEG theta peaks?
- The power spectrum shows a peak at 16 Hz that is rather unusual, at least for recordings in non-moving patients. It seems that there is a clear modulation of (real and imagined) navigation compared to the sole treadmill walking in both the power spectrum and prevalence of bouts. What is the authors' take on this clear peak?

- Ext. Data Fig. 6a: there seem to be regular fluctuations around 2 Hz in theta signal before turns, very clearly in P3 but also P1, P2, P5 and maybe even P4. Are these fluctuations reflected in the low-frequency peak of the power spectrum in Ext. Data Fig. 2?

Individual power spectra will allow more insight into this signal. What is the authors' take on these slow rhythms? Are they related to the task?

6. Learning phase of the experiment

- The learning phase of the experiment is hard to understand from the description. We would recommend having a supplementary figure that illustrates the learning phase, specifically showing where and how the visual and auditory cues were presented. Further, the authors need to spell out clearly why such a complex learning strategy was chosen, rather than something more straightforward like having visual markers along the path during the learning phase.

- "Participants were then instructed to determine their relative position to the nearest object by pressing the appropriate left/middle/right button on a wireless handheld mouse, corresponding to the first/second/third object on their route."

It is unclear what "determining their relative position to the nearest object" means here. Were the participants required to simply indicate which of the three objects they were closest to? Or were asked to estimate the distance to the nearest object? And finally, what was the role of this task during learning?

- What were the exact criteria used to conclude that the route has been learnt by a participant? How was this criteria met for P2 who, as the authors point out, clearly deviated from the route and incorporated an extra turn?

- Was data from the RNS system recorded during the learning phase? If so, this would be a very interesting dataset to understand better what may be driving the bouts of theta oscillation before turns during the main experiment. We would suggest an analysis along the following lines:

-Do similar bouts of theta occur before every turn right from the start of the learning phase?

If so, do these bouts get stronger with learning?

If not, when do they first show up?

-Is there a relation between how well the route is learned and the occurrence of these bursts?

The above analysis would allow us to understand if these bouts of theta track common computations required to spatially orient oneself as the body turns or if they in some way act as "landmarks" to routes that are memorized.

7. Correlates with eye, head movement:

- More information is needed on the control conditions the authors are comparing their navigation condition against. Please provide details for the analyses of theta activity concerning eye movements, head rotation, etc.: How was theta activity analyzed in relation to the movements? E.g., locked to onset, offset? How were those detected? Please provide detailed information.

I. 174: "However, we did not find a direct effect of eye movements on theta amplitudes during imaginations ($p = 0.58$). Instead, we observed that theta dynamics were primarily related to the relative position within the route segments ($p = 0.006$), mirroring the patterns observed during real-world navigation."

Since crucial information on the movement analyses are missing, I do not see how the authors' conclusion that "eye movements did not drive the effect" is justified.

- Previous studies reported saccade-triggered phase theta resets in human and monkey hippocampus. Can the authors replicate those findings? The way the results are described at the moment reads as if saccades generally had no impact on theta activity.

8. Discussion

- It seems that parts of the relevant literature that investigated virtual navigation using human intracranial data are overlooked, in particular those that talk about theta oscillations. To name a few, in no particular order: Quasim et al., 2021, Cell; Chen et al., 2018, Curr Biol; Kunz et al., 2019, Sci Adv; Maidenbaum et al., 2018, PNAS; Liu et al., 2023, Curr Biol;

- It would be great to read a more extensive discussion on how the current novel findings in real-world navigation relate to the previous virtual navigation studies.

Minor:

- I. 758: "Triangles indicate that peak theta activity precedes turns in 17 out of 18 channels." Did the one channel not show a peak or was it dropped out for other reasons? Please add information to the caption.

- Please indicate the participant number of the participant shown in movie 1.

- Please indicate whether participants contributed to previous studies and provide information about these studies for each individual participant.

- Were the 5 patients included in the study the only five from which the data was collected for this experiment?

- Line 158: Please provide the correlation coefficients for each condition.
- l. 169: "As a means to validate participants' engagement in imagining the distinct navigational routes, we examined their eye movements." We don't agree with the end of this statement, since the authors depict gaze in ext. data fig. 7, not (eye) movements. Please adjust.

Reviewer #3:

Remarks to the Author:

In their article "Human neural dynamics of real-world and imagined navigation," Seeber et al. use intracranial EEG to investigate neural activity in the theta-frequency range in freely moving humans ($n = 5$). The key findings are that theta oscillations occur in short bouts during real-world navigation; that theta power increases before turns; that theta power is temporally stable across trials; and that theta power is similar between actual navigation and remembered ("imagined") navigation. This is a very interesting and timely manuscript that sheds new light on the question of what kind of information neural activity in the theta-frequency range may represent. My specific comments are listed below.

Major comments

It is often difficult to distinguish between imagination and memory retrieval at a conceptual level and in my view this difficulty also applies to this study. As the patients are not asked to imagine new routes that they have never experienced before, I find it more appropriate to interpret the results in the context of memory retrieval instead of imagination. I would thus suggest that the authors tone down their statements about imagination, including the title and the abstract.

Similarly, I think that the interpretation of the results in the light of episodic memory (for example, on page 2) is somewhat exaggerated as participants were asked to navigate along predefined routes and to remember these routes. Episodic memories are often defined as unique memories for what happened where and when, and I thus do not think that the authors' task fulfills the requirements of an episodic memory task. I would suggest toning down the statements on episodic memory and being more specific to the findings at hand.

The authors state that their results show that theta dynamics encoded spatial geometry. It is not clear to me how different theta-power levels on different parts of the track allow for the conclusion that theta dynamics encode the spatial geometry of those routes. Please clarify.

The authors mention a computational model that generalizes from real-world to imagined navigation and that successfully reconstructs participants' imagined positions, but I could not find such a computational model in the manuscript (I wouldn't consider a statistical model for an analysis a computational model). Please clarify.

As the sample size is small (5 patients; 18 electrodes) it would be helpful to see theta power as a function of linearized position on the two routes for all electrodes, averaged across trials, separately for the different conditions.

The temporal consistency analyses (page 6) are useful regarding the question whether the observed neural activity is stable across trials. However, in my view, they do not allow for conclusions that comparable functional networks are involved in real-world and remembered navigation. They could, for example, be a simple side effect of the prevalence of theta oscillations in the channel-wise data (amongst many other basic properties of the channel-wise data). Please clarify or reword.

While walking on the treadmill, participants could view parts of the real-world navigation routes. Furthermore, the authors show that participants' eye movements during treadmill walking resemble their eye movements during real-world navigation. Could it thus be that the participants are viewing their remembered navigation routes while walking on the treadmill? If so, viewing the routes could drive the temporal consistency of remembered navigation routes, and their similarity to real-world navigation. Two analyses to see whether this was indeed the case would be (1) to quantify how much participants looked at the remembered navigation routes while walking on the treadmill; (2) to test whether temporal stability and similarity between real-world and remembered navigation is driven by the first half of the route that participants were able to view during treadmill walking.

The section on reconstructing imagined positions seems innovative and interesting overall, but I also find it hard to follow. For example, the authors state that it was evident that theta amplitudes were consistently modulated across the segments. I believe that this shall mean that theta power was similar at corresponding relative positions on each segment. Could the authors provide some empirical evidence for this claim, for example by correlating theta power from different segments with each other?

Could the authors provide more detailed information on the 10 x 10 cross-validation procedure? Otherwise, it is difficult to evaluate the correctness of this approach.

The time warping procedure is essential for this analysis, but I could not find a clear description of this procedure in the manuscript. Could you provide more details? How do you account for the fact that the time warping may create false positives by warping the data in a way that resembles theta activity during real-world navigation but has nothing to do with the participants' remembered navigation routes? In other words, what is your statistical baseline to evaluate your results in Fig. 4D, middle, against? I think it's not sufficient to compare them against the control periods.

The authors state that "theta dynamics exhibited a significant correlation between real-world and imagined navigation," but they don't show any plots in support of this statement. To me this result seems to be the strongest evidence that there is shared neural activity between real-world navigation and remembered ("imagined") navigation. Could you thus please show scatter plots etc. to demonstrate the robustness of this result and whether it is mainly driven by particular segments and/or positions on the segments?

Fig. 4, B vs. C: How can these two plots be consistent with each other if B shows a match with a cosine structure of each path segment, and C suggests a sine structure for some of the electrodes?

Fig. 4B: An alternative to the sinusoidal modulation of theta power would be a boxcar-shaped modulation (low power in the middle of the segment; high power at its ends). What does the cross-correlation look like when using this alternative boxcar-shaped modulation? Does the sinusoidal modulation provide a significantly better fit to the data? In other words, to what extent is the apparent cosine modulation simply driven by increased theta power shortly before the turns?

Minor comments

Fig. 1B: It wasn't clear to why all control trials on the treadmill happened before the first real-world navigation walk. Why weren't these control trials interspersed with the other trial types (real-world navigation and remembered navigation)?

Fig. 2B: Is this single-trial data or averaged across trials? Please specify in the legend.

Fig. 2B: What is the consistent neural activity at around 16 Hz?

Fig. 2C: How many channels contribute to this plot? How does this pattern look like for electrodes in the left posterior hippocampus, right anterior hippocampus, and right posterior hippocampus?

Fig. 2D: Is this plot across all electrodes? Or also just the electrodes from the left anterior hippocampus? Are the power increases significant when applying a cluster-based permutation test?

Fig. 3C: Please clarify what you mean by "weighted sum of all channels."

Fig. 3C: As suggested above, it would be good to see a plot showing theta activity of all electrodes as a function of linearized position, given the small sample size.

Fig. 4C: Is this data smoothed across electrodes? If yes, please show the unsmoothed version as smoothing across electrodes (from different hemispheres and subjects) does not seem valid.

Fig. 4D, middle: The prediction of "imagined" positions seems to be driven by the edge cases. How much of this result can be explained by the time warping procedure that may simply align periods of increased theta power (that may have nothing to do with particular positions on a segment)?

Fig. S2B: It would be useful to add a measure of dispersion.

Fig. S2B and Fig. S3: This would be useful to see for remembered ("imagined") navigation trials, too.

Fig. S4B: What do the colors represent?

Fig. S6A: Please add a measure of dispersion.

Version 1:

Decision Letter:

24th October 2024

Dear Dr Suthana,

Thank you once again for your revised manuscript, entitled "Human neural dynamics of real-world and imagined navigation," and for your patience during the re-review process.

Your manuscript has now been evaluated by the same reviewers who evaluated your original manuscript. All reviewer feedback is included at the end of this letter. Although the reviewers found your manuscript to have improved during revision, Reviewer #3 maintains some important outstanding concerns. We remain very interested in the possibility of publishing your study in *Nature Human Behaviour*, but would like to consider your response to these outstanding concerns in the form of a revised manuscript before we make a decision on publication.

Finally, your revised manuscript must comply fully with our editorial policies and formatting requirements. Failure to do so will result in your manuscript being returned to you, which will delay its consideration. To assist you in this process, I have attached a checklist that lists all of our requirements. If you have any questions about any of our policies or formatting, please don't hesitate to

contact me.

In sum, we invite you to revise your manuscript taking into account all reviewer and editor comments. We are committed to providing a fair and constructive peer-review process. Do not hesitate to contact us if there are specific requests from the reviewers that you believe are technically impossible or unlikely to yield a meaningful outcome.

We hope to receive your revised manuscript within 4-8 weeks. I would be grateful if you could contact us as soon as possible if you foresee difficulties with meeting this target resubmission date.

- Include a "Response to the editors and reviewers" document detailing, point-by-point, how you addressed each editor and referee comment. If no action was taken to address a point, you must provide a compelling argument. This response will be used by the editors and reviewers to evaluate your revision.
- Highlight all changes made to your manuscript or provide us with a version that tracks changes.

Link Redacted

We look forward to seeing the revised manuscript and thank you for the opportunity to review your work. Please do not hesitate to contact me if you have any questions or would like to discuss these revisions further.

Sincerely,

Giacomo Ariani, PhD
Senior Editor
Nature Human Behaviour

Reviewer expertise:

Reviewer #1: Hippocampal function, navigation, memory, intracranial EEG

Reviewer #2: Hippocampal function, navigation, memory, intracranial EEG

Reviewer #3: Hippocampal function, navigation, memory, intracranial EEG

REVIEWER COMMENTS:

Reviewer #1 (Remarks to the Author):

The authors completely responded to all my concerns and suggestions - I have no more to offer. Congratulations on an excellent piece of work and apologies for a delayed review!

Reviewer #2 (Remarks to the Author):

The revised version of the manuscript and the response to the referees' comments are clear, extensive, and address all our concerns. We recommend the publication of the revised manuscript in Nature Human Behaviour. However, we strongly encourage the authors to include as part of the final version a few analyses and figures that have been presented as part of the referee response but are not part of the revised manuscript. We provide a list of these below, along with a couple of other minor comments.

1. The table provided as part of the response to C2-C (R2) should be included, along with a reference to it in the main text. This information is critical to understanding the time-wrapping procedure. In addition, the authors could clarify what a negative offset means. Does this imply that the participants started imagining the route before the trial actually started?
2. The figure showing a comparison between the scalp and iEEG power spectrum, provided in response C5-D (R2), could be an Extended Data Figure, with reference in the main text where the authors discuss possible links to alpha frequency. We also disagree with the authors' comment here that there is minimal overlap: the figure clearly shows peaks that overlap between the scalp and iEEG spectra, including at theta/alpha ranges. We would thus recommend that the claim of minimal correspondence be toned down when including the figure.
3. The same for the figure provided as part of C2-C (R2) (figure time-on-task). This could also be referenced in the main text.

4. We recommend that the results of the analysis presented in response to C3-C (R2) be included in the final version of the manuscript. While we understand the figure could cause confusion, the question of how the “average” model works for P2, who follows a different route, is an important one to address in the manuscript.

5. Please state which statistical test was used for the p-values in Fig. 4G (Rayleigh test for non-uniformity?).

6. The sentence in lines 207-208 of the revised manuscript (“we observed that theta dynamics were primarily related to the relative position within the route segments ($p = 0.006$), mirroring the patterns observed during real-world navigation.”) can lead to the same confusion that position reconstruction has already been done. Given the clarifications provided by the authors, we assume that none of the results at this point of the manuscript included position reconstruction; if that is the case then “relative position” is misleading. Something along “relative time on trial” would be a more accurate descriptor and would avoid potential confusion for the reader.

As an aside, we find the results on saccadic phase reset very intriguing, as they are in obvious tension with published results. A more detailed analysis is needed to reach a clear conclusion, but we do understand that this is not within the scope of the current manuscript.

Reviewer #3 (Remarks to the Author):

The authors have thoroughly addressed all my comments, and I am happy to recommend the manuscript for publication. I would suggest, however, that the authors still address the following points that follow up on my previous comments.

Major comment C5: I could not follow why “for the imagined condition, the positions are reconstructed from pooled channels, and linearized position is only available at the group level.” Please explain.

Major comment C6: The authors’ control analyses are useful, but I was hoping to see an additional analysis demonstrating a direct relationship between theta amplitudes during real-world navigation and theta amplitudes during imagined navigation (for example across spatial bins). This would provide the most direct evidence for the authors’ claim that there are “analogous functional networks involved in both types of navigation.”

Major comment C11: Could you please clarify what each dot in the scatterplot represents? Is it pooled data across all channels?

Minor comment C5: Could you please provide the statistical output of the cluster-based permutation test? Apologies if I missed it.

Version 2:

Decision Letter:

Our ref: NATHUMBEHAV-24051932B

25th November 2024

Dear Dr Suthana,

Thank you for submitting your revised manuscript “Human neural dynamics of real-world and imagined navigation” (NATHUMBEHAV-24051932B). We have now had a chance to discuss your work internally. I am pleased to report that we find that the paper has sufficiently improved in revision. We will therefore be happy in principle to publish it in Nature Human Behaviour, pending minor revisions to comply with our editorial and formatting guidelines.

We are now performing detailed checks on your paper and will send you a checklist detailing our editorial and formatting requirements within two weeks. Please do not upload the final materials and make any revisions until you receive this additional information from us.

Sincerely,

Giacomo Ariani, PhD
Senior Editor
Nature Human Behaviour

Version 3:

Decision Letter:

Dear Dr Suthana,

We are pleased to inform you that your Article "Human neural dynamics of real-world and imagined navigation", has now been accepted for publication in Nature Human Behaviour.

With best regards,

Giacomo Ariani, PhD
Senior Editor
Nature Human Behaviour

P.S. Click on the following link if you would like to recommend Nature Human Behaviour to your librarian
<http://www.nature.com/subscriptions/recommend.html#forms>

** Visit the Springer Nature Editorial and Publishing website at <http://editorial-jobs.springernature.com?>

utm_source=ejp_NHumB_email&utm_medium=ejp_NHumB_email&utm_campaign=ejp_NHumB">www.springernature.com/editorial-and-publishing-jobs for more information about our career opportunities. If you have any questions please click here.**

Dear Reviewers,

We would like to thank you for your generous and constructive feedback on our manuscript. Each reviewer made suggestions that would greatly strengthen the manuscript. We have now incorporated each of these suggestions into a revised manuscript and detail our changes below (in blue text) made in response to each point raised by the reviewers. We hope that you will find the following updates responsive to your feedback.

Reviewer #1 [R1]: *The manuscript by Seeber et al. describes an interesting study with epilepsy patients implanted with a stimulation and recording device for seizure management, which was used to explore hippocampal LFP activities during real and imagined navigation. Increased spectral power and bouts of oscillations in a broad theta 4-12 Hz frequency range was reported to precede critical moments of turns between segments of walking in a straight path. The same pattern was reconstructed from epochs of imaginary navigation but not from control epochs of walking a treadmill without engaging imagination. Elegant and simple design combined with clear electrophysiological effects between the three conditions make the results of this study compelling in light of the adequately referenced previous work. Implications of the reported results go beyond navigation functions, providing an elegant mechanism for segmenting episodic memory or even organizing behavior at the stage of abstract thinking/mentation. Nature Human Behavior presents a suitable venue for the submitted study.*

Response: We thank the reviewer for their time and effort to review our manuscript, and for the positive feedback. Please find our responses to each additional comment below.

Comment 1 [C1]: *Classifying the LFP activities*

The authors associate the reported effects with 'hippocampal theta oscillations' based on the previous rodent and human navigation studies. Figure 2D shows that the highest amplitude of the spectral activities observed on the group level before the turns is in the classic alpha frequency range (9-14 Hz). There are other spectral activities engaged in this critical moment in the delta and even low beta frequency ranges. Power spectra and the bout prevalence plots in the Extended Data Fig. 2a summarize the group results, showing at least three distinct peaks in this low frequency range of the LFP spectrum at: 2-4 Hz (delta?), 6-10 Hz (theta?), 10-14 Hz (alpha?). It is interesting that the greatest differences between the real, imagined and control conditions are present in the third (alpha?) peak in this plot, which is not really discussed by the authors. Neither are the distinct peaks in the power spectra and their different roles in navigation. The alpha rhythm is historically known to be involved especially with visual perception - I wonder if closing of the eyes could be a factor in the differences observed for this third spectral peak... I would recommend a more cautious reporting of these intricate differences between distinct spectral activities. For example, describing a particular spectral activity in a specific frequency range, e.g. 'increased power in the theta frequency range' is more appropriate than: 'increased power of theta oscillations'. The former doesn't make any assumption about a particular class of oscillations, which is especially useful when other overlapping frequency peaks are present in the same range. A similar predicament has just been reviewed in case of the spectral activities in the high gamma/ripple frequency range (Kucewicz et al. 2024 Brain). I suggest revising the phrasing used throughout the manuscript and discussing the other peaks, especially the nearby peak in the alpha frequency range. An alternative would be to use 'low' and 'high theta' oscillations (Goyal et al. 2020 Nature Comm) but these referred to more delta (2-3 Hz peak) and actual theta (8 Hz peak), respectively, more than the alpha peak observed here.

We thank the reviewer for raising these important points. In response to the reviewer's comment, we have included individual frequency spectra (previously shown in Extended Data Fig. 2a, now in Extended Data Fig. 5a). Across all participants, we observed three distinct peaks in the power spectra: ~2 Hz, between 3-10 Hz, and ~16 Hz. Based on previous literature on hippocampal recordings¹⁻⁴, we refer to these as delta, theta, and beta, respectively. In the classical EEG literature, the term 'alpha' typically refers to neocortical activities in visual and sensorimotor areas^{5,6}, while ~8 Hz within the medial temporal lobe is most often referred to as 'theta'^{7,1-4}. However, as the reviewer suggested, we have revised the language throughout the manuscript to describe findings more cautiously as 'increased power in the theta range' rather than assuming specific oscillatory classes (e.g., lines 103, 111, 134-135, 269-270, 333, 353) and expanded the results to include other spectral peaks (pg. 4):

"In addition to the activities at theta frequencies, we observed two additional spectral peaks in the delta (1-3 Hz) and beta (13-30 Hz) frequency ranges (Extended Data Fig. 5c). We analyzed the activity within these frequency bands using the same approach as for theta, examining amplitude dynamics as a function of position during real-world navigation. The time-varying amplitudes in the delta range showed a dependency on route segments

(Extended Data Fig. 10a) comparable to theta, while the beta frequency activity increased during the third route segment (Extended Data Fig. 10c). During this segment, participants performed an additional task of identifying the closest hidden object at a random moment, signaled by an auditory cue.”

Furthermore, we describe the temporal consistency in the delta and beta frequency bands (pg. 6):

“In addition to the theta frequency band (Fig. 3d), temporal consistency across trials was significantly higher during real-world navigation compared to the control condition for the delta (Extended Data Fig. 10b) and beta (Extended Data Fig. 10d) frequency bands (all $p < 0.001$). Delta frequency dynamics during imagined navigation were more consistent compared to the control condition ($p = 0.011$, Extended Data Fig. 10b), whereas beta frequencies did not show consistent dynamics ($p = 0.128$, Extended Data Fig. 10d). Furthermore, the effect size (Cohen’s d) of temporal consistency differences between imagined navigation and the control condition was more pronounced in the theta range ($d = 1.76$) than in the delta range ($d = 0.76$).”

We have also added a discussion on the terminology and the activities in specific frequency ranges, including the novel results in the delta and beta frequency range to the manuscript (pg. 12):

“In line with previous literature on hippocampal recordings in humans, we observed spectral peaks at ~4 Hz and ~8 Hz, at the edges of the traditional theta frequency range¹⁻⁴. Consequently, we used individualized theta frequency ranges within the broader 3-12 Hz range. Although these frequencies partly overlap with scalp EEG alpha rhythms, oscillations around ~8 Hz in the MTL are functionally distinct from the alpha activities recorded over visual and sensorimotor areas on the scalp^{7,6}. Future research is needed to explore the relationship between MTL activities in the theta range and scalp recordings. Outside the theta range, we observed spectral peaks around 2 Hz in the delta range and approximately 16 Hz in the beta range. Delta-range activities overlap with the “slower theta” rhythms reported in episodic memory tasks^{1,8} and view-based navigation in virtual environments^{3,9}. This overlap may explain the structured dynamics at 1-3 Hz during both real-world and imagined navigation. However, temporal consistency effects during imagined navigation were more pronounced in the theta range than in the delta range. Beta-range activities may be linked to the self-location sub-task. Increased MTL oscillatory activity in the beta range has been observed during fast walking versus slow walking⁴, while decreased activity in the beta range has been reported during episodic memory retrieval in stationary participants¹. Additionally, a recent study in freely moving rodents suggested that beta activities might result from two independent theta-frequency inputs representing forward and reverse trajectories at roughly opposite theta phases¹⁰. The precise nature of these beta activities remain an open question, with ongoing debate about whether they are related to sensory processing¹, reflect harmonics due to changes in theta oscillation shapes⁴, or result from two distinct theta-frequency inputs to the hippocampus¹⁰.”

Extended Data Fig. 5 | Frequency spectra and theta dynamics. c. Normalized power spectra (top) and the prevalence of oscillatory bouts (bottom) are shown as averages across participants (group) and individually for each participant (P1-P5).

Extended Data Fig. 10 | **a**, Amplitude dynamics in the delta frequency range are overlaid onto motion trajectories, averaged across participants. **b**, Temporal consistency of delta dynamics was significantly higher during both imagined and real-world navigation compared to sole treadmill walking (control condition). **c**, Amplitude dynamics as a function of position on motion trajectories are shown as averages across participants. Note the increased activity during the third (downward) segment of the routes, where participants estimated and reported their location relative to hidden objects. **d**, Temporal consistency of beta dynamics was significantly higher during real-world navigation compared to both treadmill walking conditions.

Extended Data Fig. 9 | Self-location task. Time-frequency (TF) activity of the self-location task is shown for the group (top left panel) and each participant (average across channels). TF plots are aligned to auditory cues ($t = 0$, black lines), which signal participants to report their location relative to learned positions.

Finally, we have clarified in the Methods section that participants were instructed to keep their eyes open during all treadmill walking trials, including the imagined navigation trials (pg. 14):

“During all treadmill walking trials, participants were instructed to walk naturally and face their gaze forward.”

Based on the reviewer’s later suggestion, we also analyzed the percentage of time when the eye-tracking signal was lost in each condition and found no significant increase during imagination compared to real-world navigation ($p = 0.125$, Wilcoxon signed-rank test, real-world navigation: $24.3 \pm 12.1\%$, imagined navigation: $16.1 \pm 13.0\%$, control: $13.3 \pm 9.8\%$). These percentages were numerically lower during imagined navigation than during real-

world navigation. Since participants were walking in all experimental conditions —either adapting their trajectory to follow the planned route during real-world navigation or maintaining their position on the moving treadmill during imagined navigation —this constant engagement likely reduced instances of eye closure.

We hope these revisions address the reviewer’s concerns and enhance the clarity and accuracy of our findings.

C2: Bouts of oscillations

Observation of the bouts or transients of oscillations peaking at the same time as the increases in the spectral power is intriguing and suggests that the increased spectral power is driven by increased incidence/rate of these bouts. This would imply that the transients are a fundamental unit/event underlying the spectral power increases. Investigating these a little more in terms of their duration, amplitude and peak frequency could support associating the reported changes in the spectral power with actual oscillations and not other non-oscillatory sources of power (e.g. filtering of interictal epileptiform spikes or other artifacts). The authors would then be able to make more grounded claims about theta or alpha 'oscillations' rather than just spectral power.

It is interesting that similar oscillations in the higher frequency ranges have also been reported as these bouts, transients or bursts. Examples include beta/gamma bursts in monkeys (Lundqvist 2016 Neuron) or bursts of gamma, ripple and fast ripple oscillations in humans (see the Brain review above) associated with memory and cognitive functions. The principle appears to be the same across the frequency spectrum - it's the duration and amplitude of the transients that is proportionally increasing going down the spectrum. Thus, high frequency oscillations (HFOs) in the ripple and fast ripple ranges and the shortest in duration and lowest in amplitude on one side of the spectrum, whereas the gamma, beta, alpha and theta transients are progressively longer and stronger in amplitude according to the power law.

I recommend describing the reported bouts further in the manuscript.

Following the reviewer’s recommendation, we added data (Extended Data Fig. 5) quantifying bout properties across all frequencies, including bout amplitude, duration, and individual bout prevalence.

Extended Data Fig. 5 | Frequency spectra and theta dynamics. a, Frequency spectra of oscillatory bout amplitudes (top) and the average number of cycles (bottom) are depicted as averages across participants for real-world navigation (real), imagined navigation

(imag), and the control condition (ctrl). **b**, Theta amplitude dynamics and time-resolved theta rates, z-scored for comparison (left), along with theta bout rates (\pm SEM) shown as detection percentages across trials (right) for left (top) and right (bottom) walks. Vertical lines represent turns. **c**, Normalized power spectra (top) and the prevalence of oscillatory bouts (bottom) are shown as averages across participants (group) and individually for each participant (P1-P5).

Additionally, we now include the average prevalence (21.2 ± 6.6 %) and duration (0.524 ± 0.077 seconds) of oscillatory bouts in the theta frequency range in the main manuscript along with reference to the new related results (pg. 4):

“The mean prevalence of theta bouts was 21.2 ± 6.6 %, with an average duration of 0.524 ± 0.077 seconds across participants similar to previous reports^{4,13}. We analyzed the timing of these bouts of increased amplitudes at theta frequencies on a single-trial level and observed their alignment with upcoming turns (Fig. 3b). Strikingly, the percentage of trials in which theta bouts occurred at specific time points closely mirrored the trial-averaged amplitude dynamics in the theta frequency range (Extended Data Fig. 5b).”

Lastly, peak frequencies within the theta frequency range are now provided in Extended Data Table 1:

Participant		1	2	3	4	5
Age [years]		36	45	24	39	30
Gender		male	male	female	male	female
Theta range [Hz]		4.3 - 11.3	3.0 - 9.9	5.7 – 11.3	4.9 – 11.3	3.0 - 9.8
Theta peak [Hz]		8.6 ± 0.3	3.5 ± 0.1	8.6 ± 0.3	8.6 ± 0.3	8.6 ± 0.3
Lead 1	Hemisphere	left	left	left	left	left
	# MTL chan	2	2	1	2	2
	chan 1	HP/HP	HP/PRC	HP/PRC	Amy/HP	HP/HP
	chan 2	HP/Sub	PRC/PHC		HP/HP	HP/HP
Lead 2	Hemisphere	right	right	right	right	right
	# MTL chan	2	1	2	2	2
	chan 3	HP/HP	PHC/Fs	HP/PRC	Tp/PRC	HP/HP
	chan 4	HP/Sub		PRC/PHC	PRC/HP	HP/PHC
Real-world left		36	35	40	34	36
Real-world right		35	34	31	35	35
Imagined left		23	25	24	25	24
Imagined right		20	24	22	24	24
Control		25	24	24	23	24

Extended Data Table 1 | Participant demographics, theta frequency ranges, theta peaks, electrode implantation sites, and the number of trials completed for real-world left and right walks, imagined left and right walks, and treadmill control walks are provided. The theta range was determined using spectral peaks (\pm frequency bin width, see Methods) within the 3-12 Hz range, along with neighboring lower and higher spectral minima. The localization of each of the two contacts forming a bipolar channel (chan 1 and 2) is specified for various medial temporal lobe (MTL) regions, including the hippocampus (HP), perirhinal cortex (PRC), parahippocampal cortex (PHC), subiculum (Sub), Fusiform gyrus (Fs), temporal pole (Tp), and amygdala (Amy).

C3: Saccades during imagined navigation

It is not clear to me from reading the manuscript text how the saccades were detected during the imagined navigation. It is our natural tendency to close the eyelids during imagination even if one is instructed not to do so. It would be very useful for this study to report the proportion of time when the eye-tracking signal was not detected in the three conditions. My prediction is that the imagined condition would have the highest proportion. We typically see in our studies that this proportion is much higher during the recall phase of a memory task than during encoding, even though both use a fixation cross in the middle of the screen with subjects instructed to focus gaze. It turns out that when we imagine and recall memories then our gaze tends to shift to a specific corner of our field of view. In this case the three conditions cannot be compared as presented in the Extended Data Fig. 8, unless the proportion of time fixating on the center of the field is comparable.

We thank the reviewer for these insightful comments. We analyzed the percentage of time when the eye-tracking signal was lost in each condition and found no significant increase during imagination compared to real-world navigation ($p = 0.125$, Wilcoxon signed-rank test, real-world navigation: $24.3 \pm 12.1\%$, imagined navigation: $16.1 \pm 13.0\%$, control: $13.3 \pm 9.8\%$). These percentages were numerically lower during imagined navigation than during real-world navigation. Since participants were walking in all experimental conditions —either adapting their trajectory to follow the planned route during real-world navigation or maintaining their position on the moving treadmill during imagined navigation —this constant engagement likely reduced instances of eye closure.

In our analyses, saccades were detected using the ClusterFix toolbox, which we have included in our manuscript (pg. 19):

“Saccadic eye movements were identified using the ClusterFix toolbox¹¹, which identifies periods of gaze fixation using k-means clustering, and transitions between these fixations are classified as saccades.”

Furthermore, in response to the reviewer’s comment related to Extended Data Fig. 8 we have clarified the associated methods (see Methods, pg. 16):

“To explore the potential influence of eye movements on MTL recordings, we computed average time-frequency activities aligned with the onset of eye saccades and compared these plots with those aligned with turns (Extended Data Fig. 8b-e). Four saccades (Extended Data Fig. 8c) were randomly selected per trial to correspond with the four turns (shown in Extended Data Fig. 8b). Additionally, we ensured that each fixation preceding a saccade lasted longer than 600 milliseconds, as described in previous work²⁷. This approach allowed us to match the number of observations across conditions precisely, as depicted in Extended Fig. 8b-e.”

Extended Data Fig. 8 | Gaze densities and MTL activity aligned to turns and saccades. **b-c**, Time-frequency (TF) activity aligned to turns (**b**) and saccades (**c**) during real-world navigation. **d-e**, Time-frequency activity aligned to eye movements (saccades) during imagined navigation (**d**) and the control condition of treadmill walking without imagination (**e**). Note pronounced theta activities precedes physical turns but not saccades.

Reviewer #2 [R2]: *This is a highly interesting study that provides insight into unique intracranial data in freely moving humans. In particular, the here recorded hippocampus/MTL electrophysiology allows unparalleled comparisons to previous influential animal studies on navigation and memory as well as previous studies on virtual navigation in humans.*

This generally enthusiastic assessment is, however, limited by the amount of detail that is missing in the current presentation of the data, which, at this stage, prevents an in-depth understanding of the reported results. The most glaring of such missing details are regarding how time alignment was done at various stages of the analysis, especially with the imagined navigation part of the experiment; this is critical to the main conclusions of the paper and the amount of detail is very limited and often unclear.

Response: We thank the reviewer for their positive feedback and for taking their time to review the manuscript. We have revised the manuscript to incorporate the critical details suggested by the reviewer. Please find details of these changes below.

Major Comments:

C1: *On several occasions throughout the manuscript, labels are missing, in particular for colorbars; sometimes, the colorbar itself is missing. Since it is not possible to fully understand figures without units, the authors should carefully revise their manuscript and add the missing information.*

Some examples: Figs. 4a, ext. data Fig. 4: colorbar missing

Figs. 2c,d, 4c, ext. data Fig. 7: colorbar units missing.

We apologize for these omissions and have now added color bars and units wherever they were missing or previously only mentioned in the caption.

C2: Imagined navigation

The entire analysis with imagined navigation trials is very difficult to understand. Could the authors be more explicit about how the time alignment was done? More specific concerns follow:

A) *For the correlation coefficients in Figure 3(d), were these derived after the time alignment procedure laid out in lines 495-498, which involves the real-world navigation data? If so, doesn't this already lead to a spurious intratrial correlation?*

Assuming that a different method was used than this, when the authors "reconstructed" positions, we are wondering how this was done here for their imagined condition. There will be an offset between starting to walk on the treadmill and imagination of the current location on the path. How did the authors locate turns in imagined walks?

We thank the reviewer for raising this important point. For the analysis shown in Fig. 3d, the reconstruction of imagined positions was not yet performed. Instead, we focused on examining whether any temporal structure existed in the data, separately for each condition. In the imagination trials, the trials were aligned to each other based on the average time across all imagination trials, rather than being aligned to the real-world navigation data. We have edited the description on the time-warping process for the treadmill trials, which is needed to compare the trials within each condition (imagination and control) to each other (pg.15):

"The imagined navigation and control trials were linearly warped based on the average trial duration within each condition. To evaluate whether theta dynamics exhibited temporal structure across trials, we computed the temporal consistency by correlating the mean signals from randomly splitting trials within each condition."

Similarly, the control and real-world navigation conditions were analyzed by aligning trials within each condition independently. As shown in Fig. 3d, there is no significant temporal consistency during the control condition, but temporal consistency was observed during both the imagined and real-world walking conditions. Since temporal consistency is computed across trials within the same condition, neither absolute time nor alignment between conditions influences the results. Therefore, we can exclude the possibility that this approach caused spurious intra-trial correlations. For further details regarding the alignment across conditions, we refer to our response to R2, Major C2C below.

B) *As above, a description of Fig. 3c in the results section is missing. How was this output created? In Figure 3,c, the author's comment that "Note that theta activity tended to increase earlier during imagined turns compared to real-world turns." appears unsustainable to us given that the data from real-world navigation was used to*

determine the imagined position in the first place and so any inferences about timing is a circular argument. In fact, it is puzzling that there is such a consistent pattern given that the time alignment procedure would lead to theta bursts coming at very similar pre-turn intervals as in the real-world navigation.

To prevent any confusion, we have replaced Fig. 3c with an illustration of theta activity during imagined navigation for an exemplary channel, rendered on the group mean motion trajectory. This result is derived from our within-condition alignment approach, rather than from the reconstruction analysis.

Fig. 3 | Comparative theta dynamics of real-world and imagined navigation. c, Theta activity during imagined navigation of an exemplary channel rendered on the group mean motion trajectory.

We also have now included a description of Fig. 3c in the Results (pg. 6), and Methods (pg. 15) sections:

“Given the pronounced presence of task-related structured theta dynamics during real-world navigation, we explored whether analogous temporal patterns were evident during imagined navigation (Fig. 3a,c).”

“We first illustrate theta dynamics during imagined navigation, assuming the same time-to-position mapping as in real-world navigation (Fig. 3c), before reconstructing positions from neural data.”

Furthermore, as suggested by the reviewer, we have removed the comment *“Note that theta activity tended to increase earlier during imagined turns compared to real-world turns”* from the Fig. 3 caption.

C) *Is it possible that the act of imagining a route causes theta bouts but not necessarily in the same task-structure dependent way as in real-world navigation? For example, if the act of imagining spatial navigation itself leads to an enhanced theta activity, one would end up simply aligning the peaks of the theta activity via the time alignment procedure and cause peaks to appear at “imagined turns”. In this context, it would be useful to know how much time stretching and shrinking was typically needed for the alignment.*

We appreciate the reviewer’s insightful comment. In response, we examined the prevalence of theta bouts and found no credible evidence for a higher prevalence during the imagination condition compared to the control condition ($p = 0.47$), suggesting that the alignment procedure had an equal chance of aligning the data across both conditions. Importantly, we used the same algorithm for both imagined navigation and control trials, with cross-validation to prevent overfitting. However, the fact that reconstruction errors were significantly lower in the imagined navigation condition than in the control condition strongly supports the conclusion that the observed temporal consistency is not merely an artifact of the alignment process. This difference in reconstruction accuracy indicates that the imagined navigation trials contain meaningful temporal dynamics that are more consistent with real-world navigation, beyond what would be expected from chance alignment alone.

Additionally, we want to emphasize that all results involving the alignment procedure were cross-validated. The results presented in the manuscript are derived from parts of the data that were not seen by the alignment procedure, ensuring that the findings generalize beyond the data used to identify time offsets for alignment across trials and navigation conditions. Based on a comment from another reviewer (R3, Major C9) we included a more detailed description of the cross-validation procedure in the Methods section (pg. 17-18):

“To prevent overfitting, we employed 10-fold cross-validation repeated 10 times to learn the temporal relationship from sub-partitions of the data. Each participant’s single trials were randomly grouped into ten subsets, each containing roughly 10% of the unique trials. From these subsets, nine (~90% of the trials) were used to train the alignment of time and position during imagined navigation. The trained alignment was then tested on the remaining subset of previously unseen imagination trials, serving as a test of the alignment algorithm’s

generalizability. This process was repeated for all ten combinations of training and test sets. Additionally, the random grouping of subsets was repeated, and the training and testing procedure was applied across ten repetitions, resulting in a total of 100 iterations. Each iteration produced position estimates computed from the test set trials, independent of the subset used for training the alignment. The outcomes of these generalizations are presented throughout the manuscript.”

The alignment procedure allowed for limited temporal stretching or shrinking within a two-second window to accommodate variability across trials. This restriction ensured that the alignment did not overly stretch periods of theta activity, minimizing the risk of artificially aligning theta peaks that may not correspond to actual task-structure dependent events such as imagined turns. We have added these details in the Methods section (pg. 17):

“We utilized the position estimation model developed from our real-world walking data to derive an initial estimate of imagined positions. Subsequently, we applied dynamic time warping to align these estimated relative positions with the task structure, allowing for a maximal time shift of ± 2 seconds. This alignment procedure was conducted for each participant while preserving the relative timing between recording channels within each participant.”

Within this range, the imagination trials exhibited a time offset, as summarized in the table below.

	Time offset [s]
P1	0.19 ± 1.44
P2	0.06 ± 1.14
P3	-0.76 ± 1.47
P4	-0.21 ± 1.19
P5	0.35 ± 1.09

Time offsets (mean \pm standard deviation) used for temporal alignment of imagined and real-world navigation data determined using the statistical model. The standard deviation represents the fluctuation across the iterations of the cross-validation procedure.

Given that the imagination trials exhibited a time offset of under two seconds, this suggests that the alignment procedure effectively accounted for temporal variability without overfitting the data. The consistency of this offset across trials indicates that the observed theta activity is likely tied to the task structure, rather than being an artifact of the alignment process.

D) Related to the above, when comparing their navigation/imagination data to the control trials, it seems that these comparisons are always confounded by time on task. In particular, when comparing the temporal consistency across conditions this could be an issue. Can the authors exclude the possibility that the correlation values were only low in the control condition because it was always the first part of the experiment?

We appreciate the concern regarding the potential confounding effect of "time on task" when comparing the temporal consistency across conditions. We carefully considered this issue during the design phase of the experiment. However, we prioritized ensuring that participants were naïve to the imagination task during the control condition, as any prior mention of imagination could compromise the validity of the control.

To further investigate the potential impact of time on task, we conducted a supplementary analysis in which we split the control data into two parts: the first half and the second half of the trials. If there were a time-on-task effect, we would expect to observe higher temporal consistency in the second half of the trials. However, our analysis found no significant difference between the two halves ($p = 0.53$), indicating that time on task did not influence the temporal consistency in the control condition. These results are illustrated in the figure below:

Time-on-task effect. Temporal consistency was computed separately for the first and second halves of the treadmill control (ctrl) condition, and no evidence of a time-on-task effect was found for this control condition.

E) The possibility of a more informative metric to show that the imagined navigation has a similar task-related structure should be explored. For example, would it be possible to use motion/eye tracking to infer when the person may be executing an imagined turn on the trajectory?

Following the reviewer's suggestion, we attempted to align the imagination data with behavioral variables to enhance the inter-trial alignment of gaze and head motion data and applied the same time-warping to the neural data. We used temporal consistency as an indicator to quantify whether theta dynamics were structured similarly across trials, as shown in Fig. 3d of the manuscript. If these theta dynamics were straightforwardly dependent on gaze or head motion, aligning to these behavioral variables across trials would have been expected to increase the temporal consistency of theta activity. However, despite aligning to gaze position and head rotation, we found no evidence of improved neural consistency after aligning to gaze ($p = 0.44$, left panel) or head movements ($p = 0.54$, right panel), as shown in the Figure below. For further details, see response to R2, Major C2D above.

Impact of Behavioral Alignment. Using gaze trajectories ('gaze warp', left panel) and head movements ('head warp', right panel) to enhance the inter-trial alignment of gaze and head motion data did not improve the temporal consistency of the theta dynamics when warped using these behavioral variables. Both warping procedures using behavioral variables were compared to the linear time-warping using the average duration across all imagination trials ('time warp') described in the main manuscript (Fig. 3d).

This suggests that while gaze and head motion may be correlated with navigational events in real-world settings, their influence on neural activity during imagined navigation may be less straightforward.

C3: Results related to Fig. 4 and spatial decoding.

A) How exactly was the spatial decoding done? The authors refer to Agarwal et al., 2015. However, that method was dependent on high-density spatial sampling of LFPs, which was not the case here (max. 2 bipolar pairs per hemisphere). What was changed and how and why would it still work?

In our model, we pooled data across participants, utilizing all 18 channels to estimate position. While our approach was inspired by Agarwal et al. (2015)¹², particularly in modeling linear position as a circular variable using cosine and sine components, it does differ due to the lower spatial sampling density in our study (with a maximum of 2 bipolar pairs per hemisphere). The method in Agarwal et al. demonstrated the feasibility of

decoding position from LFP data, which motivated our simplified reconstruction model as a proof of concept. Despite the lower spatial resolution, we believe our approach remains effective because the variance in theta dynamics—specifically, the theta peak relative to the turns, which spans more of the walking segments—across participants and channels provides enough unique information. Each channel contributes independent data, and when combined, this enhances decoding performance. We have added further details to the Methods sections to clarify our reasoning and approach (pg. 8):

“However, distinct recording electrodes exhibited peak activations at slightly varying positions within a route segment (Fig. 4f). In light of this finding, we developed a model that considered the unique timings of each electrode to capture the relative positions of route segments, treating them as two-dimensional circular variables, a methodology based on prior research¹². Modeling the linearized position as a circular variable is a useful mathematical abstraction where movement can be thought of as a change of position on a ring. In our case, each full circle on that ring would correspond to the traversal of one route segment. This approach allows us to capture the continuous, cyclical modulation of theta activity across route segments, which can vary smoothly over time. By using sine and cosine functions to model theta dynamics, we account for these cyclical changes more effectively. Using a linear regression model, we used theta dynamics from all MTL channels, pooled across participants, as predictors and using the route structure (comprising cosine and -sine phase alignment) as response variables. Regression coefficients were learned from subsets of the data, and the model’s ability to generalize to unseen trials was tested (using 10 x 10 cross-validation) to prevent overfitting.”

In conclusion, while our approach adapts the spatial decoding model for lower spatial sampling, the combination of pooled data across participants and the use of sine and cosine functions to capture cyclical theta dynamics ensures that the model remains capable of decoding positional information, even with fewer electrodes per participant. We believe this adaptation preserves the task-related structure of the theta activity, allowing us to effectively estimate position across route segments despite the reduced spatial resolution.

B) The authors then applied the position reconstruction model to imagined navigation data and aligned estimated imagined positions and the task structure using linear time warping. How exactly was the linear warping done? Please provide more details in both the methods and results section. How would offsets and breaks during imagination not be problematic? Was the alignment using linear warping also done for real-world and control walks? How does warped alignment in these conditions compare to unwarped alignment?

Thank you to the reviewer for the thoughtful feedback. We have provided additional clarification on the linear time warping procedure and addressed concerns regarding offsets and breaks during imagination.

We first computed an initial estimate of reconstructed positions by applying the statistical model, trained on real-world walking data, to both imagined navigation and control walks. Since we expected the timing of imagined navigation to differ from that of real walks, we could not directly apply the same time-to-position mapping. To address this, we used dynamic time warping separately for imagined navigation and control walks to identify time points associated with real-world turns. For each condition, this timing-to-position mapping was determined using its respective training set from its cross-validation data. Once the key time points were identified, we performed a piecewise linear warping for each segment of the imagined navigation and control data. This approach allowed for time shifts of up to ± 2 seconds between the initial position estimate and real-world positions, accounting for variability in timing between trials. Notably, this time-to-position mapping was subsequently evaluated with the cross-validation testing set, which was unseen to the alignment procedure. This warping ensured that even if imagined turns occurred at slightly different times relative to real-world turns, the estimated positions remained aligned across conditions.

To mitigate the potential issues with offsets and breaks during imagination, we constrained the warping within each segment, preventing excessive stretching or compressing that could distort the alignment. The same warping procedure was applied to the control condition to maintain consistency. The actual position reconstruction was then evaluated using the testing set, which underwent the same alignment and warping procedure.

We acknowledge that offsets and breaks during imagined navigation could indeed pose challenges, which likely contributed to the reduced decoding performance observed in the center of segments, as shown in middle panel of Fig. 4g. This point is addressed in the caption for Fig. 4g and further elaborated on in the Methods section (pg. 17-18, text added below for your review). In summary, the time warping algorithm was applied uniformly across the imagined navigation and control conditions, enabling direct comparisons of position reconstruction across these conditions. The analyses are condition-agnostic, allowing for consistent evaluation of all data.

However, for real-world navigation, time warping was unnecessary because positions were directly obtained via motion tracking, which served as the basis for training the statistical model that mapped neural data to positions.

We added a more intuitive description in the Results section (pg. 8), described the warping procedure in more detail in the Methods section (pg. 17-18), and emphasizing that the warping procedure is effective only when different imagination trials exhibit sufficient similarities:

“We utilized the position estimation model developed from our real-world walking data to derive an initial estimate of imagined positions. Subsequently, we applied dynamic time warping to align these estimated relative positions with the task structure, allowing for a maximal time shift of ± 2 seconds. This alignment procedure was conducted for each participant while preserving the relative timing between recording channels within each participant. To prevent overfitting, we employed 10-fold cross-validation repeated 10 times to learn the temporal relationship from sub-partitions of the data. Each participant's single trials were randomly grouped into ten subsets, each containing roughly 10% of the unique trials. From these subsets, nine (~90% of the trials) were used to train the alignment of time and position during imagined navigation. The trained alignment was then tested on the remaining subset of previously unseen imagination trials, serving as a test of the alignment algorithm's generalizability. This process was repeated for all ten combinations of training and test sets. Additionally, the random grouping of subsets was repeated, and the training and testing procedure was applied across ten repetitions, resulting in a total of 100 iterations. Each iteration produced position estimates computed from the test set trials, independent of the subset used for training the alignment. The outcomes of these generalizations are presented throughout the manuscript. It is important to note that this alignment procedure is only valid if the theta dynamics from different imagination trials exhibit similarities. The same procedures for position estimation and time alignment were applied to the sole treadmill walking data, serving as control analyses. By ensuring that results from all conditions were temporally aligned, we were able to calculate position reconstruction errors across all conditions using the actual positions recorded during real-world navigation trials (Fig. 4h). Since relative position was modeled as a circular variable, we present reconstruction errors as polar histograms, showing angles between actual and estimated positions.”

This alignment via time warping compensates for individual imagined velocity profiles, resulting in time-to-position mappings that are valid as long as they are consistent across imagination trials. We explicitly tested the generalization of these time-to-position mappings to unseen data for both the imagined navigation and control walks. Without alignment, we would assume that time-to-position mappings are constant or the same as during real-world walking. As the reviewer points out, such an assumption would not be valid. Therefore, unwarping comparisons are less accurate, particularly for imagined navigation, where internal timing may differ from the real-world task structure. By allowing small time shifts (with a maximum of ± 2 seconds; for actual time shifts, see response to R2, Major C2B), the alignment process compensated for these variations, leading to improved reconstruction accuracy.

C) Was the model depicted in Fig. 4a also used with P2's data? Since P2 used another route, can the authors adjust the model and see if outcomes are improved?

We appreciate the reviewer's suggestion to adapt the model for P2's data, considering the different route taken. After implementing this adjustment, we found that the decoding results remained similar to those obtained with the original model:

Modified position reconstruction. The modified cosine route representation included an additional turn in participant 2 (left panel). Position reconstruction results for the group using this modified model for participant 2 are shown in the right panel. For comparison, the original model with one less turn is shown in Fig. 4g.

Since each of the five participants performed four left and four right turns as instructed, the statistical model was informed by a total of 40 turns, or 50 segments. It is likely that this one additional turn had only a minor effect on the reconstruction results, which explains the similarity in outcomes. Therefore, we kept the original model in the manuscript.

D) Fig. 4c: is “#electrode” arranged in an order that corresponds to individual participants? e.g. do electrodes 15 to 18 belong to a single patient? If yes, does this explain the variation across electrodes depicted in Fig. 4c?

We confirm that the electrodes are indeed sorted across participants in Fig. 4f (previously Fig. 4c). The variation observed across electrodes reflects the differences in peak theta response within each participant. For instance, you can see this variability in the peak theta responses of electrodes 1-4 in P1 and electrodes 15 versus 16-18 in P4. We did not apply any smoothing to the data in Fig. 4f, which allows these differences to be clearly observed.

E) How does the lag, provided as the number of segments in Fig. 4b, match the length of one segment on a walking route?

For these analyses, we employed a time-warping method to normalize the length of each segment across all conditions to the mean length of the real-world walking segments. This normalization ensures that the segments used in our analysis are of equal duration, independent of their actual physical length. Consequently, the "lag [segment]" on the x-axis of Fig. 4e (previously Fig. 4b) reflects this normalized segment length.

In our analysis, we calculated the cross-correlation between the sinusoidal model and the outcome of the reconstruction for both imagined navigation and control walks. By normalizing the segment lengths, we ensure that the cross-correlation analysis is consistent across segments. This approach enables us to detect recurring patterns and compare theta amplitude dynamics in a way that is independent of the original segment lengths.

The symmetric and repetitive peaks observed in the cross-correlation function thus correspond to the increases in correlation between the theta dynamics and the sinusoidal model when shifting by multiples of the normalized segment length. Specifically, if the theta dynamics are similar across segments, the cross-correlation will show symmetric, repetitive peaks at intervals corresponding to normalized segment lengths or the equivalent of one walking segment. We have clarified this approach in the updated Results and Methods sections.

Results (pg. 8):

“Cross-correlations between theta range dynamics and route layout, after normalizing for segment length discrepancies, revealed four lateral peaks at multiples of the normalized segment length, corresponding to the four turns on each walking route (Fig. 4e).”

Methods (pg. 15-16):

“To analyze the similarity of theta dynamics within each segment of a walking route, we abstracted the navigational task structure into a sinusoidal pattern that peaked at each turn. This approach ensured that each segment adhered to the same sinusoidal pattern, scaled according to its normalized length or relative time. The phase of this sinusoidal pattern corresponded to the relative position within each segment. For both imagined navigation and control walks, we first normalized segment lengths to the mean length of the real-world walking segments. This normalization process ensured that all segments in our analysis were of equal duration, regardless of their original physical length. We then computed the cross-correlation between theta dynamics (reconstruction output) and the sinusoidal model of each walking segment. The central peak of the resulting cross-correlogram represents the correlation between theta dynamics and the sinusoidal task structure, effectively capturing the spatial layout and timing across each walking route. Additionally, any side peaks in the cross-correlogram indicate similarities between theta dynamics and shifted task structures (note increases at multiples of a segment), further confirming the consistency of theta activity patterns relative to the route’s structure.”

F) Fig 4c shows that theta is high at before a turn and that activity lasts until shortly after the turn. Why does one need sine and cosine in Figure 4c? Would there be a simpler way to model this?

We appreciate the reviewer’s question. The use of sine and cosine functions allows us to capture the continuous, cyclical modulation of theta activity across route segments, which can vary smoothly over time. Since theta activity can peak at different points relative to a turn (e.g., some channels peak before, while others peak during a turn), modeling them with sine and cosine components helps us account for these cyclical changes more

effectively. A simpler approach, such as a boxcar model, might detect specific sections of the path but would be unable to capture the temporal dynamics inherent in theta modulations. We tested the simpler boxcar model (below) but found that the sine and cosine-based model produced higher correlations and more accurately reflected the gradual build-up and decay of theta activity before and after turns. Our approach, as mentioned, was inspired by Agarwal et al. (2015)¹², where linear position is modeled as a circular variable. This mathematical abstraction allows us to treat position as if it were on a ring, with each complete circle representing the traversal of a segment. We used sine and cosine components to effectively capture the periodic nature of theta activity around a turn as we now further describe in the manuscript (pg. 8):

“Modeling the linearized position as a circular variable is a useful mathematical abstraction where movement can be thought of as a change of position on a ring. In our case, each full circle on that ring would correspond to the traversal of one route segment. This approach allows us to capture the continuous, cyclical modulation of theta activity across route segments, which can vary smoothly over time. By using sine and cosine functions to model theta dynamics, we account for these cyclical changes more effectively.”

In response to the Reviewer 3’s suggestion (see response to R3, Major C13), we also tested a simpler boxcar model:

Cross-correlation of theta dynamics using a boxcar model, adjusted for segment length discrepancies, compared to the circular (cosine/sine) model in the main manuscript.

However, this model (left panel) resulted in lower correlations (right panel), as shown in the Figure above, compared to the one using sine and cosine components (Fig. 4a,e), which is included in the manuscript. These results further reinforce the effectiveness of our chosen approach.

C4: Theta Bouts

A) How do the theta bouts in this study compare to previous studies in terms of peak frequency and duration? E.g., Aghajan reports theta bouts with average durations of 400 ms, which seems rather short as compared to what is shown in Fig. 3b.

We expanded our analysis in Extended Data Fig. 5 to include additional data on theta frequency bouts, such as bout amplitudes and durations measured in cycles. Additionally, we included the following in the Results section (pg. 4):

“The mean prevalence of theta bouts was 21.2 ± 6.6 %, with an average duration of 0.524 ± 0.077 seconds across participants similar to previous reports^{4,13}.”

These findings are consistent with results reported by Aghajan et al.⁴, (e.g., theta bouts with average durations around 400 ms). Our average bout duration of approximately 524 ms aligns well with Aghajan’s findings, suggesting that the theta bouts observed in our study are of similar duration.

Extended Data Fig. 5 | Frequency spectra and theta dynamics. **a**, Frequency spectra of oscillatory bout amplitudes (top) and the average number of cycles (bottom) are depicted as averages across participants for real-world navigation (real), imagined navigation (imag), and the control condition (ctrl). **b**, Theta amplitude dynamics and time-resolved theta rates, z-scored for comparison (left), along with theta bout rates ($\pm\text{SEM}$) shown as detection percentages across trials (right) for left (top) and right (bottom) walks. Vertical lines represent turns. **c**, Normalized power spectra (top) and the prevalence of oscillatory bouts (bottom) are shown as averages across participants (group) and individually for each participant (P1-P5).

B) Please also clarify the following issue:

I. 124: “We analyzed the timing of these bouts on a single-trial level and observed their alignment with upcoming turns (Fig. 3b).” However, there is no mention of bouts in the caption of Fig. 3b or the figure itself. It seems that theta power was used instead in Fig. 3b? It seems that the figure corresponding to the bout alignment described in line 124 is missing.

We thank the reviewer for pointing this out. It is correct that the description in line 124 did not accurately match the content of Fig. 3b. We have updated the manuscript to clarify that Fig. 3b shows single-trial amplitudes in the theta range, rather than theta bouts. We have revised the description to reflect this change. Additionally, we now refer to Extended Data Fig. 5b for details on the alignment of theta bout detections with upcoming turns (see response to R2, Major C4A). This figure provides the relevant information on the rate of theta bout detections and their parallels with amplitude dynamics. We have adjusted the manuscript to ensure consistency between the text and figures (pg. 4-5):

“We analyzed the timing of these bouts of increased amplitudes at theta frequencies on a single-trial level and observed their alignment with upcoming turns (Fig. 3b). Strikingly, the percentage of trials in which theta bouts

occurred at specific time points closely mirrored the trial-averaged amplitude dynamics in the theta frequency range (Extended Data Fig. 5b).”

Extended Data Fig. 5 | Frequency spectra and theta dynamics. b. Theta amplitude dynamics and time-resolved theta rates, z-scored for comparison (left), along with theta bout rates (\pm SEM) shown as detection percentages across trials (right) for left (top) and right (bottom) walks. Vertical lines represent turns.

C5: Power spectra and theta dynamics

A) Is the focus on the frequency (3-12 Hz) range justified when considering human data? Looking at Ext. Data Fig. 2a, it seems the authors are excluding the dominant peaks at both ends of the power spectrum, 2 Hz and 16 Hz. Are we maybe missing out on the most interesting signals in this experiment?

While there is an ongoing debate about how to define frequency bands and whether it is useful to do so, a direct translation of frequency band definitions from rodents to humans seems to be inappropriate, independent of these issues. The authors should provide their motivation and also make sure they are not discarding frequencies that are clearly modulated by the task based on arbitrary definitions.

We appreciate the reviewer’s thoughtful comments on the frequency range focus. Based on another reviewer’s comment (R1, C1), we added additional data and results on power fluctuations in the delta and beta frequency ranges. This information is now included in Extended Data Figs. 5, 9, and 10, and discussed in the main text (pg. 4-5):

“In addition to the activities at theta frequencies, we observed two additional spectral peaks in the delta (1-3 Hz) and beta (13-30 Hz) frequency ranges (Extended Data Fig. 5c). We analyzed the activity within these frequency bands using the same approach as for theta, examining amplitude dynamics as a function of position during real-world navigation. The time-varying amplitudes in the delta range showed a dependency on route segments (Extended Data Fig. 10a) comparable to theta, while the beta frequency activity increased during the third route segment (Extended Data Fig. 10c). During this segment, participants performed an additional task of identifying the closest hidden object at a random moment, signaled by an auditory cue.”

Furthermore, we describe the temporal consistency in the delta and beta frequency bands (pg. 6):

“In addition to the theta frequency band (Fig. 3d), temporal consistency across trials was significantly higher during real-world navigation relative to the control condition for the delta (Extended Data Fig. 10b), beta (Extended Data Fig. 10d), and theta (Fig. 3d) frequency bands (all $p < 0.001$). Delta frequency dynamics during imagined navigation were more consistent compared to the control condition ($p = 0.011$, Extended Data Fig. 10b), whereas beta frequencies did not show consistent dynamics ($p = 0.128$, Extended Data Fig. 10d). Furthermore, the effect size (Cohen’s d) of temporal consistency differences between imagined navigation and the control condition was more pronounced in the theta range ($d = 1.76$) than in the delta range ($d = 0.76$).”

We have also added discussion of the delta frequency range findings to the Discussion section (pg. 12):

“Outside the theta range, we observed spectral peaks around 2 Hz in the delta range and approximately 16 Hz in the beta range. Delta-range activities overlap with the “slower theta” rhythms reported in episodic memory tasks^{1,8} and view-based navigation in virtual environments^{3,9}. This overlap may explain the structured dynamics

at 1-3 Hz during both real-world and imagined navigation. However, temporal consistency effects during imagined navigation were more pronounced in the theta range than in the delta range. Beta-range activities may be linked to the self-location sub-task. Increased MTL oscillatory activity in the beta range has been reported during fast walking versus slow walking⁴, while decreased activity in the beta range has been reported during episodic memory retrieval in stationary participants¹. Additionally, a recent study in freely moving rodents suggested that beta activities might result from two independent theta-frequency inputs representing forward and reverse trajectories at roughly opposite theta phases¹⁰. The precise nature of these beta activities remains an open question, with ongoing debate about whether they are related to sensory processing¹, reflect harmonics due to changes in the theta oscillation shapes⁴, or result from two distinct theta-frequency inputs to the hippocampus¹⁰.”

Extended Data Fig. 5 | Frequency spectra and theta dynamics. c. Normalized power spectra (top) and the prevalence of oscillatory bouts (bottom) are shown as averages across participants (group) and individually for each participant (P1-P5).

Extended Data Fig. 10 | **a**, Amplitude dynamics in the delta frequency range are overlaid onto motion trajectories, averaged across participants. **b**, Temporal consistency of delta dynamics was significantly higher during both imagined and real-world navigation compared to sole treadmill walking (control condition). **c**, Amplitude dynamics as a function of position on motion trajectories are shown as averages across participants. Note the increased activity during the third (downward) segment of the routes, where participants estimated and reported their location relative to hidden objects. **d**, Temporal consistency of beta dynamics was significantly higher during real-world navigation compared to both treadmill walking conditions.

Extended Data Fig. 9 | Self-location task. Time-frequency (TF) activity of the self-location task is shown for the group (top left panel) and each participant (average across channels). TF plots are aligned to auditory cues ($t = 0$, black lines), which signal participants to report their location relative to learned positions.

B) How do the here-reported power spectra compare to previous reports? For example, Quasim et al., 2021, Cell, find no peaks in the power spectra of human LFP recordings during virtual navigation (Fig. S1).

We have expanded our discussion to compare our findings with previous reports (pg. 11, also below). The power spectra reported in Quasim et al. (2021) were averaged across many channels and participants, which may have contributed to a lack of visible spectral peaks in their study. In contrast, our analysis reveals frequency-specific peaks that vary across channels. The difference could be attributed to several factors, including our experimental setup, which includes real-world navigation, as well as walking during both imagined navigation and control conditions.

Moreover, as suggested by R3, Major C8 we added this paragraph in the Discussion section (pg. 11):

“Such work has also reported that amplitudes in the theta range increase before and during movement along longer virtual paths^{14,15} but decrease as participants approach goals¹⁶. These findings are consistent with our results, suggesting that participants may have simulated the trajectory ahead of them at movement onset^{15,16} or prior to a turn leading to the next real or imagined route segment in our study. This may be similar to how rodents tile the paths ahead of them^{17–20}, or how humans navigate virtual paths toward distinct goals^{21,22} and experience episodic progression^{23,24}”

C) Do individual participants show a clear theta peak in the power spectrum? Please provide individual power spectra per condition alongside the average power spectra in Ext. Data Fig.2. Please provide the theta peaks for each participant in addition to the theta ranges in Table 1.

Thank you to the reviewer for this suggestion. We have added the individual power spectra for each participant along with the average power spectra, in Extended Data Fig. 5c (top row) as requested:

Extended Data Fig. 5 | Frequency spectra and theta dynamics. c, Normalized power spectra (top) and the prevalence of oscillatory bouts (bottom) are shown as averages across participants (group) and individually for each participant (P1-P5).

Additionally, we have provided the theta peaks for each participant in Extended Data Table 1, alongside the theta ranges:

Participant		1	2	3	4	5
Age [years]		36	45	24	39	30
Gender		male	male	female	male	female
Theta range [Hz]		4.3 - 11.3	3.0 - 9.9	5.65 – 11.3	4.92 – 11.3	3.0 - 9.8
Theta peak [Hz]		8.6 ± 0.3	3.5 ± 0.1	8.6 ± 0.3	8.6 ± 0.3	8.6 ± 0.3
Lead 1	Hemisphere	left	left	left	left	left
	# MTL chan	2	2	1	2	2
	chan 1	HP/HP	HP/PRC	HP/PRC	Amy/HP	HP/HP
	chan 2	HP/Sub	PRC/PHC		HP/HP	HP/HP
Lead 2	Hemisphere	right	right	right	right	right
	# MTL chan	2	1	2	2	2
	chan 3	HP/HP	PHC/Fs	HP/PRC	Tp/PRC	HP/HP
	chan 4	HP/Sub		PRC/PHC	PRC/HP	HP/PHC
Real-world left		36	35	40	34	36
Real-world right		35	34	31	35	35
Imagined left		23	25	24	25	24
Imagined right		20	24	22	24	24
Control		25	24	24	23	24

Extended Data Table 1 | Participant demographics, theta frequency ranges, theta peaks, electrode implantation sites, and the number of trials completed for real-world left and right walks, imagined left and right walks, and treadmill control walks are provided. The theta range was determined using spectral peaks (see Methods) within the 3-12 Hz range, along with neighboring lower and higher spectral minima. The localization of each of the two contacts forming a bipolar channel (chan 1 and 2) is specified for various medial temporal lobe (MTL) regions, including the hippocampus (HP), perirhinal cortex (PRC), parahippocampal cortex (PHC), subiculum (Sub), Fusiform gyrus (Fs), temporal pole (Tp), and amygdala (Amy).

D) The theta peaks seem to settle around 8 Hz in the average spectrum, which could be corresponding to the peak of the dominant EEG alpha rhythm. Did the patients wear surface EEG? If yes, it would be highly informative if the authors provide some basic parameters on the surface EEG data during the navigation conditions: What was each participant's alpha peak and how does it compare to the iEEG theta peaks?

Thank you to the reviewer for the insightful comment. We did indeed capture scalp EEG data alongside our iEEG recordings. However, due to the participants' free movement, the scalp EEG was contaminated with motion and muscular artifacts, and the telemetry required for RNS device access introduced additional noise. To address the reviewer's question, we applied existing correction methods including the Zapline-plus²⁵ approach to mitigate stereotypical technical artifacts and manually marking of high-frequency muscular activity epochs. We focused our analysis on two parieto-occipital channels (PO3 and PO4), where alpha rhythms are typically pronounced, to minimize muscle contamination from neck muscles during walking. We present frequency spectra averaged over these scalp channels and MTL channels for each participant during real-world walking here:

Frequency spectra of intracranial EEG in the MTL and of averaged scalp EEG (PO3, PO4), show minimal correspondence between the peak frequencies observed in intracranial and scalp EEG recordings.

The results showed minimal correspondence between the scalp EEG alpha peaks and iEEG theta peaks. For example, in P1, the scalp EEG showed an alpha peak at 12.1 Hz, while the MTL theta peak was at 8.6 Hz. Similarly, in P3, the scalp alpha peak was at 9.8 Hz, and the MTL theta peak was at 8.6 Hz. These preliminary analyses suggest that the iEEG theta peaks do not directly correspond to the dominant scalp EEG alpha rhythm, as shown in the Figure above and discussed in previous literature, which often distinguishes between theta and alpha rhythms^{5,6}. While this observation is intriguing, we believe that a more thorough investigation into the interactions between scalp and iEEG rhythms is beyond the scope of this manuscript. For this work, we have chosen to focus on the iEEG data, as analyzing mobile scalp EEG robustly would require the development of novel methods, which we plan to address in a future paper.

We added further discussion on the distinction of intracranial MTL rhythms in the theta range and scalp EEG alpha recordings (pg. 12):

“In line with previous literature on hippocampal recordings in humans, we observed spectral peaks at ~4 Hz and ~8 Hz, at the edges of the traditional theta frequency range¹⁻⁴. Consequently, we used individualized theta frequency ranges within the broader 3-12 Hz range. Although these frequencies partly overlap with scalp EEG alpha rhythms, oscillations around ~8 Hz in the MTL are functionally distinct from the alpha activities recorded over visual and sensorimotor areas on the scalp^{7,6}. Future research is needed to explore the relationship between MTL activities in the theta range and scalp recordings.”

E) *The power spectrum shows a peak at 16 Hz that is rather unusual, at least for recordings in non-moving patients. It seems that there is a clear modulation of (real and imagined) navigation compared to the sole treadmill walking in both the power spectrum and prevalence of bouts. What is the authors' take on this clear peak?*

Related to R1, C1 and R2, Major C5A, we have added new data and results showing the relationship of beta range activities during the self-location task, as illustrated in Extended Data Figs. 9 and 10. While this analysis provides insight into beta range activities, it does not fully explain the increased beta power observed during the imagined navigation. We thus have added discussion on these points (pg. 12):

“Beta-range activities may be linked to the self-location sub-task. Increased MTL oscillatory activity in the beta range has been observed during fast walking versus slow walking⁴, while decreased activity in the beta range has been reported during episodic memory retrieval in stationary participants¹. Additionally, a recent study in freely moving rodents suggested that beta activities might result from two independent theta-frequency inputs representing forward and reverse trajectories at roughly opposite theta phases¹⁰. The precise nature of these beta activities remain an open question, with ongoing debate about whether they are related to sensory processing¹, reflect harmonics due to changes in theta oscillation shapes⁴, or result from two distinct theta-frequency inputs to the hippocampus¹⁰.”

F) Ext. Data Fig. 6a: there seem to be regular fluctuations around 2 Hz in theta signal before turns, very clearly in P3 but also P1, P2, P5 and maybe even P4. Are these fluctuations reflected in the low-frequency peak of the power spectrum in Ext. Data Fig. 2? Individual power spectra will allow more insight into this signal. What is the authors' take on these slow rhythms? Are they related to the task?

Thank you to the reviewer for the observation regarding the regular fluctuations around 2 Hz in the theta signal before turns in Extended Data Fig. 6a (now Extended Data Fig. 4a). These fluctuations could be attributed to the varying alignment of theta dynamics across channels within individual participants, where there may be more than one peak per turn (see Fig. 4f and Extended Data Fig. 3). This variability in alignment can create the appearance of regular fluctuations at a lower frequency.

However, to further investigate the slow rhythms, we have included findings related to the delta frequency range (Extended Data Fig. 10a-b) and added individual power spectra as recommended also by Reviewer 1 (see our response to R1, C1). These additions provide a more detailed view of the signal, allowing us to examine whether these slow rhythms are task-related.

Extended Data Fig. 10 | a, Amplitude dynamics in the delta frequency range are overlaid onto motion trajectories, averaged across participants. **b,** Temporal consistency of delta dynamics was significantly higher during both imagined and real-world navigation compared to sole treadmill walking (control condition).

Extended Data Fig. 5 | Frequency spectra and theta dynamics. c, Normalized power spectra (top) and the prevalence of oscillatory bouts (bottom) are shown as averages across participants (group) and individually for each participant (P1-P5).

We added discussion of power spectra and other frequency bands to the manuscript (pg. 4):

“In addition to the activities at theta frequencies, we observed two additional spectral peaks in the delta (1-3 Hz) and beta (13-30 Hz) frequency ranges (Extended Data Fig. 5c). We analyzed the activity within these frequency bands using the same approach as for theta, examining amplitude dynamics as a function of position during real-world navigation. The time-varying amplitudes in the delta range showed a dependency on route segments

(Extended Data Fig. 10a) comparable to theta, while the beta frequency activity increased during the third route segment (Extended Data Fig. 10c). During this segment, participants performed an additional task of identifying the closest hidden object at a random moment, signaled by an auditory cue.

Furthermore, we describe the temporal consistency in the delta and beta frequency bands (pg. 6):

“In addition to the theta frequency band (Fig. 3d), temporal consistency across trials was significantly higher during real-world navigation relative to the control condition for the delta (Extended Data Fig. 10b), beta (Extended Data Fig. 10d), and theta (Fig. 10d) frequency bands (all $p < 0.001$). Delta frequency dynamics during imagined navigation were more consistent compared to the control condition ($p = 0.011$, Extended Data Fig. 10b), whereas beta frequencies did not show consistent dynamics ($p = 0.128$, Extended Data Fig. 10d). Furthermore, the effect size (Cohen’s d) of temporal consistency differences between imagined navigation and the control condition was more pronounced in the theta range ($d = 1.76$) than in the delta range ($d = 0.76$).”

As well as the delta frequency range findings to the manuscript (pg. 12):

“Outside the theta range, we observed spectral peaks around 2 Hz in the delta range and approximately 16 Hz in the beta range. Delta-range activities overlap with the “slower theta” rhythms reported in episodic memory tasks^{1,8} and view-based navigation in virtual environments^{3,9}. This overlap may explain the structured dynamics at 1-3 Hz during both real-world and imagined navigation. However, temporal consistency effects during imagined navigation were more pronounced in the theta range than in the delta range. Beta-range activities may be linked to the self-location sub-task. Increased MTL oscillatory activity in the beta range has been reported during fast walking versus slow walking⁴, while decreased activity in the beta range has been reported during episodic memory retrieval in stationary participants¹. Additionally, a recent study in freely moving rodents suggested that beta activities might result from two independent theta-frequency inputs representing forward and reverse trajectories at roughly opposite theta phases¹⁰. The precise nature of these beta activities remains an open question, with ongoing debate about whether they are related to sensory processing¹, reflect harmonics due to changes in the theta oscillation shapes⁴, or result from two distinct theta-frequency inputs to the hippocampus¹⁰.”

C6: Learning phase of the experiment

A) *The learning phase of the experiment is hard to understand from the description. We would recommend having a supplementary figure that illustrates the learning phase, specifically showing where and how the visual and auditory cues were presented. Further, the authors need to spell out clearly why such a complex learning strategy was chosen, rather than something more straightforward like having visual markers along the path during the learning phase.*

Thank you to the reviewer for the insightful feedback. We have added two supplementary figures to clarify the learning phase. Extended Data Fig. 1 now illustrates the learning phase. Moreover, we included a description of where and how the visual and auditory cues were presented (pg. 13):

“Initially, these route shapes were displayed on a tablet screen, and precise tracking of participants’ walking patterns was achieved through motion capture technology. At the beginning of each recording, participants learned to walk along these patterns without any visible cues indicating the route’s shapes or the positions of the turns. After each walking trial, feedback was provided by overlaying their real movement trajectories onto the prescribed ideal routes (Extended Data Fig. 1a) displayed on a tablet interface, allowing them to improve their behavior in the next trial. This initial learning phase included six life-size paper cut-out objects (three per route) positioned along the path. During the third walking segment of each left or right route (following the second turn), an auditory cue was introduced through a wireless loudspeaker after random delays. Participants were then instructed to determine which object they were closest to by pressing the appropriate left, middle, or right button on a wireless handheld mouse, corresponding to the first, second, or third object on their route. A second auditory cue signaled the accuracy of their response after the button press.”

Additionally, Extended Data Fig. 9 shows the responses to the auditory cues in the self-localization sub-task:

Extended Data Fig. 9 | Self-location task. Time-frequency (TF) activity of the self-location task is shown for the group (top left panel) and each participant (average across channels). TF plots are aligned to auditory cues ($t = 0$, black lines), which signal participants to report their location relative to learned positions.

We chose this approach of using a tablet to present the route rather than visual markers along the path to ensure that participants did not see any visible routes within the room. This design required them to rely on their memory of the routes they had seen on the tablet and to adapt their behavior based on the feedback received after each walk. We believe this method more accurately simulates real-world navigation, where visible markers are often absent, and one must rely on internalized spatial representations.

B) *“Participants were then instructed to determine their relative position to the nearest object by pressing the appropriate left/middle/right button on a wireless handheld mouse, corresponding to the first/second/third object on their route.” It is unclear what “determining their relative position to the nearest object” means here. Were the participants required to simply indicate which of the three objects they were closest to? Or were asked to estimate the distance to the nearest object? And finally, what was the role of this task during learning?*

Participants were instructed to indicate which of the three objects they were closest to by pressing the corresponding left/middle/right button on a wireless handheld mouse (see Methods, pg. 13, and below). They were not required to estimate the distance to the nearest object. The role of this task was to ensure participants maintained awareness of their position during real-world navigation. The primary role of this task was to ensure that participants maintained awareness of their position during real-world navigation. By requiring them to report their position relative to previously learned locations, we aimed to keep their spatial awareness engaged throughout the navigation process.

We have revised the method description as suggested and included additional data in Extended Data Fig. 9, with further details on beta activities during this self-location task shown in Extended Data Fig. 10 (see response to R2, Major C6A). To describe these new findings, we have added the following paragraph to the Results section (pg. 4):

“To ensure that participants maintained awareness of their position during real-world navigation, we instructed them to report their position relative to previously learned locations following the presentation of an auditory cue (see Methods for more details). This self-location task consistently induced activity in the theta and beta frequency ranges across all participants. An initial enhancement of power in the theta band was followed by a suppression of power in theta/alpha frequencies and an enhancement of beta/gamma power during the reporting phase, indicated by the participants via button press (Extended Data Fig. 9).”

Methods (pg. 13):

“Participants were then instructed to determine which object they were closest to by pressing the appropriate left, middle, or right button on a wireless handheld mouse, corresponding to the first, second, or third object on their route.”

C) *What were the exact criteria used to conclude that the route has been learnt by a participant? How was this criteria met for P2 who, as the authors point out, clearly deviated from the route and incorporated an extra turn?*

We appreciate the reviewer's question regarding the criteria for concluding that a route had been learned by participants. This is now described in the manuscript (pg. 13):

“This learning phase concluded once participants demonstrated consistent proficiency in navigating these routes. For most participants, this was evidenced by a gradual reduction in error between the actual and ideal routes over successive trials (Extended Data Fig. 1). However, for P2, after completing eight left and eight right learning trials and despite receiving multiple instructions, it became evident that their walking routes did not improve significantly from trial to trial. Recognizing this, and respecting P2's individual learning process, we instructed them to maintain consistency in their chosen routes going forward, even if these routes incorporated deviations such as the extra turn mentioned. This approach ensured that P2 could still participate in the main experiment while reflecting their unique navigation strategy. At this point, for P2 and after the other participants improved their performance, the objects were then removed, requiring participants to rely on memory for walking routes and object locations. After the learning phase, and during the main experiment, visual feedback was limited and provided only after completing three left and three right walks, serving as periodic indicators of their navigation performance.”

D) *Was data from the RNS system recorded during the learning phase? If so, this would be a very interesting dataset to understand better what may be driving the bouts of theta oscillation before turns during the main experiment. We would suggest an analysis along the following lines:*

- Do similar bouts of theta occur before every turn right from the start of the learning phase?

If so, do these bouts get stronger with learning?

If not, when do they first show up?

- Is there a relation between how well the route is learned and the occurrence of these bursts

The above analysis would allow us to understand if these bouts of theta track common computations required to spatially orient oneself as the body turns or if they in some way act as “landmarks” to routes that are memorized.

We appreciate the reviewer's insightful suggestions and are pleased to report that we had access to data from the RNS system recorded during the learning phase. We have followed the suggested analysis and included the corresponding novel data and results in the manuscript. However, most participants learned the routes quickly, limiting us to a minimum of five left and five right learning walks per participant. The results, now detailed in the ‘Theta dynamics during real-world navigation’ section and illustrated in Extended Data Fig. 1, indicate that the increase in the theta frequency range power is gradually established as participants consistently learn to navigate the instructed routes (pg. 4):

“To investigate whether oscillatory bouts in the theta range were more related to physical turns or specific instances along the routes, we compared the first five walks during and after the experiment's learning phase. After participants learned to walk the routes as instructed, the prevalence of theta bouts preceding turns was significantly higher than during the learning phase ($p = 0.021$, Extended Data Fig. 1c).”

Extended Data Fig. 1 | Learning phase. a, The instructed, ideal route is depicted alongside the actual walks performed by all participants (P1-P5), with earlier and later walks illustrated in light and dark gray colors. **b**, As participants learned to navigate the instructed, ideal routes, they gradually reduced the error between actual and ideal routes within the first five left and right walks. Dots represent each participant's mean error rates (root mean square) depending on the number of walks performed, with earlier and later walks illustrated in light and dark gray, respectively. The behavioral errors during the first five left and right walks of the main experiment are shown for comparison. **c**, A comparison of bouts in the theta frequency range preceding turns showed their prevalence to be significantly higher during the main experiment than during the learning phase. Dots represent the theta bout prevalence at each recording electrode during both the learning phase and the main experiment. The prevalence of the first five left and right walks in the learning phase and main experiment were compared, respectively.

C7: Correlates with eye, head movement:

A) More information is needed on the control conditions the authors are comparing their navigation condition against. Please provide details for the analyses of theta activity concerning eye movements, head rotation, etc.: How was theta activity analyzed in relation to the movements? E.g., locked to onset, offset? How were those detected? Please provide detailed information.

We did not lock theta activity analysis to specific events like movement onset or offset. Instead, we examined the continuous relationship between theta activity and behavioral variables throughout the trial. This approach allowed us to capture the dynamic interplay between theta activity and movement-related changes. In our study, behavioral variables, such as eye movements and head rotation, are continuous signals, recorded with the exact same duration as each trial of the neural data, eliminating the need for additional alignment. To analyze theta activity in relation to these behavioral variables, we computed cross-correlation functions between the neural and behavioral data throughout the trial, as shown in Extended Data, Fig. 7a,c. In cases where we time-warped the neural data, we applied the exact same warping to the behavioral signals to ensure comparability. We added this description in the Method section (pg. 16-17):

“The temporal relationship between behavioral variables and theta dynamics was quantified by calculating cross-correlations between each channel's theta dynamics and the behavioral variables (Extended Data Fig. 10a,c), averaged across trials. All behavioral variables were recorded as continuous signals and were time-warped,

similar to the time-varying amplitudes in the theta frequency range, to allow for comparison of their time courses with the neural data.”

Furthermore, we computed linear regression models for these single-trial behavioral variables and theta dynamics and tested the effects at the group level, as shown in Extended Data, Fig. 7b,d. We have expanded the description of this statistical assessment in the Methods section (pg. 20):

“Additionally, we computed linear regression models of theta dynamics and behavioral variables (Extended Data Fig 7b,d) on a single trial level for each recording electrode. Subsequently, we tested the effects at the group level and compared the resulting regression coefficients using multi-level block permutation tests while controlling for multiple comparisons^{26,27}.”

Extended Data Fig. 7 | Relation of theta dynamics to behavioral variables. **a**, Cross-correlation of theta dynamics and behavioral variables during real-world navigation. Colored lines illustrate the group average, while grey lines indicate standard errors across the folds of cross-validation for each behavioral variable (position [pos], eye speed, head rotation, hip rotation, and movement speed [v]). Negative lags represent instances where theta dynamics preceded the behavioral signals. The top left panel summarizes the temporal relationships, revealing that theta activities peaked approximately 1 second before physical turns, followed by head rotation, eye saccades, speed reductions and hip rotation. The positive lagged correlation between theta and movement speed reflects the speed decrease before turns, which is then followed by a reduction in theta activity. **b**, Regression coefficients of ongoing (single-trial) theta dynamics during real-world navigation, with labeled behavioral variables. **c**, Cross-correlation of theta dynamics and behavioral variables during imagined navigation, analogous to **(a)**. Theta dynamics are compared to behavioral variables after alignment to enable comparisons with position within the route segments. **d**, Regression coefficients for ongoing (single-trial) theta dynamics during imagined navigation, similar to **(b)**.

B) l. 174: “However, we did not find a direct effect of eye movements on theta amplitudes during imaginations ($p = 0.58$). Instead, we observed that theta dynamics were primarily related to the relative position within the route

segments ($p = 0.006$), mirroring the patterns observed during real-world navigation.” Since crucial information on the movement analyses are missing, I do not see how the authors’ conclusion that “eye movements did not drive the effect” is justified.

We appreciate the reviewer’s concern regarding the movement analyses. We acknowledge that our statement regarding the influence of eye movements may have suggested a more definitive conclusion than intended. As our observations are correlational, we have toned down the language to avoid implying the presence or absence of specific drivers for the observed effects. Our findings indicated that theta dynamics were primarily related to the relative position within the route segments, mirroring the patterns observed during real-world navigation. The lack of direct effect of eye movements on theta amplitudes suggests that eye movements did not have a significant impact on theta activity in this context, but we recognize that these results are correlational and do not fully establish causation. We have revised our manuscript to clarify these points (pg. 6) and Extended Data Fig. 7 (see above in response to R2, Major C7A):

“To validate participants’ engagement in imagining the distinct navigational routes, we examined their gaze. During real-world navigation, participants tended to focus their gaze ahead on the walking path, resulting in relatively higher probabilities of left or right gazes for leftward and rightward walks, respectively. While this effect was less pronounced during imagined navigations, gaze positions still subtly differentiated between imagined left and right routes (Extended Data Fig. 8a). However, we did not find a direct effect of eye velocity on theta amplitudes during imaginations ($p = 0.58$), nor evidence for increased theta amplitudes time-locked to saccades during either imagined navigation or control trials (Extended Data Fig. 8d,e). Instead, we observed that theta dynamics were primarily related to the relative position within the route segments ($p = 0.006$), mirroring the patterns observed during real-world navigation. The impact of relative position during imagined navigation was significantly greater than that of eye movements ($p = 0.003$), suggesting that eye movements did not significantly impact theta dynamics (Extended Data Fig. 7c,d, 8b-e).”

Moreover, we investigated the influence of saccades separately in Extended Data Fig. 8b-e and in response to R2, Major C7C below) and clarified the associated methods (pg. 16):

“To explore the potential influence of eye movements on MTL recordings, we computed average time-frequency activities aligned with the onset of eye saccades and compared these plots with those aligned with turns (Extended Data Fig. 8b-e). Four saccades (Extended Data Fig. 8c) were randomly selected per trial to correspond with the four turns (shown in Extended Data Fig. 8b). Additionally, we ensured that each fixation preceding a saccade lasted longer than 600 milliseconds, as described in previous work²⁷. This approach allowed us to match the number of observations across conditions precisely, as depicted in Extended Fig. 8b-e.”

Extended Data Fig. 8 | Gaze densities and MTL activity aligned to turns and saccades. b-c, Time-frequency (TF) activity aligned to turns (**b**) and saccades (**c**) during real-world navigation. **d-e**, Time-frequency activity aligned to eye movements (saccades) during imagined navigation (**d**) and the control condition of treadmill walking without imagination (**e**). Note pronounced theta activities precedes physical turns but not saccades.

C) Previous studies reported saccade-triggered phase theta resets in human and monkey hippocampus. Can the authors replicate those findings? The way the results are described at the moment reads as if saccades generally had no impact on theta activity.

We thank the reviewer for pointing out the relevance of previous studies on saccade-triggered phase resets and their impact on theta activity. In response, we conducted phase reset analyses following the methodology described by Jutras et al. (2013)²⁸. Specifically, we identified saccades that followed fixations lasting at least 600 milliseconds, filtered the neural data in the theta frequency range, and aligned single trials to saccade onsets before averaging across all saccades. Our analysis aimed to detect any increased phase alignment and

amplitude changes indicative of inter-trial phase coherence (ITC) if saccades triggered phase resets. Despite applying these methods, we did not find credible evidence for differences in theta amplitudes between the pre- and post-saccade periods, specifically 600 milliseconds before and 600 milliseconds after a 400-millisecond buffer period immediately following saccade onset:

Analyses related to potential saccade-triggered phase resets. We did not find credible evidence for differences in amplitudes between pre- and post-saccade periods across any of the experimental conditions (top row). To further confirm these findings, we compared the inter-trial coherence (ITC) of pre- and post-saccade periods (bottom row). Each dot represents data from one MTL channel.

These findings suggest that, in our study, saccades did not have a significant impact on theta activity in the manner reported by previous research. We acknowledge that our experimental setup differed from those in earlier studies, including the absence of a fixation cross, which might contribute to the observed discrepancies. Nonetheless, our results indicate that saccades did not generally impact theta dynamics as described in the literature.

C8: Discussion

It seems that parts of the relevant literature that investigated virtual navigation using human intracranial data are overlooked, in particular those that talk about theta oscillations. To name a few, in no particular order: Quasim et al., 2021, Cell; Chen et al., 2018, Curr Biol; Kunz et al., 2019, Sci Adv; Maidenbaum et al., 2018, PNAS; Liu et al., 2023, Curr Biol; It would be great to read a more extensive discussion on how the current novel findings in real-world navigation relate to the previous virtual navigation studies.

Following the reviewer's suggestion, we added this paragraph to the Discussion section (pg. 11):

“Such work has also reported that amplitudes in the theta range increase before and during movement along longer virtual paths^{14,15} but decrease as participants approach goals¹⁶. These findings are consistent with our results, suggesting that participants may have simulated the trajectory ahead of them at movement onset^{15,16} or prior to a turn leading to the next real or imagined route segment in our study. This may be similar to how rodents tile paths ahead of them^{17–20}, and how humans navigate virtual paths toward distinct goals^{16,22}, or experience episodic progression^{23,24}.”

Minor comments:

C1: *I. 758: “Triangles indicate that peak theta activity precedes turns in 17 out of 18 channels.” Did the one channel not show a peak or was it dropped out for other reasons? Please add information to the caption.*

Thank you to the reviewer for pointing this out. The one channel in question did show a peak in theta activity, but unlike the others, this peak occurred after the turn rather than before it. We have now clarified this in the figure caption to reflect that all 18 channels exhibited peak theta activity, with 17 channels showing this peak preceding the turn and 1 channel showing it afterward (Extended Data Fig. 6c caption):

“Triangles indicate that peak theta activity precedes turns in 17 out of 18 channels, with 1 triangle indicating a peak occurring after the turn.”

C2: *Please indicate the participant number of the participant shown in movie 1.*

We have now included the participant number in the caption of Extended Data Movie 1.

Extended Data Movie 1 | Synchronized intracranial EEG (iEEG) activity, behavior, and motion capture during real-world spatial navigation. (Top left) Motion camera capturing the movement of an exemplary participant (P4) while approaching turns aligned with simultaneously recorded iEEG (Bottom) in real-time. (Top middle) Participant’s synchronized movement trajectory is depicted as a white line. (Top right) Theta power (4-12 Hz) derived from an exemplary hippocampal channel during the illustrated behavior in the same participant. Here, we show an exemplary participant during real-world navigation for illustrative purposes (consent to publish obtained).

C3: *Please indicate whether participants contributed to previous studies and provide information about these studies for each individual participant.*

Participants P1 and P4 have contributed to previous studies. Specifically, P1 participated in Stangl et al. (2021)¹³, and P4 was involved in Maoz et al. (2023)²⁹. We have now included this information, along with relevant details about these studies, in the manuscript (Methods, ‘Participants’ section, pg. 13):

“Notably, participants P1 and P4 have contributed to previous studies, with P1 participating in Stangl et al. (2021)¹³ and P4 involved in Maoz et al. (2023)²⁹.”

C4: *Were the 5 patients included in the study the only five from which the data was collected for this experiment?*

Yes, we can confirm that the five participants included in the study were the only individuals from whom intracranial EEG data were collected for this experiment. Prior to data collection, we tested the paradigm with lab members who did not have RNS implants to ensure the procedure and setup were optimized.

C5: *Line 158: Please provide the correlation coefficients for each condition.*

We have added the correlation coefficients for each condition to the manuscript, as requested (pg. 6):

“Our analysis showed that the temporal consistency across trials (Fig. 3b, Extended Data Fig. 4b) was significantly higher ($p < 0.001$) for theta dynamics during imagined navigation ($r = 0.12 \pm 0.04$) compared to the control condition ($r = -0.03 \pm 0.07$) of sole treadmill walking (Fig. 3d). The highest level of temporal consistency was observed during real-world navigation ($r = 0.23 \pm 0.06$)...”

C6: *I. 169: “As a means to validate participants’ engagement in imagining the distinct navigational routes, we examined their eye movements.” We don’t agree with the end of this statement, since the authors depict gaze in ext. data fig. 7, not (eye) movements. Please adjust.*

We agree and revised the wording as suggested (pg. 6):

“To validate participants’ engagement in imagining the distinct navigational routes, we examined their gaze.”

Reviewer #3 [R3]: *In their article “Human neural dynamics of real-world and imagined navigation,” Seeber et al. use intracranial EEG to investigate neural activity in the theta-frequency range in freely moving humans (n = 5). The key findings are that theta oscillations occur in short bouts during real-world navigation; that theta power increases before turns; that theta power is temporally stable across trials; and that theta power is similar between actual navigation and remembered (“imagined”) navigation. This is a very interesting and timely manuscript that sheds new light on the question of what kind of information neural activity in the theta-frequency range may represent. My specific comments are listed below.*

Thank you to the reviewer for their positive and encouraging feedback. Below, we address each specific comment in detail.

Major comments:

C1: *It is often difficult to distinguish between imagination and memory retrieval at a conceptual level and in my view this difficulty also applies to this study. As the patients are not asked to imagine new routes that they have never experienced before, I find it more appropriate to interpret the results in the context of memory retrieval instead of imagination. I would thus suggest that the authors tone down their statements about imagination, including the title and the abstract.*

Thank you to the reviewer for this insightful comment. We recognize the complexity of distinguishing between imagination and memory retrieval, especially in the context of our study. Our experiment specifically included a condition where participants were asked to imagine their future route, which we believe aligns with the concept of imagination as it pertains to future or hypothetical scenarios^{30,31}. While we agree that imagination often intertwines with previous experiences, we argue that our study still addresses imagination in this context. Given the difficulty in fully separating the intertwined concepts of memory and imagination, we have added the following discussion to our manuscript (pg. 11):

“Imaginations are often intertwined with previous experience, making it challenging to fully separate processes related to imagination and memory retrieval. While our experiment specifically included a condition where participants were asked to imagine navigating future routes, these trajectories were previously experienced and were not novel. Nevertheless, we interpret and discuss our findings as imagined navigations, as their mental progressions along routes pertain to future or hypothetical scenarios^{31–33}.”

In light of these points, we believe that the term “imagined navigation” more accurately reflects the scope and findings of our study than “memory retrieval”. Therefore, we propose keeping the title and abstract as originally proposed. However, we have revised the manuscript to better clarify the relationship between memory retrieval and imagination, ensuring that our interpretation remains consistent and clear.

C2: *Similarly, I think that the interpretation of the results in the light of episodic memory (for example, on page 2) is somewhat exaggerated as participants were asked to navigate along predefined routes and to remember these routes. Episodic memories are often defined as unique memories for what happened where and when, and I thus do not think that the authors’ task fulfills the requirements of an episodic memory task. I would suggest toning down the statements on episodic memory and being more specific to the findings at hand.*

Thank you to the reviewer for their thoughtful feedback and we appreciate the concern regarding the interpretation of our findings in relation to episodic memory. While our study does involve participants navigating and remembering predefined routes, we understand that this may not fully capture the unique and personal nature of episodic memories, which typically involve recalling specific events with contextual details of what happened, where, and when. We have revised our manuscript to tone down the claims regarding episodic memory and have focused more specifically on the spatial navigation aspects of our findings (e.g., pg. 2):

“Identifying such mechanisms would provide important insights into the shared neural organization principles between real-world spatial navigation and memory processes, effectively bridging decades of findings across species and integrating research on spatial navigation and memory.”

C3: *The authors state that their results show that theta dynamics encoded spatial geometry. It is not clear to me how different theta-power levels on different parts of the track allow for the conclusion that theta dynamics encode the spatial geometry of those routes. Please clarify.*

We agree with the reviewer’s concern regarding the use of the term “spatial geometry.” To address this, we have removed the term and replaced it with terms such as “spatial information” or “spatial structure” throughout the

manuscript. This change more accurately reflects our findings, which demonstrate that theta dynamics are associated with different parts of the navigational route, rather than specifically encoding spatial geometry.

C4: *The authors mention a computational model that generalizes from real-world to imagined navigation and that successfully reconstructs participants' imagined positions, but I could not find such a computational model in the manuscript (I wouldn't consider a statistical model for an analysis a computational model). Please clarify.*

We thank the reviewer for this comment and agree that the term "computational model" might have been misleading in the original text. To clarify, the model we referenced is indeed a statistical model designed for the analysis of the data rather than a computational model in the traditional sense. We have revised the manuscript to reflect this distinction more accurately and removed any mention of a "computational model" to avoid confusion.

C5: *As the sample size is small (5 patients; 18 electrodes) it would be helpful to see theta power as a function of linearized position on the two routes for all electrodes, averaged across trials, separately for the different conditions.*

We appreciate the reviewer's suggestion to visualize theta power as a function of linearized position across all electrodes. For the real-world navigation condition, we have included the requested analysis, showing theta amplitudes as a function of linearized position for each electrode, averaged across trials:

Theta amplitudes during real-world navigation as a function of linearized position. Data from all 18 MTL electrodes are shown for the left (top panel) and right (bottom panel) routes during real-world navigation. Each row represents data from one MTL channel, with yellow and blue colors indicating relative amplitude increases and decreases, respectively. Black vertical lines mark the turns in the routes.

However, for the imagined navigation condition, the positions are reconstructed from pooled channels, and linearized position data is only available at the group level. Consequently, individual electrode data in the imagined condition is organized by time rather than position. This discrepancy in data structure prevents a direct comparison of theta power across conditions at the individual electrode level.

To address this, we include a comparison of theta power as a function of linearized position for the real-world condition (Figure above) and a group-level analysis for the imagined condition, which includes both the cosine and -sine components of the position reconstruction model:

Theta amplitudes during imagined navigation as a function of reconstructed position. Group-level results for both components of the position reconstruction model are shown. The reconstruction of the cosine and $-\sin$ route representations is illustrated in the left and right panels, respectively. Red/yellow colors indicate relative amplitude increase, while blue colors represent decreases.

These additions highlight the similarities and differences between conditions, as shown in Fig. 4d and g, and address the reviewer's request while acknowledging the limitations of direct electrode-level comparisons for the imagined condition.

C6: *The temporal consistency analyses (page 6) are useful regarding the question whether the observed neural activity is stable across trials. However, in my view, they do not allow for conclusions that comparable functional networks are involved in real-world and remembered navigation. They could, for example, be a simple side effect of the prevalence of theta oscillations in the channel-wise data (amongst many other basic properties of the channel-wise data). Please clarify or reword.*

Thank you to the reviewer for this insightful comment. We recognize the concern that the temporal consistency analyses might be influenced by basic channel-wise properties, such as the prevalence of theta oscillations. To address this, we conducted additional analyses to assess whether these basic properties could account for the observed results. Firstly, we tested for differences in the prevalence of theta bouts between the imagined condition and the control condition and found no significant difference ($p = 0.47$). This suggests that the observed temporal consistency is not simply due to differences in theta bout prevalence. Secondly, we examined whether basic channel-wise properties might influence the observed temporal consistency, by correlating the temporal consistency of control data with that of real-world navigation data. We found no significant correlation ($r = 0.11$, $p = 0.22$), as shown in the Figure below, suggesting that these basic properties do not account for the spatial similarity observed between real and imagined navigation. These additional analyses support the robustness of our temporal consistency findings and suggest they are not merely a byproduct of basic channel-wise properties.

Comparison of temporal consistency between experimental conditions. A significant correlation was observed between the two navigational conditions ($p = 5.5e-3$, left panel), whereas no significant correlation was found in the control condition ($p = 0.22$, right panel).

C7: *While walking on the treadmill, participants could view parts of the real-world navigation routes. Furthermore, the authors show that participants' eye movements during treadmill walking resemble their eye movements during real-world navigation. Could it thus be that the participants are viewing their remembered navigation routes while walking on the treadmill? If so, viewing the routes could drive the temporal consistency of*

remembered navigation routes, and their similarity to real-world navigation. Two analyses to see whether this was indeed the case would be (1) to quantify how much participants looked at the remembered navigation routes while walking on the treadmill; (2) to test whether temporal stability and similarity between real-world and remembered navigation is driven by the first half of the route that participants were able to view during treadmill walking.

We appreciate the reviewer's thoughtful suggestion regarding the potential influence of viewing remembered navigation routes during treadmill walking on temporal consistency. To address this concern, we conducted additional analyses as suggested.

First, as suggested, we quantified the gaze differences between right and left imagination trials separately for the first and second half of the imagination trials. Given the gaze coordinates in the visual field, gaze position values are relatively larger in the right hemifield and smaller in the left hemifield. Participants tended to direct their gaze more to the left when imagining left routes and to the right when imagining right routes consistently for both halves of the imagined routes as shown in the below Figure's left panel:

Participants biased their gaze (mean \pm std, one dot per participant) similarly in the first and second halves of their imaginations (left panel). We did not find significant left or right lateralization when comparing gaze positions between the first and second halves of the imagination task (middle panel). Additionally, there was no credible evidence ($p = 0.40$) for differences in the temporal consistency of theta dynamics between the first and second halves of imagined navigation (right panel, each dot represents an MTL electrode).

Further, if participants had been viewing their remembered navigation routes in the first half of their imaginations, we would expect to see consistently negative or positive gaze differences when comparing the first and second halves of imagined routes, depending on the direction of the imagined route. However, this was not observed (mid-panel).

Second, we compared the temporal consistency between the first and second halves of the imagined navigation trials to determine if viewing the routes during treadmill walking influenced the results. We found no significant difference in temporal consistency between the first and second halves of the imagination trials ($p = 0.40$, right panel in the Figure above).

This suggests that the observed temporal stability and similarity between real-world and imagined navigation are not driven by viewing the remembered routes during treadmill walking. These analyses support the conclusion that participants' gaze behavior during treadmill walking did not significantly impact the temporal consistency or the observed similarity between real-world and imagined navigation.

C8: *The section on reconstructing imagined positions seems innovative and interesting overall, but I also find it hard to follow. For example, the authors state that it was evident that theta amplitudes were consistently modulated across the segments. I believe that this shall mean that theta power was similar at corresponding relative positions on each segment. Could the authors provide some empirical evidence for this claim, for example by correlating theta power from different segments with each other?*

Thank you to the reviewer for this valuable feedback. To clarify our approach and provide empirical evidence for our claim that theta amplitudes were consistently modulated across the segments, we have added a detailed explanation and reformulated our description (pg. 8):

“After adjusting for variations in segment lengths, we found that theta range amplitudes were consistently similar at corresponding relative positions within each segment (Fig. 4e).”

We have also updated the Methods section under “iEEG Data Analyses” (pg. 16):

“To analyze the similarity of theta dynamics within each segment of a walking route, we abstracted the navigational task structure into a sinusoidal pattern that peaked at each turn. This approach ensured that each segment adhered to the same sinusoidal pattern, scaled according to its normalized length or relative time. The phase of this sinusoidal pattern corresponded to the relative position within each segment. For both imagined navigation and control walks, we first normalized segment lengths to the mean length of the real-world walking segments. This normalization process ensured that all segments in our analysis were of equal duration, regardless of their original physical length. We then computed the cross-correlation between theta dynamics (reconstruction output) and the sinusoidal model of each walking segment. The central peak of the resulting cross-correlogram represents the correlation between theta dynamics and the sinusoidal task structure, effectively capturing the spatial layout and timing across each walking route. Additionally, any side peaks in the cross-correlogram indicate similarities between theta dynamics and shifted task structures (note increases at multiples of a segment), further confirming the consistency of theta activity patterns relative to the route’s structure.”

C9: *Could the authors provide more detailed information on the 10 x 10 cross-validation procedure? Otherwise, it is difficult to evaluate the correctness of this approach.*

We have expanded the description of the 10 x 10 cross-validation procedure in the Methods section of the revised manuscript (pg. 17-18):

“To prevent overfitting, we employed 10-fold cross-validation repeated 10 times to learn the temporal relationship from sub-partitions of the data. Each participant’s single trials were randomly grouped into ten subsets, each containing roughly 10% of the unique trials. From these subsets, nine (~90% of the trials) were used to train the alignment of time and position during imagined navigation. The trained alignment was then tested on the remaining subset of previously unseen imagination trials, serving as a test of the alignment algorithm’s generalizability. This process was repeated for all ten combinations of training and test sets. Additionally, the random grouping of subsets was repeated, and the training and testing procedure was applied across ten repetitions, resulting in a total of 100 iterations. Each iteration produced position estimates computed from the test set trials, independent of the subset used for training the alignment. The outcomes of these generalizations are presented throughout the manuscript.”

C10: *The time warping procedure is essential for this analysis, but I could not find a clear description of this procedure in the manuscript. Could you provide more details? How do you account for the fact that the time warping may create false positives by warping the data in a way that resembles theta activity during real-world navigation but has nothing to do with the participants’ remembered navigation routes? In other words, what is your statistical baseline to evaluate your results in Fig. 4D, middle, against? I think it’s not sufficient to compare them against the control periods.*

We thank the reviewer for bringing up this important point regarding the time-warping procedure. We acknowledge that a more detailed description of the procedure is necessary, and we have expanded this explanation in the Methods section and introduced a novel section, “Reconstructing imagined positions” for better visibility (pg. 17):

“We utilized the position estimation model developed from our real-world walking data to derive an initial estimate of imagined positions. Subsequently, we applied dynamic time warping to align these estimated relative positions with the task structure, allowing for a maximal time shift of ± 2 seconds. This alignment procedure was conducted for each participant while preserving the relative timing between recording channels within each participant. To prevent overfitting, we employed 10-fold cross-validation repeated 10 times to learn the temporal relationship from sub-partitions of the data.”

Additionally, we want to emphasize that all results involving the alignment procedure were cross-validated. The results presented in the manuscript are derived from parts of the data that were not seen by the alignment procedure, ensuring that the findings generalize beyond the data used to identify time offsets for alignment across trials and navigation conditions.

To address the concern about the potential for false positives due to the time-warping process, we initially compared our results against a null distribution created by circularly shifting the data. This approach allowed us to assess whether the observed temporal patterns were significantly different from what could be expected by

chance. We also compared these results against the control condition, which we considered a more conservative baseline.

However, following the reviewer's suggestion, we recognized the need to shift the data, rather than scramble it, to ensure a more accurate comparison. Scrambling would shuffle all samples, distorting the temporal relation of neighboring samples. By shifting the data we can simulate random temporal relations between the compared conditions (e.g. real-world and imagined navigation) while maintaining the temporal correlation of neighboring samples, leading to a more appropriate and stringent permutation distribution. We have modified the analyses accordingly and updated the p-values in Fig. 4h and the associated text (pg. 19-20):

"We also compared the reconstruction errors of each condition to a null distribution generated by randomly shifting the reconstructed and modeled positions 10,000 times. This approach maintained the temporal correlation of neighboring samples while simulating random temporal relations between conditions. By doing so, we could more accurately assess the significance of our reconstruction errors and ensure a rigorous comparison."

C11: *The authors state that "theta dynamics exhibited a significant correlation between real-world and imagined navigation," but they don't show any plots in support of this statement. To me this result seems to be the strongest evidence that there is shared neural activity between real-world navigation and remembered ("imagined") navigation. Could you thus please show scatter plots etc. to demonstrate the robustness of this result and whether it is mainly driven by particular segments and/or positions on the segments?*

Thank you to the reviewer for highlighting this important aspect of our analysis. We have included a scatter plot in Fig. 4d, which illustrates the correlation between actual and estimated positions for both real-world and imagined navigation.

Fig. 4 | Reconstructing relative position from neural dynamics. d, Scatterplot of the reconstructed route representations of real-world and imagined navigation. Each dot represents data averaged over 0.5-second time bins.

Additionally, we have analyzed the accuracy of position reconstruction across different segments and positions. For real-world navigation, we found a consistent correspondence between actual and estimated positions throughout the entire segment. However, during imagined navigation, reconstruction accuracy was higher at the beginning and end of each route segment (see Fig. 4g, middle panel). This variation result from the alignment process at these critical points or error accumulation that resets at salient points along the route³⁴.

Fig. 4 | Reconstructing relative position from neural dynamics. **g**, Reconstruction of relative segment positions from theta dynamics depicted using 2D histograms that show both the actual and estimated positions. Color-coded representations illustrate the probabilities of all possible combinations of actual and estimated positions. An ideal reconstruction outcome would manifest as a diagonal pattern. During real-world navigation, the estimated positions (cross-validated) clustered closely around the actual physical positions within each segment (left panel). During imagined navigation, the estimated positions also aligned consistently with the positions estimated from the duration of the imagination periods particularly at the beginning and end of each route segment (mid-panel), illustrating heightened accuracy in those instances. However, during sole treadmill walking where imagined navigation was absent, reconstruction failed to yield accurate results (right panel). **h**, Histograms depicting the errors in reconstruction (measured in degrees as a circular variable) for each condition revealed distinct patterns. In both real-world (left panel) and imagined (middle panel) navigation conditions, the errors clustered around zero, indicating accurate reconstruction. However, during sole treadmill walking (right panel), this clustering around zero was absent, signifying less accurate reconstruction.

The fact that reconstruction errors were significantly lower in the imagined navigation condition than in the control condition strongly supports the conclusion that the observed temporal consistency is not merely an artifact of the alignment process. This difference in reconstruction accuracy indicates that the imagined navigation trials contain meaningful temporal dynamics that are more consistent with real-world navigation, beyond what would be expected from chance alignment alone. For additional details, we refer to our response to R2, Major C2C.

C12: *Fig. 4, B vs. C: How can these two plots be consistent with each other if B shows a match with a cosine structure of each path segment, and C suggests a sine structure for some of the electrodes?*

Thank you to the reviewer for pointing out this issue. The discrepancy between Fig. 4b (now moved to Fig. 4e) and Fig. 4c (now moved to Fig. 4f) resulted from correlating the sine components (model and reconstruction) but illustrating the cosine model in Fig. 4a. To clarify this, we have revised Fig. 4 to display the negative sine (–sine) component of the route representation (model) in Fig. 4a and its corresponding reconstruction in Fig. 4c. We have also updated the figure caption to provide a clearer explanation of the relationship between these components and ensure consistency between panels a-c, and e:

Fig. 4 | Reconstructing relative position from neural dynamics. **a**, The model of the navigational route structure is represented as a -sine pattern (model) peaking before turns. **b**, Reconstruction of this (-sine) route representation from theta dynamics during real-world navigation. **c**, Reconstruction of the (-sine) route representation from theta dynamics during imagined navigation. **d**, Scatterplot of the reconstructed route representations of real-world and imagined navigation. Each dot represents data averaged over 0.5-second time bins. **e**, Cross-correlation analysis between theta dynamics and the route structure (adjusted for segment length) revealed a significant correlation (lag = 0) for both real-world and imagined navigation, while sole treadmill walking did not exhibit such a correlation. The similarity and repetition of theta dynamics in each route segment led to four lateral peaks in the cross-correlations at lags that matched the length of one segment on a walking route. **f**, Theta dynamics are illustrated as a function of relative position within all segments across all 18 electrode channels. Notably, the timing of theta dynamics varied across channels, roughly following a cosine or -sine wave pattern (white lines). These two orthogonal theta modulations effectively encoded relative position as a circular variable.

C13: Fig. 4B: An alternative to the sinusoidal modulation of theta power would be a boxcar-shaped modulation (low power in the middle of the segment; high power at its ends). What does the cross-correlation look like when using this alternative boxcar-shaped modulation? Does the sinusoidal modulation provide a significantly better fit to the data? In other words, to what extent is the apparent cosine modulation simply driven by increased theta power shortly before the turns?

We thank the reviewer for suggesting the use of a boxcar-shaped modulation as an alternative to the sinusoidal modulation of theta power in Fig. 4e (previously Fig. 4b). In our analysis, we model linear position as a circular variable following the approach of Agarwal et al., (2015)¹², where each full circle on the ring corresponds to the traversal of a segment. This sinusoidal modulation effectively captures the continuous nature of theta power changes throughout the segment, including before and after turns, which we now further describe in the manuscript (pg. 8):

“Modeling the linearized position as a circular variable is a useful mathematical abstraction where movement can be thought of as a change of position on a ring. In our case, each full circle on that ring would correspond to the traversal of one route segment. This approach allows us to capture the continuous, cyclical modulation of theta activity across route segments, which can vary smoothly over time. By using sine and cosine functions to model theta dynamics, we account for these cyclical changes more effectively.”

In contrast, a boxcar model can only detect whether a participant is within a specific section of the segment, such as the beginning or end, without capturing the gradual variations in theta power across the entire segment:

Cross-correlation of theta dynamics using a boxcar model, adjusted for segment length discrepancies. This boxcar model (left panel) correlated less with the data than the circular (cosine/sine) model in the main manuscript.

We applied the boxcar model as suggested and found that it correlated less with the data compared to the sinusoidal model. This suggests that the sinusoidal modulation provides a significantly better fit to the data, likely due to its ability to account for the continuous and periodic nature of theta power modulation, including the increases observed shortly before turns.

Minor comments:

C1: *Fig. 1B: It wasn't clear to why all control trials on the treadmill happened before the first real-world navigation walk. Why weren't these control trials interspersed with the other trial types (real-world navigation and remembered navigation)?*

The real-world and treadmill walking trials were indeed interspersed. However, we decided that it was crucial to conduct all control trials on the treadmill before introducing the concept of imagined navigation. This approach ensured that participants remained naïve to the imagination task during the control condition, preventing them from inadvertently imagining walking the routes during these trials.

We also direct the reviewer to a similar comment from Reviewer 2, along with additional results provided in our response, which suggest that there is no evidence of a time-on-task effect influencing the outcomes of these trials (see response to R2, Major C2D).

C2: *Fig. 2B: Is this single-trial data or averaged across trials? Please specify in the legend.*

We have clarified in the caption of Fig. 2b that the panel illustrates theta amplitudes 'averaged across trials' as suggested.

Fig. 2 | Theta dynamics during real-world navigation. b, Time-frequency plots of exemplary hippocampal activity averaged across trials reveal task-related and temporally organized theta oscillations during both left and right walks (top and bottom panels, respectively). Black vertical lines demarcate where turns occurred in the routes.

C3: *Fig. 2B: What is the consistent neural activity at around 16 Hz?*

We added novel data and results regarding activities in the beta range in Extended Data Fig. 9 and 10 (below), where we investigated these activities further. Also see responses to Reviewer 1, C1 and Reviewer 2, Major C5A,E.

Specifically, we expanded the results to include other spectral peaks, including beta frequencies (13-30 Hz) (pg. 4-5):

“In addition to the activities at theta frequencies, we observed two additional spectral peaks in the delta (1-3 Hz) and beta (13-30 Hz) frequency ranges (Extended Data Fig. 5c). We analyzed the activity within these frequency bands using the same approach as for theta, examining amplitude dynamics as a function of position during real-world navigation. The time-varying amplitudes in the delta range showed a dependency on route segments (Extended Data Fig. 10a) comparable to theta, while the beta frequency activity increased during the third route segment (Extended Data Fig. 10c). During this segment, participants performed an additional task of identifying the closest hidden object at a random moment, signaled by an auditory cue.”

Furthermore, we describe the temporal consistency in beta frequency band (pg. 6):

“In addition to the theta frequency band (Fig. 3d), temporal consistency across trials was significantly higher during real-world navigation compared to the control condition for the delta (Extended Data Fig. 10b) and beta (Extended Data Fig. 10d) frequency bands (all $p < 0.001$). Delta frequency dynamics during imagined navigation were more consistent compared to the control condition ($p = 0.011$, Extended Data Fig. 10b), whereas beta frequencies did not show consistent dynamics ($p = 0.128$, Extended Data Fig. 10d). Furthermore, the effect size (Cohen’s d) of temporal consistency differences between imagined navigation and the control condition was more pronounced in the theta range ($d = 1.76$) than in the delta range ($d = 0.76$).”

Extended Data Fig. 10 | **a**, Amplitude dynamics in the delta frequency range are overlaid onto motion trajectories, averaged across participants. **b**, Temporal consistency of delta dynamics was significantly higher during both imagined and real-world navigation compared to sole treadmill walking (control condition). **c**, Amplitude dynamics as a function of position on motion trajectories are shown as averages across participants. Note the increased activity during the third (downward) segment of the routes, where participants estimated and reported their location relative to hidden objects. **d**, Temporal consistency of beta dynamics was significantly higher during real-world navigation compared to both treadmill walking conditions.

Extended Data Fig. 9 | **Self-location task**. Time-frequency (TF) activity of the self-location task is shown for the group (top left panel) and each participant (average across channels). TF plots are aligned to auditory cues ($t = 0$, black lines), which signal participants to report their location relative to learned positions.

C4: Fig. 2C: How many channels contribute to this plot? How does this pattern look like for electrodes in the left posterior hippocampus, right anterior hippocampus, and right posterior hippocampus?

As indicated in the caption, Fig. 2c includes data from one channel per participant in the left anterior hippocampus, with a total of five channels contributing to this plot. Regarding the anatomical regions mentioned, we only have consistent electrode coverage in the anterior hippocampal area across all participants. Unfortunately, data from electrodes in the left posterior hippocampus, right anterior hippocampus, and right posterior hippocampus are not available for all participants due to variability in electrode placement, which was determined based on clinical treatment criteria. To address this limitation, we have added a paragraph to the Discussion section (pg. 11):

“Although we ensured coverage of MTL regions during the participant recruitment, electrode placements were determined solely based on clinical treatment criteria. As a result, implantation sites vary across participants within the MTL. Apart from the anatomical cluster in the left anterior hippocampus, where we observed functional similarities across all participants, the variability in electrode placement limits our ability to make sub-regional MTL functional comparisons in this study.”

C5: Fig. 2D: Is this plot across all electrodes? Or also just the electrodes from the left anterior hippocampus? Are the power increases significant when applying a cluster-based permutation test?

The plot in Fig. 2d includes data from all electrodes across participants. To assess the statistical significance of the observed power increases, we applied a cluster-based permutation test. In the updated figure, significant activity is depicted in saturated colors, while non-significant activity is shown in faded colors. This approach highlights the regions where power increases are statistically significant.

Fig. 2 | Theta dynamics during real-world navigation. d, Average time-frequency (TF) activity aligned with all turns confirms the engagement and the temporal relationship of theta activity preceding turns (time = 0). Significant activity is indicated by saturated colors, while non-significant activity is shown in faded colors.

C6: Fig. 3C: Please clarify what you mean by “weighted sum of all channels.”

Based on a previous reviewer comment (R2, Major C2A) we have revised Fig. 3c to display activity from an exemplary channel rather than the “weighted sum of all channels”. This change ensures that the results in Fig. 3 are independent of the position reconstruction procedure, as all position reconstruction-related results have now been moved to Fig. 4b and c. This separation clarifies the temporal consistency results in Fig. 3 and their relation to the reconstructed position results shown in Fig. 4.

C7: Fig. 3C: As suggested above, it would be good to see a plot showing theta activity of all electrodes as a function of linearized position, given the small sample size.

As mentioned (see response to R3, Major C5), we include this for the real-world navigation condition. For the imagined navigation condition, the linearized position is available as a group-level reconstruction because we pooled all MTL channels for statistical reasons. Given the small sample size, this approach allows us to provide a more comprehensive view of the theta activity across all electrodes.

C8: Fig. 4C: Is this data smoothed across electrodes? If yes, please show the unsmoothed version as smoothing across electrodes (from different hemispheres and subjects) does not seem valid.

We have revised Fig. 4c (moved to Fig. 4f) as suggested. The data is now presented without smoothing across electrodes to ensure that single channels can be more clearly distinguished.

C9: Fig. 4D, middle: The prediction of “imagined” positions seems to be driven by the edge cases. How much of this result can be explained by the time warping procedure that may simply align periods of increased theta power (that may have nothing to do with particular positions on a segment)?

Thank you to the reviewer for raising this important point. The results shown in Fig. 4d (now Fig. 4g), middle panel, represent data that generalizes beyond the alignment and warping procedure. Specifically, the parts of the data illustrated here were not used in the alignment algorithm, making it unlikely that the observed results are solely due to the warping process. Moreover, we would like to refer to the response to R3, Major C9 providing more detailed description of the cross-validation procedure.

We acknowledge that imaginations might be non-continuous, with potential “jumps” and “pauses,” which could contribute to errors in position reconstruction due to accumulation of inaccuracies after resets at salient points⁵. This might be related to alignment to the start and end of imagination trials or to more definite timing associated with turns compared to mid-segments during imagined navigation. We believe these factors contribute to the observed results (see also response to R3, Major C5), rather than the warping procedure alone.

C10: Fig. S2B: It would be useful to add a measure of dispersion.

Thank you to the reviewer for the suggestion, we added a measure of dispersion (\pm SEM) in Fig. 2b (now Extended Data Fig. 5b):

Extended Data Fig. 5 | Frequency spectra and theta dynamics. b, Theta amplitude dynamics and time-resolved theta rates, z-scored for comparison (left), along with theta bout rates (\pm SEM) shown as detection percentages across trials (right) for left (top) and right (bottom) walks. Vertical lines represent turns.

C11: Fig. S2B and Fig. S3: This would be useful to see for remembered (“imagined”) navigation trials, too.

Thank you to the reviewer for the suggestion. As mentioned in our response to R3, Major C5, we have included a comparison of theta power as a function of linearized position for the real-world condition and provided the group-level analysis for the imagined navigation condition. The linearized position is presented as a reconstruction at the group level since we pooled all MTL channels for statistical reasons.

C12: Fig. S4B: What do the colors represent?

We thank the reviewer for pointing this out and apologize for the oversight. The colors in Extended Data Fig. 4b (now Extended Data Fig. 6b) indicate non-significant activities in green and significant activities in non-green colors. We have updated the figure caption to clarify this information:

Extended Data Fig. 6 | Theta activity and timing of effect. b, Statistically significant theta clusters ($p < 0.05$, determined through a cluster-based permutation test). Non-significant periods are shown in green, while significant activities are depicted in non-green colors.

C13: Fig. S6A: Please add a measure of dispersion.

Thank you to the reviewer for the suggestion. We have added a measure of dispersion (\pm SEM) to Extended Data Fig. 6a (now Extended Data Fig. 4a):

Extended Data Fig. 4 | Individual subject analyses. a, Mean theta activity (\pm SEM) across channels for each participant (P1-P5) aligned to turns ($t = 0$). Black bars on top of the plots indicate significant periods.

References:

1. Lega, B. C., Jacobs, J. & Kahana, M. Human hippocampal theta oscillations and the formation of episodic memories. *Hippocampus* **22**, 748–761 (2012).
2. Watrous, A. J. *et al.* A comparative study of human and rat hippocampal low-frequency oscillations during spatial navigation. *Hippocampus* **23**, 656–661 (2013).
3. Bohbot, V. D., Copara, M. S., Gotman, J. & Ekstrom, A. D. Low-frequency theta oscillations in the human hippocampus during real-world and virtual navigation. *Nat Commun* **8**, 14415 (2017).
4. Aghajan, Z. M. *et al.* Theta Oscillations in the Human Medial Temporal Lobe during Real-World Ambulatory Movement. *Curr Biol* **27**, 3743-3751.e3 (2017).
5. Klimesch, W. EEG alpha and theta oscillations reflect cognitive and memory performance: a review and analysis. *Brain Research Reviews* **29**, 169–195 (1999).
6. Niedermeyer, E. & Silva, F. H. L. da. *Electroencephalography: Basic Principles, Clinical Applications, and Related Fields*. (Lippincott Williams & Wilkins, 2005).
7. Buzsáki, G. & Draguhn, A. Neuronal Oscillations in Cortical Networks. *Science* **304**, 1926–1929 (2004).
8. Yoo, H. B., Umbach, G. & Lega, B. Neurons in the human medial temporal lobe track multiple temporal contexts during episodic memory processing. *NeuroImage* **245**, 118689 (2021).
9. Goyal, A. *et al.* Functionally distinct high and low theta oscillations in the human hippocampus. *Nat Commun* **11**, 2469 (2020).
10. Wang, M., Foster, D. J. & Pfeiffer, B. E. Alternating sequences of future and past behavior encoded within hippocampal theta oscillations. *Science* **370**, 247–250 (2020).
11. König, S. D. & Buffalo, E. A. A nonparametric method for detecting fixations and saccades using cluster analysis: Removing the need for arbitrary thresholds. *Journal of Neuroscience Methods* **227**, 121–131 (2014).
12. Agarwal, G. *et al.* Spatially Distributed Local Fields in the Hippocampus Encode Rat Position. *Science* **344**, 626–630 (2014).
13. Stangl, M. *et al.* Boundary-anchored neural mechanisms of location-encoding for self and others. *Nature* **589**, 420–425 (2021).

14. Kahana, M. J., Sekuler, R., Caplan, J. B., Kirschen, M. & Madsen, J. R. Human theta oscillations exhibit task dependence during virtual maze navigation. *Nature* **399**, 781–784 (1999).
15. Bush, D. *et al.* Human hippocampal theta power indicates movement onset and distance travelled. *Proceedings of the National Academy of Sciences* **114**, 12297–12302 (2017).
16. Liu, J. *et al.* Multi-scale goal distance representations in human hippocampus during virtual spatial navigation. *Current Biology* **33**, 2024–2033.e3 (2023).
17. Foster, D. J. & Wilson, M. A. Hippocampal theta sequences. *Hippocampus* **17**, 1093–1099 (2007).
18. Gupta, A. S., van der Meer, M. A. A., Touretzky, D. S. & Redish, A. D. Segmentation of spatial experience by hippocampal theta sequences. *Nat Neurosci* **15**, 1032–1039 (2012).
19. Wikenheiser, A. M. & Redish, A. D. Hippocampal theta sequences reflect current goals. *Nat Neurosci* **18**, 289–294 (2015).
20. Dragoi, G. & Buzsáki, G. Temporal Encoding of Place Sequences by Hippocampal Cell Assemblies. *Neuron* **50**, 145–157 (2006).
21. Kunz, L. *et al.* Hippocampal theta phases organize the reactivation of large-scale electrophysiological representations during goal-directed navigation. *Science Advances* **5**, eaav8192 (2019).
22. Qasim, S. E., Fried, I. & Jacobs, J. Phase precession in the human hippocampus and entorhinal cortex. *Cell* **184**, 3242–3255.e10 (2021).
23. Umbach, G. *et al.* Time cells in the human hippocampus and entorhinal cortex support episodic memory. *Proceedings of the National Academy of Sciences* **117**, 28463–28474 (2020).
24. Zheng, J. *et al.* Hippocampal Theta Phase Precession Supports Memory Formation and Retrieval of Naturalistic Experience in Humans. 2023.06.05.543539 Preprint at <https://doi.org/10.1101/2023.06.05.543539> (2023).
25. Klug, M. & Kloosterman, N. A. Zapline-plus: A Zapline extension for automatic and adaptive removal of frequency-specific noise artifacts in M/EEG. *Human Brain Mapping* **43**, 2743–2758 (2022).
26. Winkler, A. M., Ridgway, G. R., Webster, M. A., Smith, S. M. & Nichols, T. E. Permutation inference for the general linear model. *NeuroImage* **92**, 381–397 (2014).
27. Winkler, A. M., Webster, M. A., Vidaurre, D., Nichols, T. E. & Smith, S. M. Multi-level block permutation. *NeuroImage* **123**, 253–268 (2015).

28. Jutras, M. J., Fries, P. & Buffalo, E. A. Oscillatory activity in the monkey hippocampus during visual exploration and memory formation. *Proceedings of the National Academy of Sciences* **110**, 13144–13149 (2013).
29. Maoz, S. L. L. *et al.* Dynamic neural representations of memory and space during human ambulatory navigation. *Nat Commun* **14**, 6643 (2023).
30. Schacter, D. L., Addis, D. R. & Buckner, R. L. Remembering the past to imagine the future: the prospective brain. *Nat Rev Neurosci* **8**, 657–661 (2007).
31. Comrie, A. E., Frank, L. M. & Kay, K. Imagination as a fundamental function of the hippocampus. *Philos Trans R Soc Lond B Biol Sci* **377**, 20210336 (2022).
32. Buckner, R. L. The Role of the Hippocampus in Prediction and Imagination. *Annual Review of Psychology* **61**, 27–48 (2010).
33. Schacter, D. L. *et al.* The Future of Memory: Remembering, Imagining, and the Brain. *Neuron* **76**, 677–694 (2012).
34. Neupane, S., Fiete, I. & Jazayeri, M. Mental navigation in the primate entorhinal cortex. *Nature* **630**, 704–711 (2024).

Dear Reviewers,

We would like to thank you for your positive feedback and support for the publication of our manuscript. We have incorporated suggestions from Reviewers 2 and 3 into a revised manuscript and have detailed our changes below (in blue text) in response to each point raised. We hope that you find these updates responsive to your feedback.

Reviewer #1 [R1]: *The authors completely responded to all my concerns and suggestions - I have no more to offer. Congratulations on an excellent piece of work and apologies for a delayed review!*

Response: We thank you for your positive feedback on our revised manuscript and are thrilled to hear that we were able to address all of your concerns and suggestions.

Reviewer #2 [R2]: *The revised version of the manuscript and the response to the referees' comments are clear, extensive, and address all our concerns. We recommend the publication of the revised manuscript in Nature Human Behaviour. However, we strongly encourage the authors to include as part of the final version a few analyses and figures that have been presented as part of the referee response but are not part of the revised manuscript. We provide a list of these below, along with a couple of other minor comments.*

Response: We thank you for your positive feedback and for taking the time to review the revised manuscript. We have added your suggested analyses and figures into the final version of the manuscript. Please find details of these additions below.

Comment 1 [C1]: *The table provided as part of the response to C2-C (R2) should be included, along with a reference to it in the main text. This information is critical to understanding the time-wrapping procedure. In addition, the authors could clarify what a negative offset means. Does this imply that the participants started imagining the route before the trial actually started?*

Response: We have now included the aforementioned table (now Extended Data Table 3) in the revised manuscript and reference it in the main text (p. 8):

“For individual participant time shifts, see Extended Data Table 3.”

	Time offset [s]
P1	0.19 ± 1.44
P2	0.06 ± 1.14
P3	-0.76 ± 1.47
P4	-0.21 ± 1.19
P5	0.35 ± 1.09

Extended Data Table 3 | Individual participant time shifts in the time warping procedure. Mean time offsets (± standard deviation) used for aligning imagined and real-world navigation data, as determined by the statistical model. The standard deviation reflects variability across cross-validation iterations.

We have also added clarification as to what a negative offset means (p. 18):

“These time offsets represent delays between corresponding estimated imagined and actual positions (Extended Data Table 3). Positive offsets indicate that neural data associated with imagined turns occurs after real turns, whereas negative offsets suggest it occurs beforehand.”

C2: *The figure showing a comparison between the scalp and iEEG power spectrum, provided in response C5-D (R2), could be an Extended Data Figure, with reference in the main text where the authors discuss possible links to alpha frequency. We also disagree with the authors' comment here that there is minimal overlap: the figure clearly shows peaks that overlap between the scalp and iEEG spectra, including at theta/alpha ranges. We would thus recommend that the claim of minimal correspondence be toned down when including the figure.*

Response: We have now included the aforementioned figure as a supplementary figure (Supplementary Fig. 2) and referenced it in the main text (pp. 11-12):

“Although these frequencies partially overlap with scalp EEG alpha rhythms (Supplementary Fig. 2), oscillations around ~8 Hz in the MTL are considered functionally distinct from alpha activity recorded over visual and sensorimotor areas on the scalp^{1,2}. Further research is needed to clarify the relationship between MTL theta-range activity and scalp recordings.”

Additionally, as suggested, we have toned down the statement regarding minimal overlap between the scalp EEG and intracranial EEG to provide a more accurate description of the data shown in the figure:

Supplementary Fig. 2 | Preliminary analysis of intracranial and scalp EEG. Frequency spectra from individual participants (P1-P5) show peaks in both intracranial and scalp EEG recordings, with some overlap in the lower frequency ranges. Intracranial EEG recorded from the MTL is plotted alongside averaged scalp EEG recorded from posterior occipital electrodes (PO3, PO4). Future studies will be required to investigate the precise relationship between MTL and scalp EEG activity.

Additionally, we have included the necessary information on scalp EEG data acquisition and analysis methods in the Methods section (titled “**Scalp EEG**”, p. 18) to support the new supplementary figure:

“This study primarily focused on iEEG to investigate neural dynamics during real-world and imagined navigation. While scalp EEG data were collected concurrently, the primary analyses presented center on iEEG findings. Future analyses will explore the effects observed in scalp EEG and examine their relationship to the iEEG results discussed in this study, providing a broader understanding of neural activity across recording modalities. For completeness, we briefly describe the scalp EEG data acquisition and analysis methods related to the preliminary analysis presented in Supplementary Fig. 2.

In brief, scalp EEG was recorded using the eego sports system (ANT Neuro Inc.), which includes a mobile amplifier, recording tablet, and Waveguard EEG cap. Data were sampled at 1024 Hz across 64 channels. Due to the participants’ free movement, the scalp EEG data were contaminated with motion and muscular artifacts, and the telemetry required for RNS device access introduced additional noise. To mitigate stereotypical technical artifacts, we applied existing EEG correction methods, including the Zapline-plus³ approach, and rejected epochs with high-frequency muscular activity from further analyses. We analyzed two parieto-occipital channels (PO3 and PO4), selected for their typically pronounced scalp rhythms and reduced neck muscle artifacts compared to more posterior channels during walking.”

C3: The same for the figure provided as part of C2-C (R2) (figure time-on-task). This could also be referenced in the main text.

Response: We believe the reviewer is referring to the time-on-task figure that was included with our previous response to C2-D, not C2-C. Assuming this is correct, we have now included this figure as Extended Data Fig. 6g and referenced it in the main text (p. 6):

“However, we did not find evidence for a time-on-task effect when comparing the first and second halves of control trials (Extended Data Fig. 6g).”

Extended Data Fig. 6 | g, Temporal consistency was computed separately for the first and second halves of the treadmill control condition, with no evidence of a time-on-task effect observed in this control condition.

C4: We recommend that the results of the analysis presented in response to C3-C (R2) be included in the final version of the manuscript. While we understand the figure could cause confusion, the question of how the “average” model works for P2, who follows a different route, is an important one to address in the manuscript.

Response: We agree that addressing how the “average” model applies to participant 2 (P2) is relevant. We have now included this analysis in the manuscript as Supplementary Fig. 1 and have referenced it in the main text (p. 8):

“In these main analyses, we used identical route structures for all participants. However, to account for the specific route followed by participant 2 (P2), we conducted a supplemental analysis that adapted the model to include an additional turn. This adaptation yielded similar results (Supplementary Fig. 1), supporting the robustness of the model across participants.”

This information has also been added to the Methods section (p. 17):

“In our main analyses, we used identical route structures for all participants. To address the specific route followed by participant 2 (P2), we conducted a supplemental analysis that adapted the model to include an additional turn (Supplementary Fig. 1).”

Supplementary Fig. 1 | Modified position reconstruction. The modified cosine route representation for Participant 2 included an additional turn (left panel). Position reconstruction results for the group using this modified model for Participant 2 are shown in the right panel. For comparison, the original model with one fewer turn is shown in Fig. 4g.

C5: Please state which statistical test was used for the p-values in Fig. 4G (Rayleigh test for non-uniformity?).

Response: We tested whether the reconstruction errors were smaller than chance by employing a permutation test procedure, which involved randomly shifting the reconstructed and modeled positions to generate a null distribution. The p-values presented in Fig. 4h are derived from this permutation test, as detailed in the Methods (p. 20):

“We also compared the reconstruction errors of each condition to a null distribution generated by randomly shifting the reconstructed and modeled positions 10,000 times. This approach maintained the temporal

correlation of neighboring samples while simulating random temporal relations between conditions. By doing so, we could more accurately assess the significance of our reconstruction errors and ensure a rigorous comparison.”

C6: *The sentence in lines 207-208 of the revised manuscript (“we observed that theta dynamics were primarily related to the relative position within the route segments ($p = 0.006$), mirroring the patterns observed during real-world navigation.”) can lead to the same confusion that position reconstruction has already been done. Given the clarifications provided by the authors, we assume that none of the results at this point of the manuscript included position reconstruction; if that is the case then “relative position” is misleading. Something along “relative time on trial” would be a more accurate descriptor and would avoid potential confusion for the reader.*

Response: Thank you for your helpful feedback. We agree that the phrase “relative position” could lead to confusion, especially given that position reconstruction had not yet been performed at that point in the manuscript. To clarify, we have revised the relevant section by moving these sentences to the next section titled “Reconstructing Imagined Positions”, as some of the analyses related to eye movements and imagined navigation were, in fact, conducted with the output of the position reconstruction.

The updated text now reads as follows (p. 6):

“Lastly, we did not find a direct effect of eye velocity on theta amplitudes during imaginations ($p = 0.58$), nor evidence for increased theta amplitudes time-locked to saccades during either imagined navigation or control trials (Extended Data Fig. 7d, Extended Data Fig. 8d,e).”

Further, in the “Reconstructing Imagined Positions” section, we clarified that the theta dynamics were related to the reconstructed relative position within route segments (p. 8):

“We observed that theta dynamics were primarily related to the reconstructed relative position within the route segments ($p = 0.006$), mirroring the patterns observed during real-world navigation (Extended Data Fig. 7c,d). Further, the impact of reconstructed relative position during imagined navigation was significantly greater than that of other behavioral variables (e.g., head rotation, hip rotation, eye movement, speed; all $p > 0.05$), suggesting that these other factors did not significantly impact theta dynamics.”

C7: *As an aside, we find the results on saccadic phase reset very intriguing, as they are in obvious tension with published results. A more detailed analysis is needed to reach a clear conclusion, but we do understand that this is not within the scope of the current manuscript.*

Response: We thank you for your comment and agree that a more detailed analysis of the saccadic phase reset is beyond the scope of the current manuscript. Your feedback has highlighted an important area for future investigation, and we plan to explore this aspect further in our upcoming work.

Reviewer #3 [R3]: *The authors have thoroughly addressed all my comments, and I am happy to recommend the manuscript for publication. I would suggest, however, that the authors still address the following points that follow up on my previous comments.*

We thank you for your positive feedback and for recommending our manuscript for publication. Below, we address each additional point raised in detail.

Comment 1 [C1]: *Major comment C5: I could not follow why “for the imagined condition, the positions are reconstructed from pooled channels, and linearized position is only available at the group level.” Please explain.*

Response: For the imagined condition, we reconstructed positions from neural data by combining data from multiple electrodes to improve position estimation. In the real-world navigation condition, both time and position data are available for each electrode, due to the presence of motion tracking. However, in the imagined navigation condition, neural data is only available as a function of time, without direct position information. Therefore, position reconstruction was necessary to facilitate comparisons of theta dynamics between real-world and imagined navigation in relation to position.

Since the position in the imagined condition is derived from group-level reconstruction, the linearized position data is only available at the group level, rather than for individual electrodes. As a result, we were unable to present linearized position data for individual electrodes in the imagined condition. To address this, we included a comparison of theta power as a function of linearized position for the real-world condition and conducted a group-level analysis for the imagined condition. Additionally, we provide further analyses (in response to

subsequent comments) that illustrate the direct correlation of theta dynamics between the two navigation conditions in Extended Data Fig. 6h.

C2: Major comment C6: *The authors' control analyses are useful, but I was hoping to see an additional analysis demonstrating a direct relationship between theta amplitudes during real-world navigation and theta amplitudes during imagined navigation (for example across spatial bins). This would provide the most direct evidence for the authors' claim that there are "analogous functional networks involved in both types of navigation."*

Response: Thank you for your valuable feedback. In response to your suggestion, we have revised our statement from "analogous functional networks" to "analogous anatomical regions," as it was the MTL regions showing the highest temporal consistency during real-world navigation that also exhibited the highest temporal consistency during imagined navigation (Fig. 3e), rather than "networks" per se. Additionally, we have performed a new analysis comparing theta amplitudes at the group level (see R3C1) as a function of linearized position. This analysis shows the direct correlation of spatially binned theta dynamics between the two navigation conditions (Extended Data Fig. 6h).

Together, these analyses provide further evidence for the relationship between theta activity during both types of navigation, supporting the involvement of comparable anatomical regions in real-world and imagined navigation.

Extended Data Fig. 6 | h, Correlation between theta activity during real-world and imagined navigation, binned by relative reconstructed position (each dot represents a 5% bin of the total route segment length). The X and Y axes represent normalized theta activity, pooled across channels after position reconstruction, using both route representation components (-sin, cos) to generate the reconstructed positions. This is calculated for each of the 20 spatial bins, resulting in 40 data points (2 x 20).

C3: Major comment C11: *Could you please clarify what each dot in the scatterplot represents? Is it pooled data across all channels?*

Response: Each dot in the scatterplot in Fig. 4d represents theta activity levels after position reconstruction, for each 0.5-second time bin, derived from pooled data across all channels. This approach allows us to visualize the reconstructed route representations for both real-world and imagined navigation conditions. We have added this clarification to the figure caption:

"Fig. 4 | d, Scatterplot of the reconstructed route representations of real-world and imagined navigation. Each dot represents data averaged over 0.5-second time bins, derived from data pooled across all channels."

C4: Minor comment C5: Could you please provide the statistical output of the cluster-based permutation test? Apologies if I missed it

Response: We apologize for the oversight in not including the statistical output of the cluster-based permutation test in the manuscript. We have now added the relevant statistical details, including test statistics and p-value ($p = 1e-4$) for the cluster-based permutation test. The updated Fig. 2 caption is as follows:

“Fig. 2 | d, Average time-frequency (TF) activity aligned with all turns confirms the engagement and the temporal relationship of theta activity preceding turns (time = 0). Significant activity is indicated by saturated colors (cluster-based permutation test, $p = 1e-4$), while non-significant activity is shown in faded colors.”

References

1. Buzsáki, G. & Draguhn, A. Neuronal Oscillations in Cortical Networks. *Science* **304**, 1926–1929 (2004).
2. Niedermeyer, E. & Silva, F. H. L. da. *Electroencephalography: Basic Principles, Clinical Applications, and Related Fields*. (Lippincott Williams & Wilkins, 2005).
3. Klug, M. & Kloosterman, N. A. Zapline-plus: A Zapline extension for automatic and adaptive removal of frequency-specific noise artifacts in M/EEG. *Human Brain Mapping* **43**, 2743–2758 (2022).